# EFFICIENT FULL-CONTEXT RETRIEVAL FOR LONG DOCUMENTS

## ABSTRACT

Long document question answering is challenging due to the quadratic computational cost of transformer-based LLMs. In resource-constrained environments, Retrieval-Augmented Generation (RAG) uses document chunking to maintain linear computational costs but often loses sight of the global context. We introduce the Mamba retriever 130M and Mamba retriever 1.3B retrievers, capable of processing entire long documents in linear time and integrating earlier context to retrieve relevant sentences for answering questions. Mamba retrievers outperform state-of-the-art embedding models across 41 long-document Q&A benchmarks while maintaining speed and computational efficiency. Their performance is comparable to GPT-4o on long documents over 256k tokens while using significantly fewer tokens. Mamba retrievers are trained on synthetic data generated from our novel link-based method, which enhances the retrievers' ability to leverage long-range document connections. We further demonstrate the effectiveness of our link-based method over baseline synthetic data methods. All code, datasets, and model checkpoints are available at https://github.com/MambaRetriever/MambaRetriever

## 1 INTRODUCTION

Long document question answering remains a significant challenge in natural language processing due to the quadratic computational cost associated with using transformer-based Large Language Models (LLMs) (Vaswani et al., 2017). Retrieval-Augmented Generation (RAG) (Asai et al., 2024) addresses this issue by processing long documents in shorter chunks with transformer-based embedding models, thereby maintaining an approximately linear computational cost relative to context size. These embedding models then retrieve relevant chunks to serve as input for an LLM to generate an answer. However, embedding models that only process shorter chunks may lose important global and contextual information. This limitation has spurred ongoing research into context-aware embedding models (Morris & Rush, 2024).

In this paper, we introduce a new class of retrievers. Mamba retriever 130M and Mamba retriever 1.3B process long documents in their entirety with linear scaling in sequence length. They use their understanding of the entire preceding context to retrieve relevant sentences, as illustrated in Figure1.

The Mamba retriever is the first successful transformation of a state-space model (Dao & Gu, 2024) into a fine-grained, sentence-level retriever for RAG pipelines. It outperforms state-of-the-art embedding models across 41 long-document Q&A benchmarks (see Table 1). The Mamba retriever also maintains faster speed at processing documents and uses fewer FLOPS than embedding models while achieving higher accuracy (see Table 2). The Mamba retriever generalizes well on very long documents, achieving performance close to GPT-4o's full-context capabilities on documents longer than 256k tokens (see Figure 3), while using 160 times fewer tokens[1].

A key contribution of this paper is our novel link-based synthetic data generation method, which discovers real connections within a document and transforms these connections into coherent and realistic questions that require using different parts of the document to answer. When trained with our synthetic data, Mamba retrievers can leverage long-range connections within documents to more

---

[1]On average, the Mamba retriever selects 1,600 tokens from documents longer than 256k tokens: 256k/1,600 = 160 times

accurately identify relevant sentences[2]. Our experiments demonstrate the superiority of our link-based method over baseline synthetic generation methods (see Table 4).

## 2 RELATED WORK

**Long-context Language Models:** Transformer models are inefficient when processing long-context documents because they suffer from quadratic scaling in both training and inference (Liu et al., 2024a). Many works are dedicated to reducing transformer's quadratic complexity while improving global reasoning in long documents. Sparse-attention models such as Longformer (Beltagy et al., 2020) and LongT5 (Guo et al., 2022) achieve linear scaling at the expense of some performance degradation. Other work focuses on using customized synthetic data and architectures to effectively extend the context window size of language models (Zhang et al., 2024c; An et al., 2024b; Luo et al., 2024; Xiong et al., 2024; Yu et al., 2024). For the task of long-document summarization, several techniques are developed to semantically segment documents for summarization (Moro & Ragazzi, 2022; 2023), including iterative (Zhang et al., 2022), recursive summarization (Wu et al., 2021) and memory-enhanced segmentation (Moro et al., 2023).

**State Space Models:** Meanwhile, State Space Models (SSMs) emerge as an alternative to process long sequences (Fu et al., 2024; Gu et al., 2022; Peng et al., 2023; Arora et al., 2024), as they have linear scaling during training and inference. Dao & Gu (2024) incorporated input-dependent parameters into SSMs and integrate efficient parallelizable training and efficient autoregressive inference. Glorioso et al. (2024) and Waleffe et al. (2024) proposed a hybrid Mamba that combines Mamba with attention. Arora et al. (2024) used non-causal prefix-linear-attention to improve model understanding of the global context. Some recent works turn SSMs into embedding models (Hwang et al., 2024; Zhang et al., 2024a). In particular, Saad-Falcon et al. (2024) built a retrieval embedding model from the Monarch Mixer architecture (Fu et al., 2024). This line of work is tangential to our paper because we formulate Mamba retriever as a sentence-level retriever and not an embedding model.

**Retrieval-Augmented Generation (RAG):** The approach of retrieving information using an embedding model followed by generating an answer has been fundamental for processing long texts that exceed the context limits of language models (Nakano et al., 2021; Borgeaud et al., 2022; Wang et al., 2023a; Izacard & Grave, 2020; Huang et al., 2023; Liu et al., 2024c; Xu et al., 2024; Yu et al., 2024). For example, RAG systems have been successfully applied in long-document summarization (Mao et al., 2022; Liu et al., 2024b) and query-centric multi-document retrieval and summarization (Gianluca Moro & Molfetta, 2023; Moro et al., 2022).

**Embedding Models:** Transformer-based embedding models are typically used as retrievers for RAG systems. Previous embedding models focus on semantic understanding and instruction-following (OpenAI, 2024; Wang et al., 2023b; Izacard et al., 2021; Lin et al., 2023; BehnamGhader et al., 2024). Embedding models NV-Embed-v2-7B (Lee et al., 2024), GTE-Qwen2 (Li et al., 2023), Stella (Zhang, 2024) and GritLM (Muennighoff et al., 2024) excel at identifying semantically similar sentences within localized contexts (Muennighoff et al., 2023). However, sentence-level retrieval in long-context tasks requires a more comprehensive approach that needs a global understanding of the entire document. Our proposed Mamba retriever can naturally process the full document and perform sentence-level retrieval based on the global understanding of the document.

## 3 MODEL ARCHITECTURE OF MAMBA RETRIEVER

The Mamba retriever is built using the Mamba-2 architecture (Dao & Gu, 2024), a state space model with linear complexity relative to sequence length. Starting with a pretrained Mamba-2 checkpoint, we remove the language modeling head and replace it with a classification head. It is important to distinguish between Mamba retrievers, which are our trained discriminative retrievers, and Mamba-2 models, which are pretrained language models.

During training, each data point is created by prepending a synthetic query to a document. A binary label, serving as a supervision signal, is assigned to the last token of each sentence in the document. This label indicates whether the preceding sentence can help answer the question. Note that we do

---

[2]See an example in Appendix E.4 where the Mamba retriever uses contextual information.

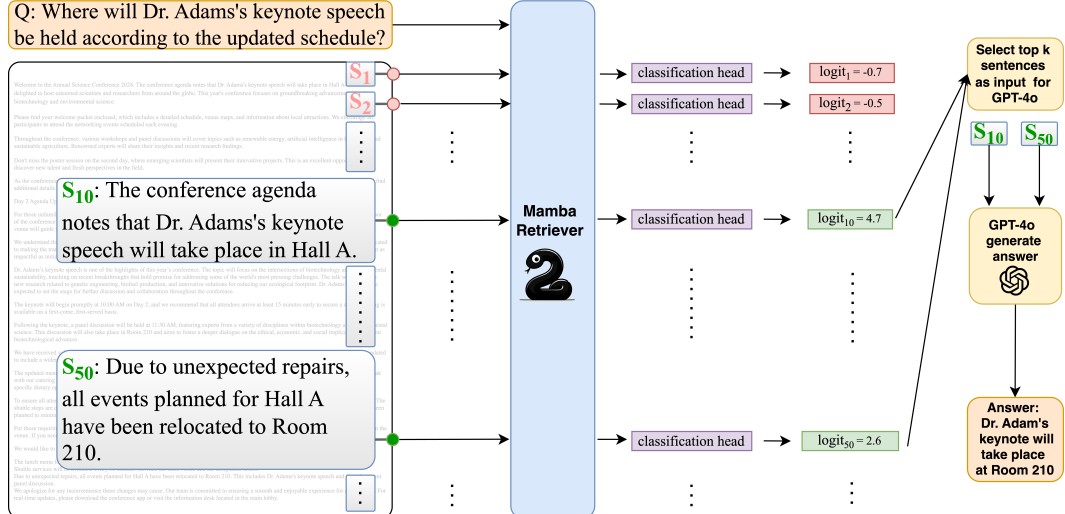

Figure 1: Documents may have long-range dependencies useful for answering questions. On the left, $S_{50}$ is relevant to $Q$ only through its dependency on $S_{10}$. On the right, Mamba models, trained as discriminative retrievers, use a classification head on the last token of each sentence to generate logits. Sentences with the top-k logits are retrieved for an LLM to generate an answer.

not introduce new tokens, such as "eos". For more details on the construction of training data, refer to Section 4.1.

After labeling, we input the entire list of tokens into the model. For the last token of each sentence, the model's classification head produces a logit value. If a sentence is important and can help answer the query, we want the logit value of its last token to be high; otherwise, it should be low. We then compute the average binary cross-entropy loss using the labels and logit values of these tokens.

During the testing phase, the question is prepended to the document, and the combined text is processed by the model. Based on the logit value of the last token in each sentence, we select the top-k sentences to input into a generator model, such as GPT-4o, for final answer generation (see Figure 1). Due to the causal structure of the model, retrieval decisions for each sentence are conditioned on all prior tokens in the document. This approach contrasts with standard embedding-based retrievers, which divide documents into relatively short chunks and process these chunks independently.

In this paper, we train Mamba retrievers to operate at sentence-level resolution and retrieve sentences. However, our model architecture is flexible and can be adapted to other levels of granularity. For example, when using paragraph-level resolution, the model would focus on the final token of each paragraph in the document. See a formal formulation of Mamba retriever in Appendix F.

## 4 SYNTHETIC DATA GENERATION

As mentioned in Section 3, the dataset for training Mamba retrievers must have labeled documents, where each sentence is labeled for its relevance to a query. While long-context documents are available, their sentences lack the relevance labels needed for training Mamba retrievers. To address this, we propose a synthetic data generation method that creates queries from existing long documents and labels sentences. In this section, we introduce our novel link-based synthetic generation strategy and compare it with two baseline approaches: chunk-based and pair-based generation.

**Chunk-based generation** creates a question based on a single chunk of a document. Specifically, we randomly select a chunk consisting of 20 sentences from a real document and use an LLM to synthetically generate a question about the selected chunk. The LLM also labels each sentence in the chunk. Models trained with chunk-based synthetic data can perform localized sentence retrieval but do not learn to integrate information from various parts of the document. See Figure 5 in Appendix B.1 for prompts used in chunk-based generation.

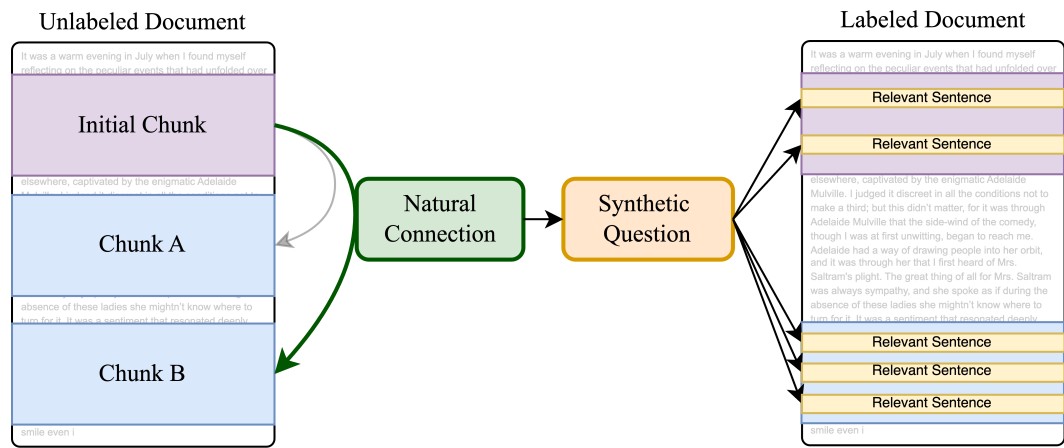

Figure 2: Illustration of the link-based synthetic data generation strategy. This approach first uses an LLM to identify natural connections between different chunks of the document; in the diagram, the initial chunk has a connection to Chunk B but not Chunk A. The LLM then generates a question related to this connection. Finally, the LLM assigns a binary relevance label to each sentence in the two chunks.

**Pair-based generation** is a method for synthetically creating questions that require combining information from two distinct parts of a document. The process begins by dividing a document into non-overlapping chunks of 20 sentences each, which are then embedded using GritLM-7B. For each generation, we first randomly select one chunk and identify a second chunk that has the highest cosine similarity to the first. An LLM is then prompted to generate a question that requires information from both chunks to answer correctly. However, we found that even semantically similar chunks often lack the necessary logical or contextual dependencies to generate realistic and coherent questions. For detailed prompts used in this pair-based generation approach, please refer to Figure 6 in Appendix B.2.

## 4.1 LINK-BASED GENERATION

The link-based generation strategy improves upon pair-based strategy and produces more realistic and coherent questions by leveraging natural connections between different sections of a document.

Since the pair-based strategy fails to identify suitable chunks to form synthetic questions, we directly employ an LLM to explore natural connections between chunks within a long document. Beginning with a randomly selected initial chunk of 20 sentences, the LLM is provided with a context of 200 surrounding sentences and is tasked with identifying any sentences or chunks that have a natural connection to the initial chunk. The LLM then outputs a list of chunks along with their connections to the initial chunk. As illustrated in Figure 2, a synthetic question is generated based on a natural connection, and relevant sentences are labeled. Unlike pair-based generation, link-based generation explores and leverages the actual structure of the document. See Appendix B.3 for prompts used.

GPT-4o-mini-0718 is the LLM used for synthetic data generation. See Table 4 for the computational (i.e. financial) cost of each strategy. See Appendix B.4 for examples of synthetic data generated from each strategy and human evaluation of these synthetic examples.

## 4.2 DATA SOURCES FOR SYNTHETIC DATA GENERATION

The synthetic generation pipeline used long documents which were collected from Project Gutenberg (Project Gutenberg, 2024), government reports dataset used in Huang et al. (2021), finance documents from U.S. Securities and Exchange Commission (2024), and legal contracts (Hendrycks et al., 2021). Synthetic data examples were generated by selecting random subsequences ranging from 2k to 10k tokens from a long document.

### 4.3 DECONTAMINATION FROM TEST SETS:

To prevent contamination between our training and testing data, we implemented the following procedure. First, we divided all documents from our 41 test sets into individual sentences, yielding a set of 2.4 million test sentences. Next, we split the document from each synthetic data point into sentences and calculate the overlap with the test set using string matching. We removed any synthetic data point where more than 1% of its sentences matched those in the set of 2.4 million test sentences. This process effectively eliminated textual overlap between the test and training sets.

## 5 EXPERIMENTAL METHODS

### 5.1 TEST SETS & VALIDATION SETS

For testing, evaluation is done on 41 QA benchmark test sets from Longbench (Bai et al., 2024), ∞ bench (Zhang et al., 2024b), L-eval (An et al., 2024a), LVeval (Yuan et al., 2024), Bamboo (Dong et al., 2024), ELITR bench (Thonet et al., 2024), docfinQA (Reddy et al., 2024) and MuLD (Hudson & Al Moubayed, 2022). The tasks are freeform and multiple-choice questions on long documents, which range from 1,000 to 780,000 tokens. For clarity of presentation, based on document types reported in their original sources, we categorize all 5735 data points from these 41 test sets into 4 categories. Statistics of these 4 categories are provided in Table 7 in Appendix A.1 including number of data points, average document length, number of freeform questions, number of multiple choice questions, average number of answers, and average answer length.

- Educational (1967 data points): Wikipedia, English tests, scientific papers, etc.
- Creative (1733 data points): movie scripts, novels, screenplays, etc.
- Official (1328 data points): legal contract, financial documents, etc.
- Conversational (707 data points): meeting transcripts, dialogues, etc.

**Validation Sets** are taken from the train sets of 8 benchmark tasks, and are only used for hyperparameter tuning. We verified validation sets are completely disjoint from test sets. See details of test sets and validation sets in Appendix A.1, A.2, A.3.

### 5.2 EVALUATED SYSTEMS

**Full-context LLMs:** LLMs such as GPT-4o and Llama 3.1 process the entire document in-context and answer the question directly. We also fine-tuned Mamba-2 for full context answer generation.

**RAG with Mamba retrievers:** Mamba retrievers process the full document in-context, and select the top 50 sentences for an LLM generator. In the Appendix A.4, we also report another setting where Mamba retrievers select the top 10 sentences.

**RAG with embedding retrievers:** For fairness, we consider two setups. **"5 chunks"** setup follows the standard RAG setup (Xu et al., 2023; Li et al., 2024) where the documents were processed in chunks of fixed-length 300 words, and embedding models retrieve the top 5 chunks. **"50 sentences"** setup is the same as Mamba retrievers. Since on average "50 sentences" contain fewer tokens than "5 chunks"(1600 v.s 2000 tokens), "5 chunks" always achieves higher performance for embedding models and BM25 (See Table 2). Therefore, we only report the "5 chunks" setup for both embedding models and BM25 in the main paper for brevity. In Appendix A.4, we also report the "50 sentences" setup for embedding models and BM25.

### 5.3 GPT-4O AS JUDGE

The accuracy of freeform answers is evaluated using GPT-4o-0806, which uses a specialized prompt to compare attempted answers with ground-truth answers, providing a binary "yes" or "no" judgment. The prompt is developed from 100 human-annotated examples. On a separate held-out test set of 180 human-annotated examples, GPT-4o's 180 yes/no judgments have a high agreement with human judgments, achieving a 0.942 macro F1 score. See Appendix C for the prompt.

## 5.4 SLIDING WINDOW

LLMs like GPT-4o have a context length limit of 128k tokens. To standardize evaluation on full-context LLMs, we employ a sliding window approach for documents exceeding 120k tokens. This approach uses a window size of 120k tokens and a stride of 60k tokens. The answers from different windows are then aggregated by the same LLM, which then produces a final answer.

Our Mamba retriever can generalize beyond its training context length of 10k tokens (see Section 7.2). For instance, the Mamba retriever 1.3B can handle up to 256k tokens without memory errors on a single node with $8\times$ 80GB H100. To ensure a fair comparison with GPT-4o, we use the same sliding window approach for Mamba retrievers when documents exceed 120k tokens. This allows both models to operate within the same effective context window. Sentences scored twice have their scores (i.e., logit values) averaged.

## 5.5 FINE-TUNING

**Mamba retrievers:** From checkpoints in Dao & Gu (2024), the Mamba-2-130M model is fine-tuned on 1 million link-based synthetic data, while the Mamba-2-1.3B model is fine-tuned on 400k data, both for one epoch without early stopping. Due to budget constraints and the lack of additional long-context training documents, we created only 1 million link-based data points. We limited the training of Mamba-2-1.3B to 400k data points because we did not observe any improvements in the validation sets when training beyond this amount.

Learning rates were the only hyperparameters optimized on validation sets. On one node with $8 \times$ 80GB H100s, training the 1.3B model took five hours, while the 130M model took three hours with their respective training data sizes. See Appendix D for our hyperparameter settings.

**Embedding Models:** We fine-tuned two embedding models, Contriever-110M (Izacard et al., 2021) and GTE-Qwen2-1.5B (Li et al., 2023), using the same 1 million link-based synthetic data for one epoch. For each query, relevant sentences are treated as positives and irrelevant sentences as negatives. We used the same contrastive loss and applied the same hyperparameter settings (e.g., scheduler, optimizer, temperature $\tau$ in InfoNCE loss) as reported in their original papers, and we optimized learning rates, batch size, and training data size on the same validation sets.

## 5.6 DOCUMENT PROCESSING SPEED AND EFFICIENCY

We evaluate the performance of various retrieval systems using our test sets. Specifically, we measure the average time it takes for a retriever to process a single long document (already on GPU devices), excluding any pre-processing, post-processing, or host-to-device transfer time. For retrieval systems utilizing embedding models, a long document is processed either in batches of sentences or in chunks of 300 words, depending on the retrieval setting. For the embedding models, batches consist of sentences or chunks from the same document. The batch size for these models is selected to maximize token throughput. Sentences and chunks of similar lengths are batched together in order to minimize the number of padding tokens. For the Mamba retrievers, a batch size of 1 is consistently used, as the entire long document must be processed at once.

When embedding models process input in batches larger than size 1, padding is necessary. Since embedding models require larger batch sizes for faster processing, the additional padding results in higher FLOPS. To provide a more informative comparison, we calculate FLOPS for embedding models both with and without padding. Note that the Mamba retriever does not use padding, so the FLOPS remain the same regardless of padding. FLOPS are calculated using standard formulas provided by Kaplan et al. (2020); Dao & Gu (2024).

Our hardware setup includes two Intel Xeon Platinum 8480+ processors (224 logical CPUs) and 8 × 80GB H100 GPUs.

For embedding models, the "50 sentences" setup retrieves an average of 1600 tokens, while the "5 chunks" setup retrieves an average of 2000 tokens. The "50 sentences" setup consistently results in lower accuracy due to retrieving fewer tokens.

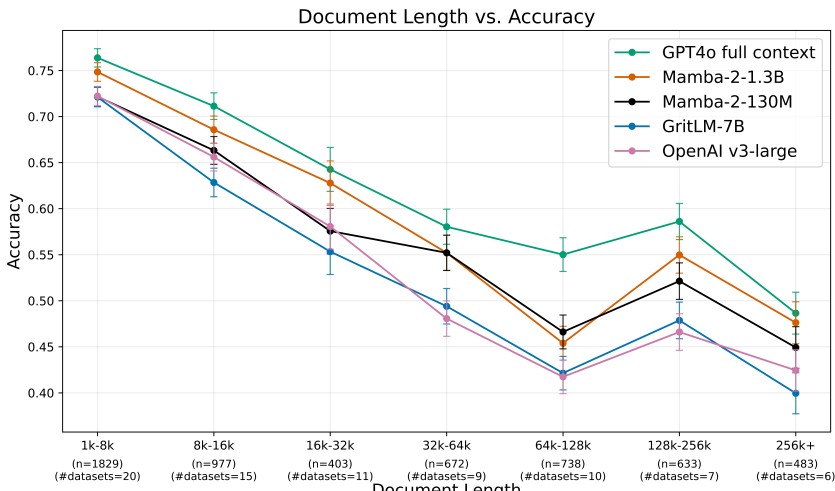

Figure 3: Retrieval models' performance across documents of different lengths with GPT-4o as the generator. Mamba retriever 1.3B approaches GPT-4o full context performance on context over 256k tokens.

Table 1: Average Accuracy is calculated across all data points. FT means fine-tuned. See Section 5.5 for details of fine-tuning embedding models. Mamba retrievers select 50 sentences, while BM25 and embedding models retrieve 5 chunks. Note, chunk-based retrieval performed better than sentence retrieval for embedding models and BM25.

| Retrievers with GPT-4o as Generator | Document Type | | | | Average Accuracy |
|---|---|---|---|---|---|
| | educational n = 1967 | creative n = 1733 | official n = 1328 | conversational n = 707 | |
| BM25 | 62.5 | 37.5 | 46.2 | 41.4 | 49.1 |
| Dragon-110M | 64.9 | 45.1 | 54.1 | 44.6 | 53.9 |
| Contriever-110M | 66.3 | 45.8 | 52.9 | 45.0 | 54.3 |
| Contriever-110M-FT | 65.5 | 48.0 | 55.5 | 41.2 | 54.8 |
| GTE-Qwen2-1.5B | 67.2 | 47.7 | 56.2 | 44.3 | 55.7 |
| GTE-Qwen2-1.5B-FT | 66.9 | 48.0 | 56.2 | 44.8 | 55.8 |
| Stella-1.5B | 66.9 | 50.7 | 54.7 | 47.9 | 56.8 |
| OpenAI v3-large | 68.3 | 50.3 | 57.8 | 48.7 | 57.6 |
| GritLM-7B | 68.3 | 49.7 | 56.2 | 48.7 | 57.2 |
| NV-Embed-v2-7B | 69.7 | 52.7 | 56.3 | 53.2 | 59.1 |
| Mamba retriever 130M | 70.4 | 54.1 | 59.5 | 49.5 | 60.0 |
| Mamba retriever 1.3B | **73.0** | 56.5 | 60.5 | 50.5 | 61.8 |
| GPT-4o Full Context | 71.6 | **62.0** | **62.5** | **62.2** | **64.6** |

## 6 MAIN RESULTS

**Mamba retrievers outperform embedding models:** Table 1 reports model performance grouped by document types and the average performance across all data points, with GPT-4o as generator. Both Mamba retrievers outperform BM25, all embedding baselines and fine-tuned embedding models, including MTEB (Muennighoff et al., 2023) leaders NV-Embed-v2-7B (Lee et al., 2024) and Stella-1.5B (Zhang, 2024). Results on individual dataset performance are in Appendix A.4.

**Mamba retrievers are computationally efficient and fast at processing documents:** Table 2 compares Mamba retrievers with the SoTA embedding models NV-Embed-v2-7B, Stella-1.5B and GTE-Qwen2-1.5B. Section 5.6 explains how speed (i.e., document processing speed) and FLOPS with and without padding are calculated. We see Mamba retriever 1.3B is slightly faster at processing documents and slightly more computationally efficient than embedding models. Mamba retriever 130M is considerably faster and uses much fewer FLOPS due to its small size.

Table 2: Section 5.6 explains speed measurements and FLOPS calculations. FLOPS is calculated with padding when the batch size is larger than 1. FLOPS w/o Pad is calculated without padding. The embedding models are evaluated in two settings: retrieving 50 sentences or 5 chunks of 300 words. Llama-3.1-70B is evaluated as a direct answer generator based on full context of a long document (see Section 7.4). The large FLOPS for Llama-3.1-70B is due to quadratic attention on long sequences.

| Model | Retrieval Setting | Speed (ms) | TFLOPS | TFLOPS w/o Pad | Params (billions) | Average Accuracy |
|---|---|---|---|---|---|---|
| Mamba retriever 130M | 50 sents | **93.4** | **19.0** | **19.0** | **0.1** | 60.0 |
| Mamba retriever 1.3B | 50 sents | 181.6 | 197.9 | 197.9 | 1.3 | **61.8** |
| NV-Embed-v2-7B | 50 sents | 592.0 | 1316.7 | 1279.4 | 7.9 | 56.6 |
| NV-Embed-v2-7B | 5 chunks | 470.8 | 1295.6 | 1287.5 | 7.9 | 59.1 |
| Stella-1.5B | 50 sents | 364.7 | 331.9 | 210.5 | 1.5 | 55.6 |
| Stella-1.5B | 5 chunks | 264.8 | 244.8 | 219.0 | 1.5 | 56.8 |
| GTE-Qwen2-1.5B | 50 sents | 364.4 | 331.9 | 210.5 | 1.5 | 54.6 |
| GTE-Qwen2-1.5B | 5 chunks | 264.9 | 244.8 | 219.0 | 1.5 | 55.7 |
| Llama-3.1-70B | Direct Answer | N/A | 28,517.9 | 28,517.9 | 69.5 | 57.8 |

**Mamba retrievers are robust to different generators:** Table 3 shows average performance when Llama-3.1 8B and 70B are used as the generators. Mamba retrievers continue to outperform the embedding-based retrievers in this setting. Appendix A.5 presents results across individual datasets.

**Mamba retrievers are comparable to GPT-4o on context over 256k tokens:** Figure 3 shows the performance of the Mamba retriever and other baselines across different document lengths. Mamba retriever shows an increasing advantage over embedding baselines as document length increases, and performance converges to GPT-4o for documents longer than 256k. This shows significant length generalization from the model, which was only fine-tuned on documents up to length 10k.

**Link-based synthetic data is more suitable to train state-space models:** Table 1 shows no improvement on GTE-Qwen-1.5B-FT (i.e., fine-tuned) and Contriever-110M-FT over their pre-trained checkpoints, suggesting Mamba retrievers learned more than superficial artifacts such as domain adaptability from training documents.

Table 3: We present performance of retrievers paired with different LLM as generators.

| | Mamba Retriever | | Other Retriever | | | | |
|---|---|---|---|---|---|---|---|
| **Generator** | 1.3B | 130M | BM25 | Dragon | Contriever | OpenAI | GritLM |
| GPT-4o | **61.8** | 60.0 | 49.1 | 53.9 | 54.3 | 57.6 | 57.2 |
| Llama-3.1-70B | **58.8** | 57.5 | 46.9 | 51.9 | 52.9 | 55.1 | 55.4 |
| Llama-3.1-8B | **47.9** | 47.4 | 39.1 | 44.1 | 44.8 | 46.0 | 44.9 |

# 7 ABLATIONS

## 7.1 COMPARING SYNTHETIC DATA STRATEGIES

Table 4: Comparison of synthetic data strategies, Mamba-2-130M is the retriever and GPT-4o is generator.

| | Document Type | | | | | Cost of 1 Million Examples | | |
|---|---|---|---|---|---|---|---|---|
| Strategy | educational n = 1967 | creative n = 1733 | official n = 1328 | conversational n = 707 | Average Accuracy | Input Token (B) | Output Token (B) | Cost ($) |
| Chunk-based | 68.2 | 49.6 | 57.8 | 46.1 | 57.2 | 0.76 | 0.10 | 71 |
| Pair-based | 66.6 | 42.8 | 49.0 | 41.3 | 51.4 | 1.49 | 0.37 | 167 |
| **Link-based** | **69.8** | **51.6** | **59.6** | **50.4** | **59.4** | 7.79 | 1.64 | 1076 |

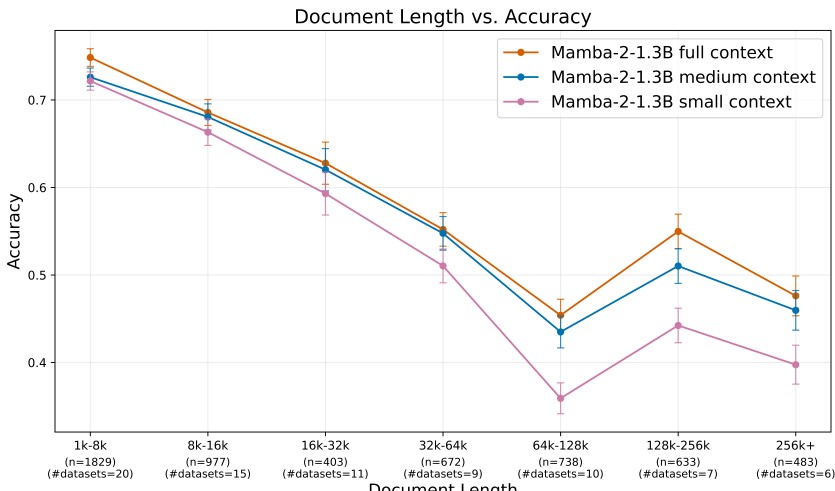

Figure 4: Retrieval performance given varying amounts of document context. GPT-4o is used as the generator.

A main contribution of this paper is our novel link-based synthetic generation strategy. We now evaluate its effectiveness against the two other baseline strategies, chunk-based and pair-based generations. We train 130M Mamba retrievers under identical experimental conditions with 500k synthetic questions created from the same documents by each of the three strategies. Table 4 shows that the link-based strategy achieves the strongest results. Interestingly, pair-based generation strongly underperforms link-based generation, suggesting flaws in its synthetic questions. Refer to Appendix B.4 Table 22 for some flawed synthetic questions generated from the pair-based strategy. By discovering connections between chunks of text in a document, the link-based strategy is able to generate more coherent and contextually relevant questions that would require information from distinct parts of the document.

Note, increasing the amount of training data from 500K to 1M examples does not yield improvement performance for Mamba 130M (59.4 vs. 60.0). This could be attributed to the limited representational power of the lightweight Mamba-130M model or may indicate that the information provided by this synthetic data has reached saturation.

### 7.2 CONTEXT SIZE ABLATIONS

To assess whether long-distance context is improving the performance of the Mamba retriever, we perform ablations on the amount of document context provided to the model. In the small context condition, the document is chunked by sentence. In the medium context condition, the document is chunked at 1024 tokens. After chunking, the model processes each chunk independently, and the retrieval results are aggregated across chunks.

Figure 4 shows model performance for the small and medium context ablations. As document length increases, the ablated Mamba retrievers perform worse relative to the full context retriever. This provides evidence that the model is able to make effective use of long-distance context when performing retrieval.

### 7.3 DISCRIMINATIVE RETRIEVERS VS. GENERATIVE RETRIEVERS

Mamba retrievers are discriminative retrievers that use a classification head to score each sentence. We investigate the feasibility of using an LLM to generate retrieved sentences via next token prediction. We evaluate models on all test sets and report average accuracy. Given a full document, GPT-4o and Llama-3.1-70B are instructed to retrieve relevant sentences for up to 2000 tokens, which is a fair comparison with the "50 sentences" setup in Mamba retrievers. Due to poor generative performance of the pre-trained Mamba-2 checkpoint, we fine-tune Mamba-2-130M to generate relevant sentences using the same 1 million link-based synthetic data. Table 5 shows that all generative retrievers are

Table 5: Average Accuracy is based on all data points from 41 test sets. FT means fine-tuned. Note, Mamba-2-130M-FT is a generative model, whereas Mamba retriever 130M is our proposed discriminative retriever.

| Retriever Type | Generative | | | Discriminative | |
|---|---|---|---|---|---|
| Model | GPT-4o | Llama-3.1 70B | Mamba-2 130M-FT | Mamba retriever 130M | Mamba retriever 1.3B |
| Average Accuracy | 52.2 | 45.9 | 33.5 | 60.0 | 61.8 |

significantly worse than discriminative retrievers. This suggest generative retrieval is often lossy in long-context setting, further showcasing the advantages of discriminative retrieval using Mamba retrievers.

## 7.4 DIRECT ANSWER GENERATION ON FULL CONTEXT.

Table 6: Direct answer generation from full context.

| Model | GPT-4o | Llama-3.1 70B | Llama-3.1 8B | Mamba-2 130M-FT | Mamba-2 130M | Mamba-2 1.3B-FT | Mamba-2 1.3B |
|---|---|---|---|---|---|---|---|
| Average Accuracy | 64.6 | 57.8 | 49.1 | 15.6 | 0.56 | 27.6 | 0.59 |

In Table 6, we investigate whether large language models such as GPT-4o, Llama-3.1-70B, Llama-3.1-8B, Mamba-2-1.3B, and Mamba-2-130M are able to directly answer questions based on the full context of long documents.

GPT-4o has the highest accuracy on test sets. Llama-3.1-70B and Llama-3.1-8B achieve worse performance than Mamba retriever 1.3B or 130M. Note, Table 2 shows Llama-3.1-70B uses 28.5 PFLOPS while Mamba retriever 130M uses 19 TFLOPS.

Pretrained Mamba-2 models are unable to answer questions based on long documents. We fine-tuned the Mamba-2-1.3B and 130M models on the same 1 million link-based data with answers generated by GPT-4o-mini and a conditional language modeling objective. While these fine-tuned checkpoints are considerably better than pretrained counterparts, they have substantially lower performance than Mamba retrievers. This shows it is challenging to train state-space models directly for answer generation on long documents.

## 7.5 FURTHER ANALYSES

In Appendix E.1, we found the increasing distance between linked-chunks and the query improves Mamba retrievers. In Appendix E.2, we found longer training document length of up to 10k tokens improves Mamba retrievers. In Appendix E.3, we observed Mamba retrievers performed slightly worse when important information is located at the end of a long document.

## 8 CONCLUSION & LIMITATION

We introduce the Mamba retriever, which excels in long document sentence-retrieval for RAG systems and approaches GPT-4o's performance for documents over 256k tokens despite being much smaller. The Mamba retriever outperforms all state-of-the-art embedding models, such as NV-Embed-v2, while using fewer FLOPS and processing documents faster. By taking into account all prior document context, the model efficiently leverages long-range dependencies for answering questions about long and complex documents. Our approach eliminates the need for document chunking, a common limitation in current retrieval systems that often results in loss of context and reduced accuracy. Additionally, we propose a novel link-based synthetic data generation strategy that proves most effective for training, helping Mamba retrievers capture long-distance dependencies more effectively. A limitation of our approach is the cost of generating synthetic data points - more than $1000 for one million examples. Finding more efficient methods for producing high-quality training data is essential for future research.

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

**Appendix: Table of Contents**

# A  DATASET

## A.1  TEST SETS & VALIDATION SETS OVERVIEW

For testing, evaluation is done on 41 QA benchmark tasks, which are collected from the following long-document understanding benchmarks: Longbench (Bai et al., 2024), $\infty$ bench (Zhang et al., 2024b), L-eval (An et al., 2024a), LVeval (Yuan et al., 2024), Bamboo (Dong et al., 2024), ELITR bench (Thonet et al., 2024), docfinQA (Reddy et al., 2024), MuLD (Hudson & Al Moubayed, 2022). The tasks are freeform and multiple-choice questions on long documents, which range from 1,000 to 780,000 tokens.

Note, for clarity of presentation in the main paper, we categorize all data points from these 41 datasets into 4 categories based on the type of long documents reported by their original documentation. Each data point can belong to only one category, but within a dataset, data points can be distributed across multiple categories. Benchmark task statistics based on these 4 categories are provided in Table 7. Details of each dataset and the categories it belongs to can be found in Appendix A.2.

- Educational: wikipedia, college exams, English tests, scientific papers, etc.
- Creative: movie scripts, novels, screenplays, etc.
- Official: legal contract, financial documents, etc.
- Conversational: meeting transcripts, dialogues, etc.

We also record model performance averaged across all data points (not averaged by the 4 categories) in the "Average" column in tables. Model performance on each dataset is in Appendix A.4.

For validation sets, we take 100 data points each from the train sets of 8 benchmark tasks. We ensured and verified no document or question from our validation sets are in any test set. Validation sets and test sets are completely disjoint. We did not use the validation sets for training any models or for guiding any synthetic data generation. The sole purpose of our validation sets is for hyperparameter tuning of our Mamba retrievers and embedding models. See full details of validation sets in Appendix A.3.

Table 7: Document type statistics for the 41 benchmark tasks.

| Category | Test Size (n) | Average Length (tokens) | Freeform Questions (n) | Multiple Choice Questions (n) | Number of Answers (n) | Average Answer Length (tokens) |
|---|---|---|---|---|---|---|
| Educational | 1967 | 45,159 | 1595 | 372 | 1.04 | 6.39 |
| Creative | 1733 | 120,585 | 1247 | 486 | 1.00 | 11.16 |
| Official | 1328 | 73,808 | 1156 | 172 | 1.11 | 23.18 |
| Conversational | 707 | 36,922 | 530 | 177 | 1.13 | 4.97 |
| Total | 5735 | 69,119 | 4528 | 1207 | 6061 | 11.67 |

## A.2 TEST SETS STATISTICS

Table 8: Dataset statistics for the 41 benchmark tasks. For larger benchmark tasks, we randomly sampled 200 data point instances from that task.

| Dataset | Category | | | | Test Size | Average Length |
|---|---|---|---|---|---|---|
| | educational | creative | official | conversational | (n) | (tokens) |
| narrativeqa | | ✓ | | | 200 | 30,551 |
| qasper | | | ✓ | | 200 | 5,039 |
| multifieldqa_en | ✓ | ✓ | ✓ | | 147 | 6,951 |
| hotpotqa | ✓ | | | | 200 | 12,802 |
| 2wikimqa | ✓ | | | | 199 | 7,152 |
| musique | ✓ | | | | 200 | 15,560 |
| longbook_choice_eng | | ✓ | | | 200 | 194,984 |
| longdialogue_qa_eng | | | | ✓ | 200 | 109,994 |
| longbook_qa_eng | | ✓ | | | 200 | 195,284 |
| loogle_CR_mixup_16k | ✓ | ✓ | ✓ | | 99 | 31,633 |
| loogle_CR_mixup_32k | ✓ | ✓ | ✓ | | 99 | 50,305 |
| loogle_CR_mixup_64k | ✓ | ✓ | ✓ | | 99 | 96,750 |
| loogle_CR_mixup_128k | ✓ | ✓ | ✓ | | 99 | 177,463 |
| loogle_CR_mixup_256k | ✓ | ✓ | ✓ | | 99 | 339,055 |
| loogle_MIR_mixup_16k | ✓ | ✓ | ✓ | | 139 | 33,240 |
| loogle_MIR_mixup_32k | ✓ | ✓ | ✓ | | 139 | 49,991 |
| loogle_MIR_mixup_64k | ✓ | ✓ | ✓ | | 139 | 97,818 |
| loogle_MIR_mixup_128k | ✓ | ✓ | ✓ | | 139 | 178,771 |
| loogle_MIR_mixup_256k | ✓ | ✓ | ✓ | | 139 | 340,596 |
| multifieldqa_en_mixup_16k | ✓ | ✓ | ✓ | | 101 | 28,194 |
| multifieldqa_en_mixup_32k | ✓ | ✓ | ✓ | | 101 | 52,810 |
| multifieldqa_en_mixup_64k | ✓ | ✓ | ✓ | | 101 | 101,375 |
| multifieldqa_en_mixup_128k | ✓ | ✓ | ✓ | | 101 | 197,624 |
| multifieldqa_en_mixup_256k | ✓ | ✓ | ✓ | | 101 | 390,300 |
| muld_CAC | | ✓ | | | 86 | 48,993 |
| ELITR_Bench_QA | | | | | 130 | 11,147 |
| altqa_4k | ✓ | | | | 199 | 3,223 |
| altqa_16k | ✓ | | | | 199 | 13,011 |
| meetingpred_4k | | | | ✓ | 100 | 3,676 |
| meetingpred_16k | | | | ✓ | 100 | 11,692 |
| meetingqa_4k | | | | ✓ | 86 | 2,731 |
| meetingqa_16k | | | | ✓ | 91 | 9,720 |
| paperqa_4k | | | ✓ | | 82 | 3,114 |
| paperqa_16k | | | ✓ | | 90 | 6,671 |
| tpo | ✓ | | | | 200 | 3,555 |
| financial_qa | | | ✓ | | 68 | 5,050 |
| legal_contract_qa | | | ✓ | | 130 | 25,529 |
| scientific_qa | | | ✓ | | 161 | 4,405 |
| quality | | ✓ | | | 200 | 5,959 |
| coursera | ✓ | | | | 172 | 8,269 |
| docfinQA | | | ✓ | | 200 | 212,751 |

## A.3 VALIDATION SETS STATISTICS

Table 9: Dataset statistics for validation set.

| Dataset | Test Size (n) | Average Length (tokens) |
|---|---|---|
| docfinQA | 100 | 142,328 |
| muld_CAC | 100 | 45,520 |
| ELITR_Bench | 100 | 10,328 |
| narrativeqa | 100 | 71,843 |
| qasper | 100 | 5,274 |
| wiki | 100 | 823 |
| hotpotqa | 100 | 1,313 |
| musique | 100 | 2,230 |

## A.4 GPT-4O AS GENERATOR

Table 10: QA accuracy across 41 datasets with GPT-4o as generator. When not paired with a retriever, GPT-4o is provided with the full document in-context. Results continue to next page.

| Model | Average | narrativeqa | qasper _en | multifieldqa | hotpotqa | 2wikimqa | musique _eng | longbook _choice_eng | longdialogue _qa |
|---|---|---|---|---|---|---|---|---|---|
| | n = 5735 | n = 200 | n = 200 | n = 147 | n = 200 | n = 199 | n = 200 | n = 200 | n = 200 |
| | | | | | **Retrieving 10 sentences** | | | | |
| | | | | | Accuracy | | | | |
| Mamba retriever 1.3B | **54.1** | **44.5** | **48.5** | 81.0 | **72.5** | **74.9** | 57.5 | 60.0 | 21.0 |
| Mamba retriever 130M | 51.8 | 41.0 | 48.0 | **83.7** | 72.0 | 71.9 | 55.0 | 54.0 | **36.5** |
| BM25 | 37.4 | 21.5 | 28.0 | 66.0 | 69.0 | 41.7 | 42.5 | 44.0 | 9.5 |
| Contriever | 46.2 | 30.5 | 32.0 | 70.1 | 63.0 | 56.3 | 45.0 | 61.5 | 8.5 |
| Contriever-FT | 46.6 | 32.5 | 38.0 | 74.1 | 65.5 | 59.8 | 46.0 | 57.5 | 4.5 |
| GritLM | 49.4 | 32.5 | 43.5 | 81.0 | 69.5 | 64.3 | **58.0** | 66.5 | 7.5 |
| NV-Embed-v2-7B | 49.2 | 31.5 | 40.0 | 76.9 | 71.0 | 65.8 | 51.5 | **67.0** | 13.0 |
| Stella-1.5B | 46.8 | 33.0 | 36.0 | 76.2 | 70.0 | 61.8 | 54.5 | 60.5 | 9.0 |
| GTE-Qwen2-1.5B | 46.7 | 30.5 | 40.0 | 72.1 | 64.0 | 61.8 | 54.0 | 64.5 | 11.0 |
| | | | | | F1 | | | | |
| Mamba retriever 1.3B | **29.6** | **17.3** | 29.0 | 46.9 | **39.0** | **41.4** | 30.3 | N/A | 21.2 |
| Mamba retriever 130M | 28.6 | 17.1 | **30.3** | **48.1** | 36.7 | 36.1 | 30.7 | N/A | **37.3** |
| BM25 | 21.6 | 12.0 | 20.8 | 42.3 | 36.8 | 23.2 | 21.6 | N/A | 9.7 |
| Contriever | 25.0 | 13.0 | 22.8 | 41.7 | 30.4 | 30.9 | 24.1 | N/A | 9.5 |
| Contriever-FT | 25.5 | 14.0 | 24.6 | 45.2 | 32.6 | 32.4 | 26.2 | N/A | 5.4 |
| GritLM | 26.6 | 14.5 | 26.8 | 45.7 | 38.3 | 32.8 | 28.2 | N/A | 8.5 |
| NV-Embed-v2-7B | 27.1 | 16.1 | 25.7 | 45.5 | 36.1 | 35.2 | 28.0 | N/A | 12.6 |
| Stella-1.5B | 26.0 | 14.6 | 22.8 | 44.6 | 36.0 | 33.9 | 28.9 | N/A | 9.7 |
| GTE-Qwen2-1.5B | 25.8 | 14.0 | 25.0 | 44.1 | 31.8 | 31.7 | **31.7** | N/A | 10.6 |
| | | | | | **Retrieving 50 sentences** | | | | |
| | | | | | Accuracy | | | | |
| Mamba retriever 1.3B | **61.8** | **57.5** | **57.5** | 83.7 | **82.0** | **84.9** | 63.0 | **75.5** | 29.0 |
| Mamba retriever 130M | 60.0 | 50.0 | 53.5 | **87.1** | 82.0 | 78.9 | 61.0 | 64.0 | **37.0** |
| BM25 | 44.6 | 33.0 | 38.5 | 74.1 | 73.5 | 63.3 | 55.5 | 53.5 | 11.0 |
| Contriever | 53.1 | 37.5 | 48.5 | 79.6 | 75.5 | 73.9 | 58.5 | 67.5 | 16.0 |
| Contriever-FT | 54.8 | 44.5 | 49.5 | 82.3 | 77.0 | 82.4 | 57.5 | 69.5 | 14.5 |
| GritLM | 56.7 | 47.5 | 49.5 | 83.7 | 80.0 | 83.4 | 60.0 | 71.0 | 14.0 |
| NV-Embed-v2-7B | 56.6 | 46.0 | 52.5 | 83.0 | 77.0 | 82.4 | 61.0 | 70.5 | 12.0 |
| Stella-1.5B | 55.6 | 44.5 | 51.0 | 86.4 | 76.0 | 80.9 | 61.5 | 70.5 | 13.5 |
| GTE-Qwen2-1.5B | 54.6 | 46.0 | 49.5 | 83.0 | 75.5 | 77.9 | 58.5 | 74.0 | 15.5 |
| | | | | | F1 | | | | |
| Mamba retriever 1.3B | **32.2** | **20.1** | **32.6** | 46.6 | **48.2** | 46.7 | **37.1** | N/A | 28.4 |
| Mamba retriever 130M | 32.0 | 18.8 | 32.2 | **48.2** | 45.6 | 43.2 | 34.6 | N/A | **36.4** |
| BM25 | 24.8 | 15.8 | 24.8 | 44.9 | 37.5 | 35.5 | 29.0 | N/A | 10.6 |
| Contriever | 28.9 | 16.7 | 30.8 | 45.6 | 38.5 | 38.3 | 32.3 | N/A | 16.7 |
| Contriever-FT | 29.9 | 20.6 | 30.4 | 46.7 | 43.0 | 45.9 | 33.0 | N/A | 13.7 |
| GritLM | 29.9 | 19.6 | 30.3 | **48.2** | 39.9 | **49.1** | 34.0 | N/A | 12.5 |
| NV-Embed-v2-7B | 30.1 | 18.6 | 31.6 | 44.6 | 39.5 | 45.1 | 31.8 | N/A | 11.3 |
| Stella-1.5B | 29.6 | 17.8 | 30.1 | 47.8 | 42.5 | 43.1 | 34.7 | N/A | 12.9 |
| GTE-Qwen2-1.5B | 29.0 | 18.6 | 30.1 | 46.1 | 37.7 | 41.8 | 34.1 | N/A | 14.9 |
| | | | | | **Retrieving 5 chunks** | | | | |
| | | | | | Accuracy | | | | |
| BM25 | 49.1 | 39.5 | 48.0 | 75.5 | 76.5 | 70.4 | 53.5 | 60.5 | 9.5 |
| Contriever | 54.3 | 45.5 | 50.0 | 80.3 | 73.5 | 78.9 | 55.5 | 69.5 | 18.5 |
| Contriever-FT | 54.8 | **53.0** | 49.5 | 80.3 | 71.5 | 77.9 | 52.0 | 69.5 | 8.5 |
| GritLM | 57.2 | 46.5 | 52.5 | **85.0** | 76.5 | 81.4 | **58.5** | 69.0 | 30.0 |
| NV-Embed-v2-7B | **59.1** | 49.5 | 50.5 | **85.0** | 78.5 | 83.4 | 58.0 | 73.5 | **39.0** |
| Stella-1.5B | 56.8 | 43.5 | 48.5 | 83.0 | 75.5 | 81.9 | 52.0 | 74.0 | 22.5 |
| GTE-Qwen2-1.5B | 55.7 | 44.5 | **52.5** | 78.9 | 73.5 | 76.4 | 54.0 | **74.5** | 15.0 |
| | | | | | F1 | | | | |
| BM25 | 26.0 | 17.4 | 29.3 | 42.9 | **41.7** | 39.5 | 30.4 | N/A | 9.2 |
| Contriever | 28.7 | 18.2 | 30.0 | 44.9 | 36.8 | 44.2 | 29.0 | N/A | 18.7 |
| Contriever-FT | 28.1 | 18.8 | 29.2 | **47.3** | 35.9 | 40.7 | 27.4 | N/A | 8.2 |
| GritLM | 29.9 | 18.9 | 29.7 | 46.1 | 39.5 | 45.2 | **31.4** | N/A | 29.8 |
| NV-Embed-v2-7B | **30.6** | 19.1 | 29.8 | 45.0 | 38.5 | 45.1 | 29.1 | N/A | **39.2** |
| Stella-1.5B | 29.9 | 19.0 | 29.1 | 46.6 | 40.0 | **45.8** | 27.8 | N/A | 21.0 |
| GTE-Qwen2-1.5B | 29.2 | **19.1** | **31.3** | 45.1 | 35.2 | 42.0 | 30.2 | N/A | 14.3 |
| | | | | | **Full context** | | | | |
| | | | | | Accuracy | | | | |
| GPT-4o Full Context | 64.6 | 59.0 | 58.5 | 85.7 | 83.0 | 84.9 | 64.5 | 83.5 | 47.0 |
| | | | | | F1 | | | | |
| GPT-4o Full Context | 33.0 | 22.0 | 33.8 | 48.7 | 50.2 | 46.8 | 39.8 | N/A | 40.0 |

Table 11: Results continued

| Model | longbook _qa_eng n = 200 | loogle_CR _mixup_16k n = 99 | loogle_CR _mixup_32k n = 99 | loogle_CR _mixup_64k n = 99 | loogle_CR _mixup_128k n = 99 | loogle_MIR _mixup_16k n = 139 | loogle_CR _mixup_256k n = 99 | loogle_MIR _mixup_32k n = 139 |
|---|---|---|---|---|---|---|---|---|
| | | | | **Retrieving 10 sentences** | | | | |
| | | | | Accuracy | | | | |
| Mamba retriever 1.3B | **41.0** | 36.4 | 37.4 | **39.4** | **40.4** | **32.4** | **41.4** | **36.0** |
| Mamba retriever 130M | 37.5 | **39.4** | 37.4 | 37.4 | 32.3 | 25.9 | 40.4 | 24.5 |
| BM25 | 16.5 | 19.2 | 18.2 | 18.2 | 16.2 | 9.4 | 16.2 | 7.9 |
| Contriever | 24.5 | 32.3 | 34.3 | 30.3 | 31.3 | 27.3 | 27.3 | 26.6 |
| Contriever-FT | 26.5 | 32.3 | 31.3 | 25.3 | 30.3 | 28.1 | 28.3 | 28.1 |
| GritLM | 28.5 | 35.4 | **38.4** | 37.4 | 32.3 | 29.5 | 32.3 | 26.6 |
| NV-Embed-v2-7B | 30.5 | 35.4 | 33.3 | 31.3 | 30.3 | 30.9 | 30.3 | 30.2 |
| Stella-1.5B | 24.5 | 33.3 | 30.3 | 33.3 | 30.3 | 25.9 | 29.3 | 24.5 |
| GTE-Qwen2-1.5B | 24.5 | 32.3 | 35.4 | 37.4 | 37.4 | 21.6 | 38.4 | 20.1 |
| | | | | F1 | | | | |
| Mamba retriever 1.3B | 16.5 | **22.4** | **21.3** | **23.6** | **23.0** | **25.6** | **24.3** | **25.3** |
| Mamba retriever 130M | **18.2** | 21.0 | 21.2 | 21.6 | 19.0 | 23.5 | 20.7 | 22.3 |
| BM25 | 9.5 | 18.2 | 17.9 | 17.2 | 16.3 | 15.2 | 17.3 | 14.4 |
| Contriever | 10.8 | 17.8 | 18.5 | 19.8 | 18.8 | 21.1 | 18.7 | 22.4 |
| Contriever-FT | 12.6 | 19.2 | 19.1 | 18.4 | 19.3 | 21.3 | 17.0 | 21.5 |
| GritLM | 14.4 | 20.3 | 20.6 | 22.2 | 20.8 | 21.8 | 20.1 | 22.3 |
| NV-Embed-v2-7B | 13.9 | 19.7 | 20.4 | 20.5 | 19.4 | 23.6 | 19.5 | 23.4 |
| Stella-1.5B | 14.2 | 20.1 | 18.9 | 20.0 | 19.9 | 21.1 | 20.4 | 20.1 |
| GTE-Qwen2-1.5B | 12.9 | 20.1 | 20.6 | 20.9 | 20.6 | 19.1 | 20.7 | 19.3 |
| | | | | **Retrieving 50 sentences** | | | | |
| | | | | Accuracy | | | | |
| Mamba retriever 1.3B | **54.0** | **56.6** | **51.5** | **46.5** | **51.5** | 43.2 | **48.5** | 41.0 |
| Mamba retriever 130M | 50.0 | 55.6 | **51.5** | **46.5** | 47.5 | **45.3** | 47.5 | **46.0** |
| BM25 | 26.5 | 31.3 | 25.3 | 22.2 | 26.3 | 16.5 | 25.3 | 14.4 |
| Contriever | 33.5 | 40.4 | 35.4 | 40.4 | 38.4 | 36.0 | 41.4 | 30.9 |
| Contriever-FT | 39.0 | 41.4 | 39.4 | 35.4 | 39.4 | 34.5 | 34.3 | 34.5 |
| GritLM | 40.5 | 46.5 | 43.4 | **46.5** | 39.4 | 33.8 | 44.4 | 36.7 |
| NV-Embed-v2-7B | 42.0 | 51.5 | 46.5 | **46.5** | **51.5** | 35.3 | 44.4 | 38.1 |
| Stella-1.5B | 37.0 | 45.5 | 44.4 | 44.4 | 47.5 | 33.8 | 40.4 | 35.3 |
| GTE-Qwen2-1.5B | 37.0 | 46.5 | 39.4 | 37.4 | 39.4 | 33.8 | 35.4 | 33.1 |
| | | | | F1 | | | | |
| Mamba retriever 1.3B | **23.7** | 24.3 | 23.7 | **24.8** | 24.7 | **26.2** | 23.1 | **29.0** |
| Mamba retriever 130M | 23.1 | **25.1** | **25.5** | 24.6 | 24.2 | 25.1 | 24.3 | 27.9 |
| BM25 | 13.0 | 19.6 | 17.8 | 16.4 | 19.8 | 17.6 | 17.2 | 15.4 |
| Contriever | 15.7 | 23.0 | 23.1 | 23.3 | 22.7 | 22.4 | 20.1 | 21.7 |
| Contriever-FT | 18.0 | 25.1 | 21.7 | 22.1 | 23.4 | 25.7 | 21.7 | 25.4 |
| NV-Embed-v2-7B | 18.3 | 23.4 | 23.8 | 22.8 | **25.0** | 25.2 | 22.2 | 27.1 |
| Stella-1.5B | 18.5 | 24.6 | 21.5 | 23.4 | 23.6 | 24.5 | **23.8** | 24.0 |
| GTE-Qwen2-1.5B | 15.9 | 24.4 | 24.9 | 22.5 | 19.9 | 21.9 | 20.9 | 22.4 |
| | | | | **Retrieving 5 chunks** | | | | |
| | | | | Accuracy | | | | |
| BM25 | 26.5 | 36.4 | 44.4 | 35.4 | 35.4 | 18.7 | 35.4 | 18.0 |
| Contriever | 40.0 | 48.5 | 51.5 | 42.4 | 43.4 | 24.5 | 41.4 | 23.0 |
| Contriever-FT | 40.5 | 49.5 | 43.4 | 41.4 | 39.4 | 30.2 | 45.5 | 28.8 |
| GritLM | 48.5 | 52.5 | 51.5 | 45.5 | 48.5 | 31.7 | 46.5 | **36.7** |
| NV-Embed-v2-7B | **49.5** | 52.5 | **52.5** | 56.6 | 58.6 | 30.9 | 50.5 | 34.5 |
| Stella-1.5B | 46.5 | **54.5** | 47.5 | 49.5 | 50.5 | **34.5** | 44.4 | 31.7 |
| GTE-Qwen2-1.5B | 41.5 | 49.5 | 49.5 | 41.4 | 42.4 | 29.5 | 39.4 | 29.5 |
| | | | | F1 | | | | |
| BM25 | 12.7 | 21.0 | 21.1 | 19.6 | 20.2 | 17.4 | 19.2 | 14.8 |
| Contriever | 18.6 | 22.4 | 23.3 | 23.5 | 21.2 | 19.4 | 20.6 | 21.3 |
| Contriever-FT | 19.4 | 21.0 | 20.5 | 21.0 | 21.3 | 22.4 | 20.3 | 19.1 |
| GritLM | 20.1 | 21.9 | 22.1 | 22.7 | 20.4 | 22.5 | 21.4 | 22.5 |
| NV-Embed-v2-7B | **20.6** | 22.0 | 22.4 | 21.3 | 22.9 | **23.3** | **22.9** | **23.1** |
| Stella-1.5B | 20.5 | 24.0 | **24.6** | **24.2** | **23.1** | 21.1 | 21.7 | 21.8 |
| GTE-Qwen2-1.5B | 17.9 | **24.1** | 23.8 | 20.7 | 21.6 | 22.0 | 21.7 | 21.6 |
| | | | | **Full context** | | | | |
| | | | | Accuracy | | | | |
| GPT-4o Full Context | 66.5 | 51.5 | 59.6 | 48.5 | 57.6 | 48.2 | 49.5 | 46.0 |
| | | | | F1 | | | | |
| GPT-4o Full Context | 19.8 | 26.9 | 27.2 | 22.3 | 19.2 | 29.8 | 13.1 | 27.8 |

Table 12: Results continued

| Model | loogle_MIR _mixup_64k n = 139 | loogle_MIR _mixup_128k n = 139 | loogle_MIR _mixup_256k n = 139 | multifieldqa_en _mixup_16k n = 101 | multifieldqa_en _mixup_32k n = 101 | multifieldqa_en _mixup_64k n = 101 | multifieldqa_en _mixup_128k n = 101 |
|---|---|---|---|---|---|---|---|
| **Retrieving 10 sentences** | | | | | | | |
| *Accuracy* | | | | | | | |
| Mamba retriever 1.3B | **33.1** | **34.5** | **32.4** | **52.5** | 55.4 | **53.5** | 52.5 |
| Mamba retriever 130M | 25.9 | 25.2 | 23.7 | 51.5 | **56.4** | 52.5 | **53.5** |
| BM25 | 8.6 | 5.0 | 5.8 | 36.6 | 38.6 | 38.6 | 34.7 |
| Contriever | 25.2 | 27.3 | 25.9 | 47.5 | 45.5 | 46.5 | 45.5 |
| Contriever-FT | 25.2 | 23.0 | 22.3 | 48.5 | 51.5 | 47.5 | 43.6 |
| GritLM | 28.8 | 30.2 | 27.3 | 46.5 | 45.5 | 42.6 | 40.6 |
| NV-Embed-v2-7B | 28.1 | 29.5 | 29.5 | 47.5 | 48.5 | 46.5 | 47.5 |
| Stella-1.5B | 23.7 | 24.5 | 25.2 | 45.5 | 41.6 | 48.5 | 43.6 |
| GTE-Qwen2-1.5B | 20.1 | 20.1 | 19.4 | 45.5 | 38.6 | 36.6 | 42.6 |
| *F1* | | | | | | | |
| Mamba retriever 1.3B | **25.2** | **25.7** | **25.1** | **28.7** | 29.7 | **29.4** | 29.0 |
| Mamba retriever 130M | 23.3 | 22.2 | 22.1 | 26.9 | **31.1** | 29.3 | **30.1** |
| BM25 | 14.1 | 12.0 | 12.4 | 26.1 | 23.6 | 22.9 | 24.9 |
| Contriever | 19.9 | 21.6 | 20.4 | 26.5 | 26.3 | 26.8 | 27.9 |
| Contriever-FT | 23.3 | 21.7 | 18.9 | 27.7 | 28.1 | 26.5 | 27.5 |
| GritLM | 23.3 | 23.5 | 22.4 | 25.7 | 26.9 | 27.1 | 26.2 |
| NV-Embed-v2-7B | 22.3 | 21.5 | 23.6 | 26.9 | 27.3 | 28.1 | 26.2 |
| Stella-1.5B | 21.5 | 21.5 | 22.0 | 26.0 | 26.8 | 27.0 | 26.2 |
| GTE-Qwen2-1.5B | 19.9 | 19.0 | 19.5 | 27.8 | 26.4 | 25.9 | 27.0 |
| **Retrieving 50 sentences** | | | | | | | |
| *Accuracy* | | | | | | | |
| Mamba retriever 1.3B | 43.9 | **42.4** | 41.0 | **55.4** | 59.4 | 49.5 | 52.5 |
| Mamba retriever 130M | 39.6 | 39.6 | 37.4 | 54.5 | 59.4 | 54.5 | 58.4 |
| BM25 | 15.8 | 12.2 | 11.5 | 42.6 | 40.6 | 43.6 | 45.5 |
| Contriever | **73.9** | 25.0 | **44.6** | 20.0 | **87.2** | **78.0** | **81.7** |
| Contriever-FT | 30.9 | 33.8 | 25.9 | 49.5 | 57.4 | 48.5 | 54.5 |
| GritLM | 35.3 | 32.4 | 37.4 | 51.5 | 50.5 | 47.5 | 47.5 |
| NV-Embed-v2-7B | 32.4 | 36.7 | 31.7 | 48.5 | 51.5 | 46.5 | 49.5 |
| Stella-1.5B | 34.5 | 33.8 | 29.5 | 53.5 | 56.4 | 48.5 | 51.5 |
| GTE-Qwen2-1.5B | 30.2 | 33.1 | 32.4 | 47.5 | 52.5 | 53.5 | 47.5 |
| *F1* | | | | | | | |
| Mamba retriever 1.3B | **26.5** | 25.7 | **26.4** | 28.6 | 28.8 | 27.2 | 28.2 |
| Mamba retriever 130M | 26.2 | 26.4 | 26.2 | **30.8** | **30.7** | **29.6** | **30.9** |
| BM25 | 16.0 | 15.4 | 13.9 | 27.4 | 25.9 | 25.6 | 26.6 |
| Contriever | 22.8 | 22.0 | 22.8 | 29.1 | 27.2 | 26.9 | 27.8 |
| Contriever-FT | 25.4 | 23.8 | 22.2 | 28.0 | 29.1 | 27.2 | 28.5 |
| GritLM | 23.3 | 23.9 | 24.2 | 27.9 | 27.6 | 28.9 | 27.2 |
| NV-Embed-v2-7B | 25.6 | **26.6** | 25.4 | 28.5 | 27.1 | 27.0 | 26.8 |
| Stella-1.5B | 23.7 | 25.5 | 24.2 | 28.0 | 28.1 | 28.2 | 28.6 |
| GTE-Qwen2-1.5B | 22.1 | 24.1 | 23.8 | 29.2 | 26.3 | 28.3 | 27.1 |
| **Retrieving 5 chunks** | | | | | | | |
| *Accuracy* | | | | | | | |
| BM25 | 12.2 | 15.8 | 13.7 | 44.6 | 52.5 | 54.5 | 53.5 |
| Contriever | 25.9 | 24.5 | 25.2 | 50.5 | **53.5** | 57.4 | 55.4 |
| Contriever-FT | 29.5 | 23.0 | 25.9 | 51.5 | 50.5 | 51.5 | 47.5 |
| GritLM | 27.3 | 30.9 | 27.3 | 45.5 | 51.5 | **62.4** | 53.5 |
| NV-Embed-v2-7B | 31.7 | **35.3** | 30.9 | **52.5** | 50.5 | 49.5 | 50.5 |
| Stella-1.5B | 32.4 | 28.1 | **30.9** | 48.5 | 52.5 | 59.4 | 51.5 |
| GTE-Qwen2-1.5B | **34.5** | 28.1 | 28.8 | 51.5 | 51.5 | 53.5 | **58.4** |
| *F1* | | | | | | | |
| BM25 | 15.9 | 16.1 | 14.5 | 27.6 | 26.8 | 28.1 | 29.6 |
| Contriever | 21.4 | 20.5 | 22.1 | 28.1 | 29.4 | 29.0 | 29.1 |
| Contriever-FT | 21.9 | 19.0 | 21.3 | 28.9 | 27.5 | 28.1 | 26.6 |
| GritLM | 20.7 | **23.7** | 21.8 | 26.7 | 28.7 | 28.4 | **30.0** |
| NV-Embed-v2-7B | 22.2 | 23.2 | 20.5 | 29.9 | 27.5 | 27.9 | 28.6 |
| Stella-1.5B | **23.0** | 22.4 | **22.3** | 27.5 | 28.3 | **31.1** | 29.6 |
| GTE-Qwen2-1.5B | 21.4 | 22.8 | 20.6 | **30.1** | **29.6** | 29.7 | 28.9 |
| **Full context** | | | | | | | |
| *Accuracy* | | | | | | | |
| GPT-4o Full Context | 45.3 | 41.0 | 36.7 | 57.4 | 62.4 | 55.4 | 56.4 |
| *F1* | | | | | | | |
| GPT-4o Full Context | 27.7 | 20.5 | 13.3 | 32.2 | 32.3 | 28.8 | 24.9 |

Table 13: Results continued

| Model | altqa _4k n = 199 | altqa _16k n = 199 | meetingpred _4k n = 100 | multifieldqa_en _mixup_256k n = 101 | meetingpred _16k n = 100 | meetingqa _4k n = 86 | meetingqa _16k n = 91 | paperqa _4k n = 82 | paperqa _16k n = 90 |
|---|---|---|---|---|---|---|---|---|---|
| **Retrieving 10 sentences** | | | | | | | | | |
| Accuracy | | | | | | | | | |
| Mamba retriever 1.3B | **77.4** | 71.4 | 26.0 | **58.4** | **21.0** | 80.2 | 76.9 | **84.1** | 81.1 |
| Mamba retriever 130M | 73.9 | 69.8 | 18.0 | 49.5 | 19.0 | 80.2 | 74.7 | 80.5 | 77.8 |
| BM25 | 66.3 | 55.3 | 14.0 | 32.7 | 14.0 | 84.9 | 72.5 | 82.9 | **83.3** |
| Contriever | 73.9 | 73.9 | 25.0 | 44.6 | 20.0 | **87.2** | 78.0 | 81.7 | 78.9 |
| Contriever-FT | 74.9 | 69.8 | 19.0 | 45.5 | 20.0 | 79.1 | **80.2** | 78.0 | 81.1 |
| GritLM | **77.4** | 70.4 | 26.0 | 45.5 | 19.0 | 82.6 | 78.0 | 82.9 | 81.1 |
| NV-Embed-v2-7B | 75.4 | 71.9 | 30.0 | 45.5 | 20.0 | 81.4 | 75.8 | 81.7 | 80.0 |
| Stella-1.5B | 70.9 | 73.4 | **34.0** | 43.6 | 14.0 | 77.9 | 74.7 | 79.3 | 80.0 |
| GTE-Qwen2-1.5B | 72.4 | **74.4** | 28.0 | 37.6 | 18.0 | 79.1 | 79.1 | 78.0 | 80.0 |
| F1 | | | | | | | | | |
| Mamba retriever 1.3B | **77.4** | 71.4 | 26.3 | 28.4 | **19.9** | N/A | N/A | N/A | N/A |
| Mamba retriever 130M | 73.9 | 69.8 | 17.0 | **29.5** | 17.7 | N/A | N/A | N/A | N/A |
| BM25 | 66.3 | 55.3 | 11.8 | 24.8 | 13.0 | N/A | N/A | N/A | N/A |
| Contriever | 73.9 | 73.9 | 22.1 | 26.6 | 17.7 | N/A | N/A | N/A | N/A |
| Contriever-FT | 74.9 | 69.8 | 16.4 | 27.6 | 18.1 | N/A | N/A | N/A | N/A |
| GritLM | **77.4** | 70.4 | 23.4 | 26.8 | 17.1 | N/A | N/A | N/A | N/A |
| NV-Embed-v2-7B | 75.4 | 71.9 | 28.0 | 28.0 | 16.8 | N/A | N/A | N/A | N/A |
| Stella-1.5B | 70.9 | 73.4 | **29.4** | 25.8 | 12.2 | N/A | N/A | N/A | N/A |
| GTE-Qwen2-1.5B | 72.4 | **74.4** | 25.6 | 24.9 | 16.9 | N/A | N/A | N/A | N/A |
| **Retrieving 50 sentences** | | | | | | | | | |
| Accuracy | | | | | | | | | |
| Mamba retriever 1.3B | 81.9 | **79.9** | 37.0 | 50.5 | **35.0** | 84.9 | **84.6** | 85.4 | 85.6 |
| Mamba retriever 130M | **83.9** | 77.9 | 28.0 | **55.4** | 24.0 | 82.6 | 78.0 | 80.5 | 85.6 |
| BM25 | 72.9 | 67.8 | 29.0 | 40.6 | 14.0 | 77.9 | 79.1 | 81.7 | 82.2 |
| Contriever | 83.4 | 75.9 | 30.0 | 48.5 | 26.0 | 84.9 | 80.2 | 80.5 | 84.4 |
| Contriever-FT | 83.4 | 73.9 | 37.0 | 47.5 | 21.0 | 82.6 | 83.5 | 84.1 | 85.6 |
| GritLM | 81.4 | 74.4 | 29.0 | 47.5 | 27.0 | **88.4** | 80.2 | **86.6** | **86.7** |
| NV-Embed-v2-7B | 80.4 | 76.9 | **39.0** | 50.5 | 27.0 | 84.9 | 80.2 | 79.3 | **86.7** |
| Stella-1.5B | 79.9 | 75.4 | 31.0 | 53.5 | 23.0 | 81.4 | 75.8 | **86.6** | 82.2 |
| GTE-Qwen2-1.5B | 81.4 | 73.9 | 33.0 | 51.5 | 21.0 | 80.2 | 80.2 | 81.7 | 85.6 |
| F1 | | | | | | | | | |
| Mamba retriever 1.3B | 81.9 | **79.9** | **33.0** | 27.0 | **31.7** | N/A | N/A | N/A | N/A |
| Mamba retriever 130M | **83.9** | 77.9 | 24.4 | **30.7** | 22.2 | N/A | N/A | N/A | N/A |
| BM25 | 72.9 | 67.8 | 24.1 | 26.0 | 12.6 | N/A | N/A | N/A | N/A |
| Contriever | 83.4 | 75.9 | 26.6 | 27.9 | 23.9 | N/A | N/A | N/A | N/A |
| Contriever-FT | 83.4 | 73.9 | 31.7 | 28.7 | 19.1 | N/A | N/A | N/A | N/A |
| GritLM | 81.4 | 74.4 | 24.9 | 26.8 | 24.5 | N/A | N/A | N/A | N/A |
| NV-Embed-v2-7B | 80.4 | 76.9 | 35.5 | 28.0 | 24.5 | N/A | N/A | N/A | N/A |
| Stella-1.5B | 79.9 | 75.4 | 28.1 | 27.9 | 19.6 | N/A | N/A | N/A | N/A |
| GTE-Qwen2-1.5B | 81.4 | 73.9 | 29.8 | 27.0 | 18.9 | N/A | N/A | N/A | N/A |
| **Retrieving 5 chunks** | | | | | | | | | |
| Accuracy | | | | | | | | | |
| BM25 | 69.8 | 60.8 | 33.0 | 50.5 | 18.0 | 81.4 | **84.6** | 79.3 | 82.2 |
| Contriever | 79.4 | 72.4 | 34.0 | 50.5 | 21.0 | **83.7** | 79.1 | 80.5 | 84.4 |
| Contriever-FT | 80.9 | 70.9 | 36.0 | **56.4** | 20.0 | 80.2 | 81.3 | 79.3 | **88.9** |
| GritLM | 80.9 | 74.9 | 39.0 | 49.5 | 23.0 | 82.6 | 78.0 | **82.9** | 81.1 |
| NV-Embed-v2-7B | 81.9 | **79.4** | 40.0 | 52.5 | **27.0** | 83.7 | 81.3 | 80.5 | 84.4 |
| Stella-1.5B | 80.9 | 72.9 | **42.0** | 50.5 | 24.0 | 80.2 | 81.3 | 80.5 | 83.3 |
| GTE-Qwen2-1.5B | **82.4** | 75.4 | 31.0 | 51.5 | 25.0 | 82.6 | 81.3 | 81.7 | 83.3 |
| F1 | | | | | | | | | |
| BM25 | 69.3 | 60.8 | 27.5 | 27.6 | 16.9 | N/A | N/A | N/A | N/A |
| Contriever | 79.4 | 72.4 | 29.9 | 29.4 | 18.6 | N/A | N/A | N/A | N/A |
| Contriever-FT | 80.9 | 70.9 | 31.4 | 29.0 | 17.2 | N/A | N/A | N/A | N/A |
| GritLM | 80.9 | 74.9 | 33.8 | 29.3 | 20.2 | N/A | N/A | N/A | N/A |
| NV-Embed-v2-7B | 81.9 | **79.4** | **34.6** | 28.7 | **24.7** | N/A | N/A | N/A | N/A |
| Stella-1.5B | 80.9 | 72.9 | 35.8 | **29.9** | 21.1 | N/A | N/A | N/A | N/A |
| GTE-Qwen2-1.5B | **82.4** | 75.4 | 27.9 | 29.0 | 23.2 | N/A | N/A | N/A | N/A |
| **Full context** | | | | | | | | | |
| Accuracy | | | | | | | | | |
| GPT-4o Full Context | 81.9 | 77.4 | 57.0 | 51.5 | 52.0 | 83.7 | 75.8 | 84.1 | 83.3 |
| F1 | | | | | | | | | |
| GPT-4o Full Context | 81.9 | 77.4 | 50.0 | 21.5 | 46.1 | N/A | N/A | N/A | N/A |

Table 14: Results continued

| Model | tpo | financial _qa | legal_contract _qa | scientific _qa | quality _qa | coursera | docfinQA | muld _CAC | ELITR _Bench |
|---|---|---|---|---|---|---|---|---|---|
| | n = 200 | n = 68 | n = 130 | n = 161 | n = 200 | n = 172 | n = 200 | n = 86 | n = 130 |
| **Retrieving 10 sentences** | | | | | | | | | |
| *Accuracy* | | | | | | | | | |
| Mamba retriever 1.3B | 81.0 | 66.2 | **64.6** | **53.4** | 65.5 | 73.8 | **32.5** | 76.7 | **49.2** |
| Mamba retriever 130M | 80.0 | **67.6** | 59.2 | **53.4** | 61.5 | 74.4 | 29.5 | 84.9 | 46.2 |
| BM25 | 83.5 | 38.2 | 31.5 | 32.9 | 56.5 | 73.3 | 0.5 | 72.1 | 27.7 |
| Contriever | 87.5 | 30.9 | 52.3 | 37.9 | 61.0 | 79.1 | 7.5 | 81.4 | 29.2 |
| Contriever-FT | 76.0 | 41.2 | 61.5 | 44.7 | **68.0** | 74.4 | 11.0 | 79.1 | 36.9 |
| GritLM | **88.0** | 48.5 | 58.5 | 40.4 | 62.5 | **80.2** | 20.5 | 83.7 | 34.6 |
| NV-Embed-v2-7B | 79.5 | 57.4 | 56.9 | 43.5 | 62.5 | 73.8 | 17.5 | 84.9 | 35.4 |
| Stella-1.5B | 80.0 | 41.2 | 60.8 | 42.2 | 62.5 | 72.1 | 17.5 | 80.2 | 31.5 |
| GTE-Qwen2-1.5B | 78.5 | 52.9 | 53.1 | 41.0 | 66.5 | 72.1 | 15.0 | **86.0** | 33.1 |
| *F1* | | | | | | | | | |
| Mamba retriever 1.3B | N/A | **43.2** | **24.2** | **28.7** | N/A | N/A | 2.1 | N/A | **24.8** |
| Mamba retriever 130M | N/A | 42.5 | **24.2** | 27.4 | N/A | N/A | **2.4** | N/A | 23.1 |
| BM25 | N/A | 34.6 | 19.9 | 18.1 | N/A | N/A | 0.5 | N/A | 18.0 |
| Contriever | N/A | 35.9 | 22.6 | 20.0 | N/A | N/A | 1.0 | N/A | 18.0 |
| Contriever-FT | N/A | 35.1 | 24.7 | 24.2 | N/A | N/A | 1.6 | N/A | 20.2 |
| GritLM | N/A | 36.1 | 22.8 | 22.4 | N/A | N/A | 2.1 | N/A | 20.3 |
| NV-Embed-v2-7B | N/A | 40.2 | 23.6 | 23.8 | N/A | N/A | 2.0 | N/A | 21.4 |
| Stella-1.5B | N/A | 36.5 | 22.9 | 23.5 | N/A | N/A | 2.3 | N/A | 19.7 |
| GTE-Qwen2-1.5B | N/A | 35.7 | 23.5 | 22.8 | N/A | N/A | 1.5 | N/A | 19.9 |
| **Retrieving 50 sentences** | | | | | | | | | |
| *Accuracy* | | | | | | | | | |
| Mamba retriever 1.3B | 88.0 | 76.5 | **63.8** | 59.0 | **73.0** | 79.1 | **48.5** | **90.7** | 59.2 |
| Mamba retriever 130M | 80.0 | **79.4** | 53.8 | **59.6** | 71.0 | 73.8 | 47.5 | 89.5 | **63.1** |
| BM25 | 79.0 | 63.2 | 43.8 | 41.6 | 64.0 | 73.3 | 1.5 | 80.2 | 33.8 |
| Contriever | 81.5 | 67.6 | 56.2 | 50.9 | 70.5 | 72.7 | 13.0 | 86.0 | 47.7 |
| Contriever-FT | 80.0 | 69.1 | 57.7 | 54.0 | **73.0** | 72.1 | 20.5 | 84.9 | 60.8 |
| GritLM | **89.5** | 72.1 | 59.2 | 54.0 | 71.0 | **82.6** | 30.0 | 87.2 | 55.4 |
| NV-Embed-v2-7B | 80.0 | 66.2 | 61.5 | 54.7 | 70.5 | 76.2 | 27.5 | 88.4 | 55.4 |
| Stella-1.5B | 78.0 | 73.5 | 60.0 | 50.9 | 71.5 | 72.1 | 29.0 | 88.4 | 45.4 |
| GTE-Qwen2-1.5B | 79.0 | 67.6 | 57.7 | 57.1 | 71.0 | 72.1 | 28.0 | 88.4 | 52.3 |
| *F1* | | | | | | | | | |
| Mamba retriever 1.3B | N/A | **41.6** | 24.5 | **30.1** | N/A | N/A | **4.0** | N/A | 27.1 |
| Mamba retriever 130M | N/A | 41.3 | 23.3 | 29.1 | N/A | N/A | 3.0 | N/A | **28.4** |
| BM25 | N/A | 37.3 | 22.1 | 23.1 | N/A | N/A | 0.5 | N/A | 20.4 |
| Contriever | N/A | 39.6 | **25.2** | 27.2 | N/A | N/A | 1.3 | N/A | 23.8 |
| Contriever-FT | N/A | 40.2 | 23.8 | 27.5 | N/A | N/A | 2.0 | N/A | 25.9 |
| GritLM | N/A | 40.0 | 24.2 | 29.4 | N/A | N/A | 2.9 | N/A | 25.4 |
| NV-Embed-v2-7B | N/A | 40.0 | 24.1 | 29.3 | N/A | N/A | 2.9 | N/A | 25.4 |
| Stella-1.5B | N/A | 40.4 | 24.2 | 27.0 | N/A | N/A | 2.7 | N/A | 23.9 |
| GTE-Qwen2-1.5B | N/A | 39.7 | 24.1 | 29.8 | N/A | N/A | 3.5 | N/A | 24.7 |
| **Retrieving 5 chunks** | | | | | | | | | |
| *Accuracy* | | | | | | | | | |
| BM25 | 79.5 | 60.3 | 37.7 | 52.2 | 70.0 | **73.8** | 4.5 | 86.0 | 58.5 |
| Contriever | 77.0 | 66.2 | 39.2 | 53.4 | 70.5 | 73.3 | 35.5 | 86.0 | 63.1 |
| Contriever-FT | 79.5 | 69.1 | **61.5** | 55.3 | 74.0 | 71.5 | 37.5 | 87.2 | 57.7 |
| GritLM | 79.5 | **77.9** | 51.5 | 55.3 | 74.0 | **73.8** | 36.5 | 84.9 | 61.5 |
| NV-Embed-v2-7B | 79.0 | 70.6 | 56.2 | 53.4 | **74.5** | 72.7 | 44.5 | **90.7** | **65.4** |
| Stella-1.5B | 79.0 | 69.1 | 47.7 | 53.4 | 73.0 | 73.3 | 41.0 | 86.0 | **65.4** |
| GTE-Qwen2-1.5B | **80.0** | 69.1 | 46.9 | **57.1** | 69.5 | **73.8** | **46.0** | 86.0 | 63.1 |
| *F1* | | | | | | | | | |
| BM25 | N/A | 40.5 | 21.7 | 26.4 | N/A | N/A | 1.1 | N/A | 26.5 |
| Contriever | N/A | 37.5 | 21.3 | 26.5 | N/A | N/A | 3.5 | N/A | 27.9 |
| Contriever-FT | N/A | 40.4 | 23.5 | 28.2 | N/A | N/A | 3.1 | N/A | 27.7 |
| GritLM | N/A | 41.7 | 23.0 | 27.2 | N/A | N/A | 3.0 | N/A | 27.8 |
| NV-Embed-v2-7B | N/A | 41.7 | **24.0** | 28.1 | N/A | N/A | 3.2 | N/A | 27.8 |
| Stella-1.5B | N/A | **42.1** | 22.4 | 26.7 | N/A | N/A | 3.4 | N/A | 28.3 |
| GTE-Qwen2-1.5B | N/A | 40.6 | 22.3 | **29.0** | N/A | N/A | **3.7** | N/A | **28.4** |
| **Full context** | | | | | | | | | |
| *Accuracy* | | | | | | | | | |
| GPT-4o Full Context | 84.0 | 67.6 | 63.1 | 62.7 | 83.5 | 73.8 | 51.5 | 94.2 | 73.8 |
| *F1* | | | | | | | | | |
| GPT-4o Full Context | N/A | 43.1 | 24.2 | 30.0 | N/A | N/A | 4.4 | N/A | 31.6 |

## A.5 LLAMA-3.1 AS GENERATOR

Table 15: QA accuracy across 41 datasets with Llama-3.1 as generator with retrieval of 50 sentences. When not paired with a retriever, Llama-3.1-8B and Llama-3.1-70B are provided with the full document in-context. Results continue to next page.

| Model | Average | narrativeqa | qasper_en | multifieldqa | hotpotqa | 2wikimqa | musique_choice_eng | longbook_eng | longdialogue_qa |
|---|---|---|---|---|---|---|---|---|---|
| | n = 5735 | n = 200 | n = 200 | n = 147 | n = 200 | n = 199 | n = 200 | n = 200 | n = 200 |
| **Llama-3.1 8B** | | | | | | | | | |
| Accuracy | | | | | | | | | |
| Mamba retriever 1.3B | **47.9** | 44.5 | 50.5 | **86.4** | 64.5 | 48.2 | 36.0 | **49.5** | 13.0 |
| Mamba retriever 130M | 47.4 | 40.0 | **51.5** | 85.7 | **68.0** | 49.7 | **42.0** | 49.0 | **19.0** |
| GritLM | 36.5 | **44.7** | 47.0 | 84.4 | 67.0 | **51.8** | 37.5 | 47.0 | 9.5 |
| BM25 | 39.1 | 31.5 | 46.0 | 77.6 | 59.5 | 36.2 | 26.0 | 39.5 | 9.0 |
| Contriever | 44.8 | 36.0 | 49.0 | 82.3 | 56.0 | 43.7 | 31.0 | **49.5** | 13.5 |
| Dragon | 44.1 | 33.5 | 48.5 | 82.3 | 57.0 | 40.7 | 28.5 | 47.5 | 16.0 |
| OpenAI v3-large | 46.0 | 38.5 | 48.5 | 85.0 | 56.5 | 47.2 | 27.5 | 48.0 | 18.5 |
| F1 | | | | | | | | | |
| Mamba retriever 1.3B | **29.2** | **20.5** | 35.1 | 48.6 | **39.9** | 26.2 | 25.6 | N/A | 12.1 |
| Mamba retriever 130M | **29.2** | 20.3 | 31.8 | **50.0** | 39.2 | 26.3 | 28.0 | N/A | 16.6 |
| GritLM | 28.3 | 18.3 | 33.8 | 48.7 | 39.0 | **29.0** | 29.5 | N/A | 8.8 |
| BM25 | 24.1 | 14.3 | 29.2 | 48.7 | 33.2 | 18.8 | 19.5 | N/A | 8.8 |
| Contriever | 27.0 | 17.4 | 32.8 | 49.1 | 33.0 | 24.1 | 20.0 | N/A | 11.9 |
| Dragon | 26.9 | 15.8 | 31.2 | 49.8 | 32.7 | 22.7 | 19.1 | N/A | 13.7 |
| OpenAI v3-large | 28.2 | 19.0 | 32.6 | 49.1 | 34.6 | 27.3 | 20.1 | N/A | **16.8** |
| **Llama-3.1 70B** | | | | | | | | | |
| Accuracy | | | | | | | | | |
| Mamba retriever 1.3B | **58.8** | **55.5** | 58.0 | 89.1 | **79.0** | 73.9 | **59.5** | 64.5 | 29.5 |
| Mamba retriever 130M | 57.5 | 52.0 | **60.0** | **90.5** | 76.5 | 67.8 | 57.5 | 60.5 | **40.0** |
| GritLM | 54.2 | 45.5 | 56.5 | 87.1 | 75.5 | 70.4 | 59.0 | **65.5** | 17.0 |
| BM25 | 46.9 | 40.0 | 50.0 | 86.4 | 72.5 | 60.8 | 48.5 | 51.0 | 5.5 |
| Contriever | 52.9 | 46.5 | 52.5 | 85.0 | 73.5 | 65.8 | 45.5 | 65.0 | 16.0 |
| Dragon | 51.9 | 44.0 | 54.0 | 85.7 | 71.0 | 60.8 | 46.0 | 58.0 | 16.0 |
| OpenAI v3-large | 55.1 | 47.5 | 54.5 | 86.4 | 74.5 | 65.3 | 50.0 | 64.0 | 22.0 |
| F1 | | | | | | | | | |
| Mamba retriever 1.3B | 30.0 | **18.0** | 32.9 | **49.7** | 24.2 | **21.9** | **18.4** | N/A | 20.7 |
| Mamba retriever 130M | **31.8** | 16.6 | 33.2 | 49.3 | 23.6 | 20.2 | 16.9 | N/A | **27.4** |
| GritLM | 28.0 | 16.3 | 31.3 | 49.2 | 22.8 | 20.9 | 16.9 | N/A | 12.5 |
| BM25 | 23.7 | 13.1 | 29.8 | 46.8 | 23.7 | 20.0 | 15.4 | N/A | 4.0 |
| Contriever | 26.8 | 15.8 | 30.6 | 48.3 | **24.7** | 21.3 | 13.7 | N/A | 12.2 |
| Dragon | 26.6 | 15.5 | 30.7 | 48.1 | 23.0 | 19.6 | 15.1 | N/A | 12.5 |
| OpenAI v3-large | 28.2 | 17.1 | **33.4** | 47.2 | 22.9 | 19.7 | 15.6 | N/A | 16.3 |
| **Full context** | | | | | | | | | |
| Accuracy | | | | | | | | | |
| Llama-3.1-8B Full Context | 49.1 | 51.0 | 47.5 | 81.6 | 78.0 | 78.4 | 51.0 | 47.0 | 10.0 |
| Llama-3.1 70B Full Context | 57.8 | **56.0** | **62.0** | **85.0** | **80.5** | **85.4** | **62.5** | **65.0** | **17.0** |
| F1 | | | | | | | | | |
| Llama-3.1-8B Full Context | 30.9 | **25.7** | **37.1** | 48.7 | **55.6** | **59.8** | **38.3** | N/A | 8.5 |
| Llama-3.1 70B Full Context | **31.3** | 20.1 | 34.0 | **48.9** | 30.4 | 39.9 | 29.2 | N/A | **12.6** |

Table 16: Results continued

| Model | longbook_qa_eng n = 200 | loogle_CR_mixup_16k n = 99 | loogle_CR_mixup_32k n = 99 | loogle_CR_mixup_64k n = 99 | loogle_CR_mixup_128k n = 99 | loogle_MIR_mixup_16k n = 139 | loogle_CR_mixup_256k n = 99 | loogle_MIR_mixup_32k n = 139 |
|---|---|---|---|---|---|---|---|---|
| | | | | **Llama-3.1 8B** | | | | |
| | | | | Accuracy | | | | |
| Mamba retriever 1.3B | 37.5 | 42.4 | **45.5** | 43.4 | 35.4 | **33.8** | 31.3 | 29.5 |
| Mamba retriever 130M | **39.0** | **44.4** | 40.4 | 36.4 | 32.3 | 30.9 | **35.4** | **34.5** |
| GritLM | 33.5 | 36.4 | 36.4 | 40.4 | 33.3 | 25.2 | 32.3 | 25.2 |
| BM25 | 19.5 | 29.3 | 35.4 | 29.3 | 31.3 | 15.1 | 25.3 | 8.6 |
| Contriever | 31.5 | 41.4 | 38.4 | 30.3 | **36.4** | 21.6 | 30.3 | 18.7 |
| Dragon | 34.5 | 38.4 | 37.4 | 29.3 | 32.3 | 22.3 | 31.3 | 21.6 |
| OpenAI v3-large | 34.5 | 36.4 | 35.4 | 38.4 | 35.4 | 20.1 | 32.3 | 25.2 |
| | | | | F1 | | | | |
| Mamba retriever 1.3B | 19.9 | 24.3 | 22.5 | 23.2 | **24.9** | **22.3** | 21.0 | **21.5** |
| Mamba retriever 130M | **20.4** | **24.6** | **23.6** | **24.5** | 23.3 | 20.9 | **22.1** | 21.1 |
| GritLM | 18.4 | 21.8 | 21.0 | 21.1 | 22.5 | 20.7 | 21.7 | 21.3 |
| BM25 | 10.8 | 19.9 | 19.7 | 20.2 | 20.7 | 13.8 | 18.4 | 13.7 |
| Contriever | 17.5 | 20.4 | 19.6 | 18.9 | 18.2 | 18.2 | 20.6 | 17.1 |
| Dragon | 19.6 | 22.4 | 19.4 | 17.6 | 17.6 | 17.1 | 21.2 | 17.8 |
| OpenAI v3-large | 18.0 | 22.4 | 20.9 | 19.8 | 21.2 | 16.8 | 20.2 | 17.9 |
| | | | | **Llama-3.1 70B** | | | | |
| | | | | Accuracy | | | | |
| Mamba retriever 1.3B | **56.5** | **54.5** | 50.5 | 49.5 | 44.4 | **46.8** | 38.4 | **41.0** |
| Mamba retriever 130M | 49.5 | 45.5 | 45.5 | 45.5 | **47.5** | 40.3 | 41.4 | **41.0** |
| GritLM | 40.5 | 44.4 | 43.4 | 40.4 | 42.4 | 34.5 | 42.4 | 36.7 |
| BM25 | 25.5 | 39.4 | 38.4 | 31.3 | 35.4 | 18.0 | 31.3 | 15.8 |
| Contriever | 40.5 | 46.5 | 44.4 | 44.4 | 40.4 | 26.6 | 41.4 | 26.6 |
| Dragon | 40.5 | 43.4 | 46.5 | 40.4 | 41.4 | 26.6 | 40.4 | 25.2 |
| OpenAI v3-large | 47.0 | 50.5 | 48.5 | 46.5 | 43.4 | 33.1 | **44.4** | 30.2 |
| | | | | F1 | | | | |
| Mamba retriever 1.3B | **19.5** | **22.5** | 21.0 | **22.2** | **21.4** | **22.9** | 21.6 | **22.3** |
| Mamba retriever 130M | 18.1 | 21.3 | 21.0 | 22.1 | 20.8 | 21.1 | 21.1 | 21.6 |
| GritLM | 16.3 | 20.1 | **21.1** | 21.8 | **21.4** | 21.0 | 18.6 | 21.3 |
| BM25 | 9.6 | 17.8 | 18.5 | 17.6 | 18.3 | 13.9 | 16.4 | 13.3 |
| Contriever | 15.2 | 20.4 | 20.0 | 19.1 | 18.2 | 16.7 | 18.2 | 16.9 |
| Dragon | 14.8 | 20.1 | 20.4 | 18.1 | 18.9 | 16.9 | 18.1 | 17.0 |
| OpenAI v3-large | 17.5 | 22.2 | 21.0 | 20.3 | 20.1 | 18.5 | **21.9** | 17.4 |
| | | | | **Full context** | | | | |
| | | | | Accuracy | | | | |
| Llama-3.1-8B Full Context | 42.0 | 42.4 | 40.4 | 27.3 | 27.3 | 31.7 | 27.3 | 32.4 |
| Llama-3.1 70B Full Context | 52.5 | **60.6** | **57.6** | **42.4** | **37.4** | **43.9** | **31.3** | **38.1** |
| | | | | F1 | | | | |
| Llama-3.1-8B Full Context | **29.2** | **25.3** | **24.8** | 22.1 | 13.5 | **26.2** | 17.6 | **23.8** |
| Llama-3.1 70B Full Context | 27.5 | 23.1 | 22.1 | **22.2** | **21.6** | 21.7 | **20.3** | 22.2 |

Table 17: Results continued

| Model | loogle_MIR _mixup_64k n = 139 | loogle_MIR _mixup_128k n = 139 | loogle_MIR _mixup_256k n = 139 | multifieldqa_en _mixup_16k n = 101 | multifieldqa_en _mixup_32k n = 101 | multifieldqa_en _mixup_64k n = 101 | multifieldqa_en _mixup_128k n = 101 |
|---|---|---|---|---|---|---|---|
| | | | **Llama-3.1 8B** | | | | |
| | | | Accuracy | | | | |
| Mamba retriever 1.3B | **32.4** | **30.9** | 30.9 | 51.5 | 48.5 | 51.5 | 45.5 |
| Mamba retriever 130M | 26.6 | 30.2 | **31.7** | 50.5 | 52.5 | 48.5 | 52.5 |
| GritLM | 24.5 | 27.3 | 23.0 | 49.5 | 47.5 | 49.5 | 44.6 |
| BM25 | 11.5 | 7.2 | 10.1 | 45.5 | 43.6 | 46.5 | 40.6 |
| Contriever | 18.7 | 21.6 | 23.0 | 48.5 | **57.4** | **57.4** | 55.4 |
| Dragon | 20.9 | 22.3 | 20.9 | **57.4** | 56.4 | 49.5 | 50.5 |
| OpenAI v3-large | 19.4 | 28.1 | 27.3 | 53.5 | 52.5 | 49.5 | **57.4** |
| | | | F1 | | | | |
| Mamba retriever 1.3B | **20.6** | **22.1** | 18.6 | 33.1 | 34.4 | 33.0 | 30.8 |
| Mamba retriever 130M | 19.7 | 19.8 | **21.2** | 31.8 | **36.1** | 32.9 | **35.5** |
| GritLM | 19.9 | 21.1 | 20.4 | 31.5 | 32.9 | 30.5 | 30.4 |
| BM25 | 13.0 | 10.2 | 11.1 | 31.9 | 29.4 | 28.5 | 29.5 |
| Contriever | 16.4 | 17.7 | 17.7 | 33.2 | 34.7 | 33.1 | 33.1 |
| Dragon | 18.1 | 16.9 | 18.1 | **34.2** | 33.9 | 33.8 | 33.9 |
| OpenAI v3-large | 18.4 | 18.7 | 18.2 | 32.5 | 35.4 | **36.9** | 34.3 |
| | | | **Llama-3.1 70B** | | | | |
| | | | Accuracy | | | | |
| Mamba retriever 1.3B | **41.0** | **41.7** | 39.6 | 63.4 | 63.4 | 59.4 | 62.4 |
| Mamba retriever 130M | 37.4 | 33.1 | 36.0 | **64.4** | 63.4 | **62.4** | **63.4** |
| GritLM | 35.3 | 35.3 | 36.0 | 57.4 | 56.4 | 54.5 | 54.5 |
| BM25 | 15.1 | 17.3 | 13.7 | 54.5 | 56.4 | 56.4 | 56.4 |
| Contriever | 26.6 | 25.9 | 22.3 | 58.4 | 61.4 | 60.4 | **63.4** |
| Dragon | 25.2 | 18.0 | 20.1 | 57.4 | **64.4** | **62.4** | 61.4 |
| OpenAI v3-large | 30.2 | 30.9 | 31.7 | 59.4 | 61.4 | 57.4 | 62.4 |
| | | | F1 | | | | |
| Mamba retriever 1.3B | **22.2** | **22.0** | **22.6** | 31.4 | 32.5 | 31.8 | 30.9 |
| Mamba retriever 130M | 20.5 | 19.7 | 20.4 | 31.1 | 31.2 | 31.9 | 31.1 |
| GritLM | 20.9 | 20.3 | 21.3 | 31.0 | 31.3 | 29.9 | 30.2 |
| BM25 | 12.6 | 12.9 | 11.0 | 28.9 | 30.1 | 28.7 | 28.3 |
| Contriever | 17.8 | 16.5 | 15.6 | 29.6 | 32.4 | 32.1 | 31.2 |
| Dragon | 17.7 | 16.3 | 16.0 | 31.2 | 31.9 | **33.0** | 31.7 |
| OpenAI v3-large | 18.8 | 17.2 | 19.1 | **32.7** | **33.4** | 32.1 | **31.8** |
| | | | **Full context** | | | | |
| | | | Accuracy | | | | |
| Llama-3.1 8B Full Context | 18.0 | 19.4 | 12.2 | 49.5 | 39.6 | 33.7 | 26.7 |
| Llama-3.1 70B Full Context | **28.8** | **23.0** | **20.9** | **58.4** | **43.6** | **39.6** | **33.7** |
| | | | F1 | | | | |
| Llama-3.1 8B Full Context | 16.8 | 13.1 | 7.1 | **37.7** | **29.1** | 25.1 | 21.8 |
| Llama-3.1 70B Full Context | **21.7** | **17.6** | **13.4** | 30.0 | 28.2 | **26.4** | **25.1** |

Table 18: Results continued

| Model | altqa _4k n = 199 | altqa _16k n = 199 | meetingpred _4k n = 100 | multifieldqa_en _mixup_256k n = 101 | meetingpred _16k n = 100 | meetingqa _4k n = 86 | meetingqa _16k n = 91 | paperqa _4k n = 82 | paperqa _16k n = 90 |
|---|---|---|---|---|---|---|---|---|---|
| | | | | **Llama-3.1 8B** | | | | | |
| | | | | Accuracy | | | | | |
| Mamba retriever 1.3B | **78.9** | 72.9 | **37.0** | 50.5 | **21.0** | **67.4** | 61.5 | **76.8** | 64.4 |
| Mamba retriever 130M | **78.9** | 72.9 | 29.0 | 45.5 | 18.0 | 65.1 | 60.4 | 73.2 | 66.7 |
| GritLM | 75.9 | **74.4** | 32.0 | 43.6 | 20.0 | 64.0 | 62.6 | 74.4 | **70.0** |
| BM25 | 62.8 | 55.3 | 27.0 | 49.5 | 18.0 | 66.3 | 64.8 | 74.4 | 68.9 |
| Contriever | 72.9 | 72.9 | 35.0 | 57.4 | **21.0** | 62.8 | 65.9 | 74.4 | **70.0** |
| Dragon | 69.8 | 66.8 | 35.0 | 56.4 | 16.0 | 60.5 | **69.2** | 69.5 | 62.2 |
| OpenAI v3-large | 75.9 | 72.4 | 36.0 | **55.4** | **21.0** | 64.0 | 64.8 | 74.4 | 65.6 |
| | | | | F1 | | | | | |
| Mamba retriever 1.3B | **78.5** | 71.4 | 30.8 | **33.8** | 17.8 | N/A | N/A | N/A | N/A |
| Mamba retriever 130M | 78.4 | 72.1 | 24.0 | 32.1 | 15.2 | N/A | N/A | N/A | N/A |
| GritLM | 75.6 | **73.4** | 26.4 | 31.3 | 16.6 | N/A | N/A | N/A | N/A |
| BM25 | 62.3 | 54.8 | 23.7 | 31.4 | 18.2 | N/A | N/A | N/A | N/A |
| Contriever | 72.4 | 72.6 | 29.5 | 33.1 | **18.8** | N/A | N/A | N/A | N/A |
| Dragon | 69.9 | 65.3 | **32.7** | 32.7 | 14.1 | N/A | N/A | N/A | N/A |
| OpenAI v3-large | 75.4 | 71.9 | 30.3 | 33.5 | 18.6 | N/A | N/A | N/A | N/A |
| | | | | **Llama-3.1 70B** | | | | | |
| | | | | Accuracy | | | | | |
| Mamba retriever 1.3B | 79.9 | 74.9 | **58.0** | 59.4 | **36.0** | 74.4 | 67.0 | 72.0 | **80.0** |
| Mamba retriever 130M | 79.9 | 73.4 | 47.0 | 60.4 | 31.0 | **76.7** | 67.0 | 74.4 | **80.0** |
| GritLM | 77.4 | 74.9 | 50.0 | 55.4 | 30.0 | 73.3 | 61.5 | 76.8 | 78.9 |
| BM25 | 68.8 | 60.3 | 42.0 | 59.4 | 16.0 | 74.4 | 64.8 | **78.0** | **80.0** |
| Contriever | 79.4 | 73.9 | 53.0 | 57.4 | 26.0 | 73.3 | **71.4** | 74.4 | **80.0** |
| Dragon | 73.9 | 70.9 | 51.0 | 58.4 | 20.0 | 75.6 | 65.9 | **78.0** | 77.8 |
| OpenAI v3-large | **81.4** | **76.4** | 45.0 | **61.4** | 34.0 | 74.4 | 67.0 | 74.4 | 75.6 |
| | | | | F1 | | | | | |
| Mamba retriever 1.3B | 79.0 | 74.9 | **59.2** | 31.1 | **35.4** | N/A | N/A | N/A | N/A |
| Mamba retriever 130M | 78.5 | 72.1 | 46.8 | 31.6 | 29.8 | N/A | N/A | N/A | N/A |
| GritLM | 76.0 | 73.9 | 48.4 | 30.6 | 28.8 | N/A | N/A | N/A | N/A |
| BM25 | 68.4 | 58.9 | 38.9 | 29.7 | 16.5 | N/A | N/A | N/A | N/A |
| Contriever | 78.5 | 73.4 | 50.8 | 30.2 | 23.9 | N/A | N/A | N/A | N/A |
| Dragon | 72.9 | 70.4 | 50.7 | 31.8 | 19.1 | N/A | N/A | N/A | N/A |
| OpenAI v3-large | **80.5** | **76.4** | 42.9 | **32.8** | 31.3 | N/A | N/A | N/A | N/A |
| | | | | **Full context** | | | | | |
| | | | | Accuracy | | | | | |
| Llama-3.1 8B Full Context | **80.4** | 77.9 | 42.0 | 32.7 | 35.0 | **82.6** | **78.0** | 81.7 | 76.7 |
| Llama-3.1 70B Full Context | 79.4 | **79.9** | **68.0** | **34.7** | **58.0** | 82.6 | 74.7 | **86.6** | **87.8** |
| | | | | F1 | | | | | |
| Llama-3.1-8B Full Context | 78.2 | 77.9 | 35.8 | 22.3 | 30.4 | N/A | N/A | N/A | N/A |
| Llama-3.1 70B Full Context | **79.4** | **79.9** | **71.3** | **23.1** | **57.2** | N/A | N/A | N/A | N/A |

Table 19: Results continued

| Model | tpo | financial _qa | legal_contract _qa | scientific _qa | quality | coursera | docfinQA | muld _CAC | ELITR _Bench |
|---|---|---|---|---|---|---|---|---|---|
| | n = 200 | n = 68 | n = 130 | n = 161 | n = 200 | n = 172 | n = 200 | n = 86 | n = 130 |
| | | | | **Llama-3.1 8B** | | | | | |
| | | | | Accuracy | | | | | |
| Mamba retriever 1.3B | 59.0 | **54.4** | **36.2** | 54.7 | 37.5 | 50.0 | **23.0** | **83.7** | 51.5 |
| Mamba retriever 130M | 59.5 | 45.6 | 34.6 | **57.8** | 36.0 | **52.3** | **23.0** | 80.2 | 53.8 |
| GritLM | 61.0 | 38.2 | 32.3 | 53.4 | 41.5 | 48.8 | 13.5 | 76.7 | 40.0 |
| BM25 | 60.5 | 47.1 | 23.1 | 47.8 | 37.5 | 47.7 | 0.5 | 70.9 | 56.9 |
| Contriever | **62.0** | 47.1 | 26.9 | 42.2 | **42.0** | 50.6 | 14.0 | 79.1 | 49.2 |
| Dragon | 61.5 | 50.0 | 27.7 | 46.0 | 40.5 | 43.0 | 13.5 | 82.6 | 57.7 |
| OpenAI v3-large | **62.0** | 51.5 | 35.4 | 50.9 | 39.5 | 47.7 | 22.0 | 74.4 | **59.2** |
| | | | | F1 | | | | | |
| Mamba retriever 1.3B | N/A | 42.6 | 23.0 | **30.4** | N/A | N/A | 3.3 | N/A | 23.7 |
| Mamba retriever 130M | N/A | 41.0 | **24.2** | 29.1 | N/A | N/A | **3.7** | N/A | 25.0 |
| GritLM | N/A | 41.7 | 22.8 | 30.1 | N/A | N/A | 1.5 | N/A | 23.2 |
| BM25 | N/A | 39.9 | 22.2 | 27.1 | N/A | N/A | 0.3 | N/A | **27.4** |
| Contriever | N/A | 39.0 | 20.1 | 27.2 | N/A | N/A | 2.0 | N/A | 25.2 |
| Dragon | N/A | 42.2 | 20.6 | 27.8 | N/A | N/A | 2.1 | N/A | 25.7 |
| OpenAI v3-large | N/A | **42.7** | 22.3 | 28.4 | N/A | N/A | 2.9 | N/A | 26.3 |
| | | | | **Llama-3.1 70B** | | | | | |
| | | | | Accuracy | | | | | |
| Mamba retriever 1.3B | 75.0 | 57.4 | 44.6 | **61.5** | 54.0 | 63.4 | **44.0** | **86.0** | 63.1 |
| Mamba retriever 130M | **76.0** | 58.8 | 45.4 | 60.2 | 54.0 | **65.1** | 40.5 | 84.9 | 63.1 |
| GritLM | 75.0 | 50.0 | 42.3 | 58.4 | **59.0** | 62.2 | 26.0 | 84.9 | 55.4 |
| BM25 | 74.5 | 48.5 | 27.7 | 53.4 | 53.5 | 63.4 | 3.0 | 75.6 | 58.5 |
| Contriever | 74.0 | 50.0 | 32.3 | 50.9 | 56.0 | 64.5 | 30.0 | 84.9 | 58.5 |
| Dragon | 75.0 | 51.5 | 39.2 | 54.0 | 54.5 | 61.0 | 26.5 | 84.9 | 62.3 |
| OpenAI v3-large | 74.5 | 51.5 | 42.3 | 54.0 | 54.0 | 64.0 | 35.0 | 84.9 | **66.9** |
| | | | | F1 | | | | | |
| Mamba retriever 1.3B | N/A | 43.5 | 24.3 | 29.6 | N/A | N/A | **2.9** | N/A | 26.2 |
| Mamba retriever 130M | N/A | **44.0** | **24.7** | **30.1** | N/A | N/A | 2.8 | N/A | **28.1** |
| GritLM | N/A | 42.2 | 23.6 | 29.1 | N/A | N/A | 2.0 | N/A | 24.7 |
| BM25 | N/A | 41.7 | 22.7 | 26.9 | N/A | N/A | 0.5 | N/A | 24.8 |
| Contriever | N/A | 39.4 | 21.5 | 27.0 | N/A | N/A | 2.2 | N/A | 24.8 |
| Dragon | N/A | 41.8 | 23.5 | 27.0 | N/A | N/A | 2.0 | N/A | 25.8 |
| OpenAI v3-large | N/A | 41.0 | 23.5 | 28.3 | N/A | N/A | 2.4 | N/A | 26.4 |
| | | | | **Full context** | | | | | |
| | | | | Accuracy | | | | | |
| Llama-3.1 8B Full Context | 77.0 | 67.6 | 24.6 | 56.5 | 58.0 | 65.1 | **16.5** | 84.9 | 60.8 |
| Llama-3.1 70B Full Context | 82.0 | **69.1** | **38.5** | 60.2 | 77.5 | 77.3 | 14.0 | **91.9** | 69.2 |
| | | | | F1 | | | | | |
| Llama-3.1 8B Full Context | N/A | 42.7 | 18.0 | **37.6** | N/A | N/A | **6.4** | N/A | **31.7** |
| Llama-3.1 70B Full Context | N/A | **45.0** | **25.1** | 30.9 | N/A | N/A | 1.8 | N/A | 30.3 |

# B SYNTHETIC DATA GENERATION

## B.1 CHUNK-BASED GENERATION

See Figure 5 for chunk-based generation prompt.

---

Given an indexed list of sentences, generate a question and answer pair from the sentences following these rules:
1. The question must be concrete, i.e. it should question a specific content in the given sentences.
2. The question and answer pair must depend on multiple sentences that spread throughout the chunk.
3. The answer must be coherent and concise.

Indexed list of sentence: {**indexed_list_of_sentences**}

First generate the question and answer. Output the question after the keyword **QUESTION:**. Output the answer after the keyword **ANSWER:**. After generating question and answer, output the indices of the sentences on which the question and answer pair depends on. Output a list of indices [index1, index2, ...] after the keyword **SENTENCES:**.

---

Figure 5: Prompt for chunk-based generation.

## B.2 PAIR-BASED GENERATION

See Figure 6 for a pair-based generation prompt.

---

Given only two pieces of information extracted from a document, invent a circumstance where there is a logical and reasonable connection between these two sentences. From this circumstance, create a question and its answer pair, such that to answer the question, one would need both pieces of information. Make sure it is IMPOSSIBLE to answer the question without knowing BOTH information. Importantly, the question must depend on this two information in a profound, non-superficial, non-apparent and NO KEYWORD OVERLAPPING way. Invent the circumstance and the connection between these two pieces of information first, output the connection after the keyword "CONNECTION:" Based on the connection, briefly explain step by step as to why it is impossible to answer the question using only one piece of information. Output all explanation and reasoning after the keyword "REASON:" Based on your reasoning and explanation, while making sure the question itself does not use ANY keyword directly from these two information, output the question after the keyword "QUESTION:" Output the answer after the keyword "ANSWER:"

Information 1: {**Chunk_1**}

Information 2: {**Chunk_2**}

---

Figure 6: Prompt for pair-based generation.

## B.3 LINK-BASED GENERATION

See Figure 7 for the prompt to discover natural connections within a document.

Given a chunk of sentences
Chunk:
{chunk}
What other sentences in the document are highly connected to this chunk? Output each
sentence index that is highly connected to the chunk, and explain the reason.
Document:
{document}

Figure 7: Prompt to discover natural connections within a document.

See Figure 8 for the synthetic question generation prompt.

Two chunks of text  in a document are connected with the following connection. Use this
connection to build a question. Step by step explain how you would take advantage of this
connection, and build a short, concise, one-sentence, concrete, non-conceptual, non-ambiguous
question. IMPORTANTLY, you must use exact words from the connection given to you, but you
must never refer to the chunks, never mention words such as "connection", "alignment",
"relationship" between chunks. The question must be self-contained, and cannot not refer to the
chunks, and must standalone makes sense.

Output your reasoning, especially how you would take advantage of this connection, and your
verification, especially why the question is non-conceptual, and concrete, and self-contained and
standalone makes sense, after the keyword "REASON:"
Based on your step-by-step reasoning and verification, then output the question after the
keyword "QUESTION:"

Connection:
{connection}

Figure 8: Prompt for synthetic question generation based on natural connections.

See Figure 9 for labeling sentences as relevant or irrelevant prompt.

Given a question, and a list of indexed text elements. Select the indices of all relevant text
element(s) that would be helpful to answer the question.

Question:
{question}

List of text elements:
{list_of_text_elements}

Recall, your task is to select indices for all relevant text elements that can help you answer the
question. Provide a step-by-step explanation after the keyword 'REASON:'
Based on your explanation, output a list of indices for text elements that are relevant and helpful
to answer the question, in this format, [index1, index2, ...] ,after the keyword "LIST:"
If no text element is helpful and relevant to answer the question, output an empty list [] after the
keyword "LIST:"

Figure 9: Prompt for labeling sentences as relevant or irrelevant to a synthetic query.

## B.4 SYNTHETIC DATA QUALITY EVALUATION

We provide and evaluate a few synthetic data examples generated from different synthetic data strategies in Tables 20, 21, and 22.

**Link-based** data (20) typically generates questions that are more coherent because these questions arise from natural connections searched within the document. The labeled sentences are implicitly linked to the question through these connections, with sentences from different parts of the document collectively forming the answer. When a model is trained on such data, identifying the first chunk can guide it to locate the second chunk. This training process teaches the model to use information from previously encountered content when evaluating the relevance of each new sentence. By training Mamba retriever in this way, it learns to use its global understanding of the entire document to determine which sentences are important for answering the question.

**Human Evaluation:** The first example in Table 20 explores the significance of the little things in a marriage. In both contexts, the highlighted sentences offer intriguing insights into this topic. In the first context, a young girl asks her mother about things that might not matter in a marriage. The mother responds by emphasizing that these small details, which can take the edge off, do indeed matter. In the second context, the sentence about a husband and wife working together highlights the importance of collaboration in a marriage, i.e. *pulling together*. Thus, both contexts provide valuable information to address the question. Without the first context, the *"simple secret"* would not resonate as strongly, as it refers to the seemingly trivial yet significant details discussed by Marie and her mother. Interestingly, the female character in the second context is also the same Marie. Therefore, the first context sets the stage for the Mamba retriever to identify the second context. This ability to utilize long-range connections is crucial for a deeper understanding of subsequent contexts.

The second example in Table 20 examines the tension between Lester's internal conflict with his father and his struggle to navigate societal rejection. Both contexts illuminate distinct yet interconnected aspects of this conflict. In the first context, Lester grapples with his father's disapproval and his own hesitancy to act decisively to mend their relationship. His introspection reveals his uncertainty about standing alone in the face of societal judgment. The second context depicts the external consequences of Lester's actions. Together, the two contexts demonstrate the layered nature of Lester's struggle, where his need for personal reform and decisive action is tied to both his father's approval and his standing in society. By establishing Lester's introspective conflict in the first context, Mamba retriever learns to recognize and leverage this psychological groundwork when identifying relevant connections in the second context.

For chunk-based data (21), GPT-4o-mini processes each text chunk in its entirety and directly generates questions based on the information within that chunk. This approach ensures that the generated questions are highly relevant to the content, as the model can focus on the specific details present in each text segment. However, this method may lead to issues with superficial textual overlap when training the Mamba retriever. The retriever might learn to search for semantically similar sentences rather than identifying deeper connections between individual sentences and the given query.

Pair-based synthetic data (22) are generated from two chunks of a long document that have high cosine similarity. High cosine similarity indicates significant textual overlap between the chunks but does not ensure logical or contextual dependencies, as demonstrated in Table 22. Consequently, the questions generated may appear unnatural. Additionally, creating questions directly from these chunks can result in questions that either consist of two merged smaller questions or are unrelated to both chunks.

The first example in 22 involves a question about an event that prompted an inquiry regarding a specific time during the group's evening activities. Context 1 effectively answers this question by discussing these activities. However, Context 2, which frequently uses keywords like "evening" and "I," creates a high semantic similarity with Context 1. Despite this similarity, Context 2 does not talk about the same event as Context 1 and does not contribute to answering the question in any sense. Similarly, the second example inquires about Thomas's motivation for confessing. Context 1 clearly explains that Thomas confessed because he felt sorry for Mary. In contrast, Context 2 is unrelated to the question; it only contains negative words that might have some semantic similarity to the question.

Table 20: Linked-based Synthetic Data Examples

| | **Question/Context; Important Sentences Highlighted** | **Connection** |
|---|---|---|
| Example 1 | *Question:* What are some little things that matter in a marriage?

*Context 1:* "I shan't own anything of the kind till you've been married three months, and he's had some bad dinners, and late breakfasts, and has got a bit sick of the butcher's bill. Then we'll see.""Little things like these can't matter between people who really love each other. You don't understand." "It's just these little things that take the edge off. "Marie's mother looked in and smiled to see her girl fingering her pretty things. "Aren't you two nearly ready to leave the inspection and come to tea?"

*Context 2:* they had made their beds and made them wrong; the great thing, the simple secret, was to make them right.A husband and wife must pull together, in everything. Pulling together would be sheer joy."Osborn," she said, "how well we understand each other, don't we?""I should think we do," whispered the young man. "Few married people seem really happy.""They must manage life badly, mustn't they?" "I remember mother and father; mother likes the idea of my getting married, but they used often to be nagging about something. | This sentence highlights the connection and understanding between partners in a marriage, which resonates with the chunk's exploration of love and the little things that matter in a relationship. |
| Example 2 | *Question:* How does Lester's internal conflict regarding his relationship with his father influence his need for decisive action in the face of social rejection and the need for reform?

*Context 1:* It was a long time before he stirred.And still, in the bottom of his heart, his erring son continued to appeal to him.CHAPTER XL Lester returned to Chicago. He realized that he had offended his father seriously, how seriously he could not say.In all his personal relations with old Archibald he had never seen him so worked up. But even now Lester did not feel that the breach was irreparable; he hardly realized that it was necessary for him to act decisively if he hoped to retain his father's affection and confidence. As for the world at large, what did it matter how much people talked or what they said. He was big enough to stand alone.But was he?People turn so quickly from weakness or the shadow of it.

*Context 2:* or at least the more conservative part of it would not.There were a few bachelors, a few gay married men, some sophisticated women, single and married, who saw through it all and liked him just the same, but they did not make society.He was virtually an outcast, and nothing could save him but to reform his ways; in other words, he must give up Jennie once and for all.But he did not want to do this. The thought was painful to him–objectionable in every way.Jennie was growing in mental acumen. She was beginning to see things quite as clearly as he did.She was not a cheap, | This sentence highlights Lester's internal conflict regarding his relationship with his father and the need for decisive action, which connects to the chunk's theme of social rejection and the need for reform. |

Table 21: Chunk-based Synthetic Data Examples

| | **Question/Context; Important Sentences Highlighted** |
|---|---|
| Example 1 | *Question:* What happens to previously granted Incentive Awards after the termination of the Plan?

*Context:*
6.1 EFFECTIVE DATE AND GRANT PERIOD

This Plan shall be effective as of the date of Board approval, March 24, 1998. Unless sooner terminated by the Board, the Plan shall terminate on March 24, 2008, unless extended. After the termination of the Plan, no Incentive Awards may be granted under the Plan, but previously granted awards shall remain outstanding in accordance with their applicable terms and conditions. |
| Example 2 | *Question:* What does Mr. Pennimore emphasize about the purpose of Gerald's time at the school?

*Context:* Dan nodded."You'd better believe he does! If he says you can't play baseball or football you can't, and that's all there is to it. But he's square, all right, is 'Muscles,' and you want to do just as he tells you. He's a wonder!" Gerald considered this in silence a moment. Then: "If a fellow can't play baseball and things I don't see any use of coming here," he murmured. Mr.Pennimore laughed. "So that's your idea, is it, son? Well, let me tell you that you're here to fit yourself for college. You wanted to come here, Gerald, and you've had your way. Now there must be no backing down, my boy. Life isn't all play, as you'll find out when you get older, but you can make it seem like play by taking an interest in work.You mustn't think that because I've got money enough for us both that you're going to sit down and twiddle your thumbs and watch the procession go by. No, sir!You're going to march with the rest, and I want to see you marching at the head. |

Table 22: Pair-based Synthetic Data Examples

| | **Question/Context; Important Sentences Highlighted** |
|---|---|
| Example 1 | *Question:* What significant event occurred that prompted a query about a specific time during the group's evening activities?

*Context 1:*
I walked to the house of a banker who entertained me.Naturally, my evening thoughts reverted to my home, and after reading a few verses in my Testament, ==I walked about the room until nearly eleven, thinking of my wife, and breathing the prayer, 'God bless you.' "I might not have recalled all the circumstances, save for the letter I received by the next post from her, with the query put in: 'Tell me what you were doing within a few minutes of eleven o'clock on Friday evening?I will tell you in my next why I ask; for something happened to me.'==In the middle of the week the letter came, and these words in it:–'I had just awoke from a slight repose, when I saw you in your night-dress bend over me, and utter the words, "God bless you!"I seemed also to feel your breath as you kissed me.'

*Context 2:*
I was deputed along with a medical officer to proceed to the nearest railway station at that time Allahabad, in charge of a sick officer.I will call myself Brown, the medical officer Jones, and the sick officer Robertson.We had to travel very slowly, Robertson being carried by coolies in a doolie, and on this account we had to halt at a rest-house, or pitch our camp every evening.One evening, when three marches out of Banda, I had just come into Robertson's room about midnight to relieve Jones, for Robertson was so ill that we took it by turns to watch him, when Jones took me aside and whispered that he was afraid our friend was dying, that he did not expect him to live through the night, and though I urged him to go and lie down, and that I would call him on any change taking place, he would not leave.We both sat down and watched. |
| Example 2 | *Question:* What motivated Thomas to seek forgiveness and confess his past actions?

*Context 1:*
"Oh, no!He's a gen—" but was drowned in laughter.He threw his head up and laughed to the sky."You're a wonder, I must say.I beg him ten thousand pardons—I forgot.Of course, he's a gentleman."
Mary was piqued."That's not very kind of you," she said, with reproach in her tones, and he humbled himself at once.==''I'm very sorry, but I'll confess the whole.The fact is, you've jumped into a little pit which I had dug for you—headlong.==Upon my word, I beg your pardon.But don't you know that these class-boxes into which you plump every mother's son of us, and are at such pains to keep guarded, lest one of us should step out, are the very things I'm vowed to destroy?

*Context 2:*
Only, when desire fades in us, o' God's name let us die.Our friend here cried in his heart that his had never bloomed before.Spell-bound to a beautiful vision, he walked enraptured in the light of it, travelling up the path of its beam, sighing, not that it should be so long, but that his steps should lag so short of his urgency.And to the lips of his heart—as it were—recurred and recurred the dear, familiar phrases, true once and true now to who so love.The well-found hearth, and One beside it: surely, happily there!Denied him for so long; now in full sight!The buffeting, windy world outside, the good door barred, the ruddy fire, the welcoming arms, the low glad voice! |

# C    TEST SET EVALUATION

## C.1    FREEFORM QUESTION-ANSWER JUDGING PROMPT

See Figure 10 for an freeform question-answer judging prompt.

---

Given a question, a groundtruth answer, and an attempted answer, use the following criteria to determine if the attempted answer accurately reflects the groundtruth answer.
Criteria:
- The majority of the information in the attempted answer should overlap with the groundtruth answer. Note that the attempted answer may include additional information derived from the question.
- The attempted answer may extend the groundtruth answer while covering all its aspects, however, the attempted answer should not be contradicting the groundtruth answer.
- If the groundtruth contains numbers, the attempted answer must match when rounded to the same precision as the groundtruth.
Example 1:
Groundtruth Answer: 1983
Attempted Answer: 1.983 million
Reason: The groundtruth answer 1983 is a whole number without any units. The attempted answer uses 1.983, which is different from the whole number 1983 and thus should be considered incorrect.
Decision: NO
Example 2:
Groundtruth Answer: 93
Attempted Answer: 93 million
Reason: The attempted answer is 93 million, which uses the same digits as the attempted answer and thus should be considered correct.
Decision: YES
Question:
**{question}**
Groundtruth Answer:
**{gt_answer}**
Attempted Answer:
**{answer}**
Think step by step when you compare these two answers. Based on the reasoning, output a YES/NO decision after the keyword "DECISION:".

---

Figure 10: Prompt for judging the correctness of an answer to an freeform question.

## C.2    MULTIPLE CHOICE QUESTION QUESTION-ANSWER JUDGING PROMPT

See Figure 11 for multiple choice question-answer judging prompt.

---

Given a multiple-choice question, a ground truth answer, and an attempted answer, the attempted answer should be the same option as the ground truth answer. It should not include any other options beyond those in the ground truth answers. Some parts of the attempted answer may overlap with information from the question.

Question:
**{question}**
Ground Truth Answer:
**{gt_answer}**
Attempted Answer:
**{answer}**
Think step by step when you compare these two answers. Based on the reasoning, output a YES/NO decision after the keyword "DECISION:".

---

Figure 11: Prompt for judging the correctness of an answer to a multiple choice question.

## D   TRAINING HYPERPARAMETER SETTING

Our training process used a peak learning rate of $1 \times 10^{-4}$, optimized on the validation set, and a minimum learning rate of $1 \times 10^{-5}$. We used an effective batch size of $64$ by setting the gradient accumulation steps to $8$ and applied a maximum gradient norm of $1$. Optimization was performed using the AdamW optimizer ($Loshchilov \& Hutter$, 2019) with $\beta = (0.9, 0.95)$ and a weight decay of $0.01$. A cosine learning rate scheduler with a $10\%$ warmup phase was employed. Additionally, mixed-precision training with BF16 was utilized to enhance computational efficiency and reduce exploding gradient issue.

# E  FURTHER ANALYSES

## E.1  ABLATION FOR RELATIVE POSITION OF LINKED-CHUNKS

We investigate whether the relative positions of chunks (i.e., labeled sentences) impact the training and performance of Mamba models. From the 1 million link-based synthetic data points, we select instances where the relative positions of both chunks (with respect to the full document) fall within the first 33%, between 33% and 67%, and after 67%. For each group, we randomly select 100k data points to train the Mamba retriever 130M model. From Table 23, we observe an incremental pattern in Mamba retriever's performance: it is worst when the chunks are located in the first third of the document, improves when the chunks are situated between the first and second thirds, and is best when the chunks are positioned after the second third. This pattern may be due to the increasing distance from the query (at the beginning of the document); the further apart the labeled sentences are from the query, the more challenging the training data becomes, leading to better performance for Mamba.

Table 23: Ablation study for the relative positions of the two linked chunks in a document.

| | Document Type | | | | |
|---|---|---|---|---|---|
| Synthetic Data Strategy | educational | creative | official | conversational | Average |
| Mamba-2-130M trained on 100k data | n = 1967 | n = 1733 | n = 1328 | n = 707 | Accuracy |
| **Linked-Chunks' Relative Positions** | | | | | |
| • Both in 0-33% of the document | 55.8 | 27.5 | 38.3 | 37.5 | 39.5 |
| • Both in 33-67% of the document | 56.5 | 30.6 | 41.8 | 38.9 | 42.0 |
| • Both in 67-100% of the document | 63.0 | 41.5 | 49.8 | 45.1 | 50.9 |

## E.2  ABLATION FOR TRAINING DOCUMENT LENGTH

Table 24: Ablation study for the training document sequence length.

| Mamba-2-130M trained on 600m tokens | | | Document Type | | | | |
|---|---|---|---|---|---|---|---|
| Input Sequence Length | | | educational | creative | official | conversational | Average |
| 2k tokens | 5k tokens | 10k tokens | n = 1967 | n = 1733 | n = 1328 | n = 707 | Accuracy |
| 300k data | 0 | 0 | 62.2 | 34.9 | 50.0 | 41.3 | 47.2 |
| 86k data | 86k data | 0 | 64.9 | 41.4 | 52.6 | 43.4 | 51.6 |
| 35k data | 35k data | 35k data | 66.9 | 49.3 | 57.8 | 47.1 | 56.4 |

We study whether the training document length has an impact on the performance of Mamba retrievers. We designed three training sets, each with a total of 600 million tokens. The first set purely contains documents of 2k tokens. The second set contains half 2k-token documents and half 5k-token documents. The third set contains an equal amount of 2k-token, 5k-token, and 10k-token documents. From table 24, we see the training set where 2k, 5k and 10k-token documents are mixed leads to the best Mamba performance. However, when we increase document length to 15k tokens, we observe unstable gradient norm behaviors that lead to quickly deteriorating performance of Mamba on validation sets, similar to the exploding gradient issue reported in state-space models Gu & Dao (2024); Dao & Gu (2024).

## E.3  ARE MAMBA RETRIEVERS LOST IN THE MIDDLE?

"Lost in the Middle" is a phenomenon identified by Liu et al. (2024a), where large language models (LLMs) tend to lose track of information in the middle of a long document, favoring information at the beginning and end. To investigate the behavior of Mamba retrievers when processing long

documents, we first identify useful and important information within a document and record their positions. We then examine whether Mamba retrievers are more likely to forget or ignore important information from specific locations within long documents.

We designed an LLM-powered pipeline that scans through a long document using a sliding window of 200 sentences, with a stride of 100 sentences. The goal is to identify all sentences potentially relevant to providing the ground-truth answer to a given question. We use GPT-4o, supplying it with both the question and the reference ground-truth answer for all data points with document lengths up to 120k tokens. Since the main paper employs a sliding window approach to aggregate logits produced by Mamba retrievers, it is not practical to investigate potential "lost in the middle" issues for documents exceeding 120k tokens.

With knowledge of both the question and the ground-truth answer, GPT-4o is better equipped to identify relevant sentences within a 200-sentence window. The sliding window approach is designed to mitigate potential long-context issues with GPT-4o.

After GPT-4o identifies relevant sentences in each window, we aggregate these sentences from different windows and present them to GPT-4o for a final selection. Once GPT-4o selects a final list of sentences, we ask it again whether these sentences can yield the correct ground-truth answer to the question. This step serves as a filtering process. After filtering, we have 3,067 data points with documents under 120k tokens. We manually reviewed a random subset of 200 data points to validate the quality of this pipeline.

We now have a set of 3,067 data points with documents annotated for relevant sentences. We also have Mamba retriever 1.3B top 50 retrieved sentences for each of these data points. For each relative position, we calculate the number of relevant sentences retrieved by Mamba retriever 1.3B, divided by the total number of relevant sentences found in that position. This metric is known as sentence recall at a certain relative position. Note that relative position is used because documents vary in length.

In Figure 12, we observed an interesting pattern. Mamba retriever's recall performance is noticeably better for smaller relative positions (i.e., the beginning of the document). Mamba retriever's recall performance drops to its lowest for the last 10% of relative positions (i.e., the end of the document). We also observed a general decreasing trend in Mamba retriever's recall as the relative position increases. This suggests that the Mamba retriever is less effective at retrieving relevant sentences when they are located farther from the beginning of the document (i.e., where the query is). While there is no discernible "lost in the middle" pattern in Figure 12, we did find that Mamba retriever tends to lose track at the end of the document.

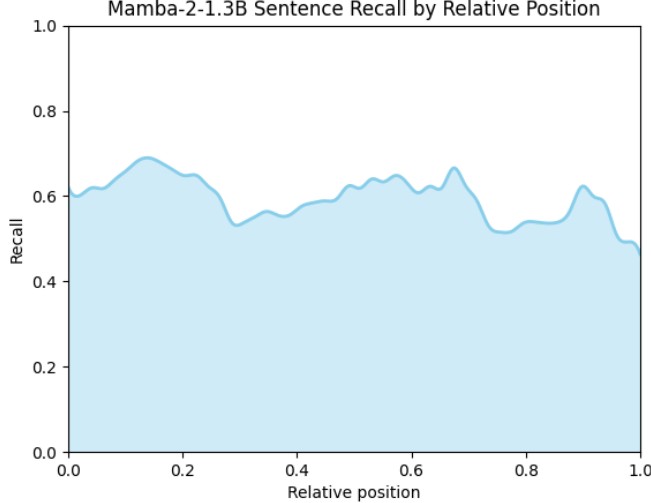

Figure 12: The recall of Mamba retriever 1.3B at different relative positions.

### E.4 RETRIEVAL COMPARISON BETWEEN MAMBA RETRIEVER AND NV-EMBED-v2

We demonstrate that the Mamba retriever retrieves more relevant context than the embedding model by comparing the sentences retrieved by Mamba retriever 1.3B and NV-Embed-v2-7B for the following data points in the test set (examples where NV-Embed-v2-7B retrieves 50 sentences are in Tables 25, 26; examples where NV-Embed-v2-7B retrieves 5 chunks are in Tables 27).

In Table 25, NV-Embed-v2-7B successfully retrieves the semantically relevant sentence *"Uh, like a test of availability,"* which aligns with the query about *"test components' availability."* However, NV-Embed-v2-7B failed to retrieve a crucial follow-up sentence identifying PERSON 7 as the individual responsible for the test. This limitation highlights that, while NV-Embed-v2-7B effectively identifies phrases with high semantic similarity, it failed to capture broader contextual relationships needed for comprehensive information retrieval. In contrast, the Mamba retriever 1.3B demonstrated a stronger contextual understanding by successfully retrieving the sentence *"So that's another thing that, that [PERSON7], uh, uh, uh, should <unintelligible> on,"* which was crucial for fully answering the question.

The example in Table 26 reveals NV-Embed's ability to handle conversational text. The model retrieves relevant dialogue between Castiel and Mr.Soren, where the conversation includes several keyword matches, such as "book," "Castiel," and "Mr. Soren" found in the query. However, identifying the book as "The History of the Devil" requires a deeper contextual understanding, as this connection established in earlier parts of the conversation. The Mamba retriever 1.3B demonstrates this capability by successfully retrieving the key sentence: *"The 'History of the Devil,' by Daniel Defoe,-not quite the right book for a little girl."* Additionally, with prior context, the Mamba retriever 1.3B also retrieves *"I advise you to put by the 'History of the Devil'."* This additional context enables the generator model to provide a more accurate response to the query about the conversation.

Table 27 compares the retrieval performance of Mamba retriever 1.3B and NV-Embed. In this comparison, Mamba retriever retrieves 50 sentences from the document, while NV-Embed-v2-7B retrieves 5 chunks. While NV-Embed's retrieved chunks contain multiple mentions of "MAVERICK" and "ROOSTER" that are semantically relevant to the query, the model misses crucial sentences that describe Rooster's frustration with Maverick for withdrawing his Naval Academy application. Specifically, in the "5 chunks" setting, NV-Embed-v2-7B fails to retrieve a key section where Rooster explicitly states: *"Maverick. He pulled my papers. He pulled my application to the Naval Academy. He set me back four years."* This omission results in the generator model producing a less accurate and incomplete response, as these sentences directly answer the query and provide vital context about the strained relationship between Rooster and Maverick. In contrast, Mamba retriever 1.3B successfully captures both the sentences that explicitly describe Rooster's frustration and Maverick's actions. As a result, the generator gives an attempted answer that aligns more closely with the reference answer.

Table 28 demonstrates another comparison between the Mamba retriever and NV-Embed-v2-7B when retrieving the top 5 chunks. The question asks about the age of Aelis' first husband and Birdy's father. To answer the question accurately, the model needs to retrieve sentences indicating that "LORD ROLLO" is Birdy's father. As shown in the left column, the Mamba retriever successfully retrieves this information and, using this context, retrieves the relevant sentence stating "Lord Rollo - 41 years of age." In contrast, the right column shows that NV-Embed-v2-7B retrieves a highly relevant-seeming chunk containing phrases such as "LORD SIDEBOTTOM, Aelis' father," "LORD GIDEON SIDEBOTTOM - 81 years of age - oldest man in his province - oldest father," and "Birdy." While this chunk includes information about "Aelis," "Birdy," ages, and "father," it fails to answer the specific question at hand. In the final portion of the text, both retrievers successfully obtain information about Aelis' husband's age. However, the key difference is that the LLM can provide a correct answer using the Mamba retriever's results, while it can only make an educated guess based on the incomplete information retrieved by NV-Embed.

Table 25: Example 1: Comparison of retrieval results between the Mamba retriever 1.3B and NV-Embed-v2-7B in the "50 sents" setting. A portion of the document is displayed, with retrieved sentences highlighted for both models in yellow. Information important for answering the question but missed by NV-Embed-v2-7B is highlighted in red in the text.

| | |
|---|---|
| *Question:* Who is in charge of writing the code to test components' availability during live demos?
*Reference Answer:* [PERSON7]. | |
| **Mamba retriever 1.3B** | **NV-Embed** |
| *Attempt:* [PERSON7] is in charge of writing the code to test components' availability during live demos. | *Attempt:* [PERSON13] is in charge of writing the code to test components' availability during live demos. |
| That we would know, uh, which of the parts of the pipeline are, performing badly in terms of translation quality.Uh, I just, uh-.It just occurred to me that there should be one more compilation target. And that would be like probing whether the components of the pipeline are up and running.Uh, like a test of availability. So that's another thing that, that [PERSON7], uh, uh, uh, should <unintelligible> on-. if you could put this on, uh, on the to do list or on the enhancement options, that would also be very useful.Uh, and another thing would be, uh, like live debugging, uh, of a of a pipeline such a speed of, uh, of that.Okay.And, uh, yes, and uh, so, and then the second item you have in your list [PERSON7].Please comment on that.(PERSON7) All, right.So, uh, next Friday, uh, like, next week somewhere there there is going to be a conference about [PROJECT13] and we are going to provide life subtitles and transcription.And because we will have some non native English speakers in there, so we will need to get some feedback, from the people that are using our subtitles.Preferab-, refe-, preferably life.So we can, uh, see a moments like, uh, where it was working and moments where it was not working.So, uh ,I will make some, uh, quick took-.(PERSON13) [PERSON15] already has such simple tool that you could adapt.Um, what is more-.What is missing is, uh, description, of, ah, like how to use the tool.And also more like a generic description of how people should look at the outputs.So, uh, it would be best, if you could get in touch with [PERSON18].Because I've asked [PERSON18] to like handle the soft, uh, soft things with the participants and also with the organisers.And, uh, um, you and [PERSON18] should prepare very simple instructions that the participants could follow. | That we would know, uh, which of the parts of the pipeline are, performing badly in terms of translation quality.Uh, I just, uh-.It just occurred to me that there should be one more compilation target. And that would be like probing whether the components of the pipeline are up and running.Uh, like a test of availability. So that's another thing that, that [PERSON7], uh, uh, uh, should <unintelligible> on-.. if you could put this on, uh, on the to do list or on the enhancement options, that would also be very useful.Uh, and another thing would be, uh, like live debugging, uh, of a of a pipeline such a speed of, uh, of that.Okay.And, uh, yes, and uh, so, and then the second item you have in your list [PERSON7].Please comment on that.(PERSON7) All, right.So, uh, next Friday, uh, like, next week somewhere there there is going to be a conference about [PROJECT13] and we are going to provide life subtitles and transcription.And because we will have some non native English speakers in there, so we will need to get some feedback, from the people that are using our subtitles.Preferab-, refe-, preferably life.So we can, uh, see a moments like, uh, where it was working and moments where it was not working.So, uh ,I will make some, uh, quick took-.(PERSON13) [PERSON15] already has such simple tool that you could adapt.Um, what is more-.What is missing is, uh, description, of, ah, like how to use the tool.And also more like a generic description of how people should look at the outputs.So, uh, it would be best, if you could get in touch with [PERSON18].Because I've asked [PERSON18] to like handle the soft, uh, soft things with the participants and also with the organisers.And, uh, um, you and [PERSON18] should prepare very simple instructions that the participants could follow. |

Table 26: Example 2: comparison of retrieval results between the Mamba retriever 1.3B and NV-Embed-v2-7B in the "50 sents" setting. A portion of the document is displayed, with retrieved sentences highlighted for both models in yellow. Information important for answering the question but missed by NV-Embed-v2-7B is highlighted in red in the text.

| | |
|---|---|
| *Question:* What book does Castiel show Mr. Soren that she is reading? | |
| *Reference Answer:* "The History of the Devil". | |
| **Mamba retriever 1.3B** | **NV-Embed** |
| *Attempt:* The 'History of the Devil,' by Daniel Defoe. | *Attempt:* "History of the Decline and Fall of the Roman Empire." |
| Mr. Roberta had listened to this exposition of Castiel's with petrifying wonder. "Why, what book is it the wench has got hold on?"he burst out at last. "The 'History of the Devil,' by Daniel Defoe,–not quite the right book for a little girl, " said Mr. Soren."How came it among your books, Mr.Roberta?" Castiel looked hurt and discouraged, while her father said,– "Why, it's one o' the books I bought at Partridge's sale. They was all bound alike,–it's a good binding, you see,–and I thought they'd be all good books. There's Sara Taylor's 'Holy Living and Dying' among 'em.I read in it often of a Sunday" (Mr. Roberta felt somehow a familiarity with that great writer, because his name was Sara); "and there's a lot more of 'em,–sermons mostly, I think,–but they've all got the same covers, and I thought they were all o' one sample, as you may say.But it seems one mustn't judge by th' outside.This is a puzzlin' world." "Well," said Mr. Soren, in an admonitory, patronizing tone as he patted Castiel on the head, "I advise you to put by the 'History of the Devil,' and read some prettier book. Have you no prettier books?""Oh, yes," said Castiel, reviving a little in the desire to vindicate the variety of her reading. "I know the reading in this book isn't pretty; but I like the pictures, and I make stories to the pictures out of my own head, you know.But I've got 'AEsop's Fables,' and a book about Kangaroos and things, and the 'Pilgrim's Progress.'""Ah, a beautiful book," said Mr. Soren; "you can't read a better." | Mr. Roberta had listened to this exposition of Castiel's with petrifying wonder. "Why, what book is it the wench has got hold on?"he burst out at last. "The 'History of the Devil,' by Daniel Defoe,–not quite the right book for a little girl, " said Mr. Soren."How came it among your books, Mr.Roberta?" Castiel looked hurt and discouraged, while her father said,– "Why, it's one o' the books I bought at Partridge's sale. They was all bound alike,–it's a good binding, you see,–and I thought they'd be all good books. There's Sara Taylor's 'Holy Living and Dying' among 'em.I read in it often of a Sunday" (Mr. Roberta felt somehow a familiarity with that great writer, because his name was Sara); "and there's a lot more of 'em,–sermons mostly, I think,–but they've all got the same covers, and I thought they were all o' one sample, as you may say.But it seems one mustn't judge by th' outside.This is a puzzlin' world." "Well," said Mr. Soren, in an admonitory, patronizing tone as he patted Castiel on the head, "I advise you to put by the 'History of the Devil,' and read some prettier book. Have you no prettier books?""Oh, yes," said Castiel, reviving a little in the desire to vindicate the variety of her reading. "I know the reading in this book isn't pretty; but I like the pictures, and I make stories to the pictures out of my own head, you know.But I've got 'AEsop's Fables,' and a book about Kangaroos and things, and the 'Pilgrim's Progress.'""Ah, a beautiful book," said Mr. Soren; "you can't read a better." |

Table 27: Example 3: Comparison of retrieval results between the Mamba retriever 1.3B and NV-Embed-v2-7B in the "5 chunks" setting. A portion of the document is displayed, with retrieved sentences highlighted for both models in yellow. Information important for answering the question but missed by NV-Embed-v2-7B is highlighted in red in the text.

| Question: Why do Rooster hate MAVERICK? |
|---|
| Reference Answer: Because MAVERICK pulled Rooster's application to the Naval academy. |

| Mamba retriever 1.3B | NV-Embed |
|---|---|
| Attempt: Rooster hates Maverick because Maverick pulled his application to the Naval Academy, setting him back four years, which Rooster sees as an unjust hindrance to his career. | Attempt: Rooster hates Maverick because he blames him for the incident involving his father's death. |
| SKIES - SORTIE 4 114 114 MAVERICK (TO SELF) Sorry, Rooster.[MAVERICK LEVELS OUT, STRIKES WITH A COBRA MANEUVER, FORCING ROOSTER AND HANGMAN TO SPLIT AND OVERSHOOT HIM.] Now Mav s instantly in chase position for a shot of his own.INT. ROOSTER'S F-18 - SORTIE 4 115 115 Rooster hears the tone. MAVERICK That s a kill.INT.ROOSTER S F-18 - SORTIE 4 116 116 Rooster seethes, outwitted, but concedes the fight... ROOSTER Copy kill.INT.READY ROOM - SORTIE 4 117 117 Everyone exhales, shares a collective look.This is next level shit, even for them.EXT.TARMAC - ELSEWHERE - DUSK 118 118 Close on Rooster, sweating and furious as he does push-ups on the tarmac, punishing himself. (CONTINUED)CHERRY 11.25.19 - OFFICIAL 62. 8FLiX.com FYC SCREENPLAY DATABASE 20221226HONDO Alright. That s enough man.Rooster, that s enough.Hondo pats Rooster on the shoulder. HONDO (CONT D) Tomorrow s another day.Rooster sits up, exhausted.Feet appear next to him.He looks up to see Phoenix above him.PHOENIX What is going on with you? You trying to get kicked out?Breaking the hard deck.Insubordination.That wasn t you up there. Talk to me.What s up?ROOSTER Don t worry about it. PHOENIX I m going on this mission.But if you get kicked out, you could leave us flying with Hangman.So what the hell was that- ROOSTER HE PULLED MY PAPERS. PHOENIX What?Who?ROOSTER Maverick. He pulled my application to the Naval academy. He set me back four years.Phoenix processes.PHOENIX Why would he do that? Rooster does not answer.INT.READY ROOM 119 119 Hangman is staring at something on the wall. HANGMAN Yo, Coyote.CONTINUED: 118 118 (CONTINUED)CHERRY 11.25.19 - OFFICIAL 63. 8FLiX.com FYC SCREENPLAY DATABASE 20221226Coyote walks over and follows Hangman s eyes to a photo from the CLASS OF 86. | SKIES - SORTIE 4 114 114 MAVERICK (TO SELF) Sorry, Rooster.[MAVERICK LEVELS OUT, STRIKES WITH A COBRA MANEUVER, FORCING ROOSTER AND HANGMAN TO SPLIT AND OVERSHOOT HIM.] Now Mav s instantly in chase position for a shot of his own.INT. ROOSTER'S F-18 - SORTIE 4 115 115 Rooster hears the tone. MAVERICK That s a kill.INT.ROOSTER S F-18 - SORTIE 4 116 116 Rooster seethes, outwitted, but concedes the fight... ROOSTER Copy kill.INT.READY ROOM - SORTIE 4 117 117 Everyone exhales, shares a collective look.This is next level shit, even for them.EXT.TARMAC - ELSEWHERE - DUSK 118 118 Close on Rooster, sweating and furious as he does push-ups on the tarmac, punishing himself. (CONTINUED)CHERRY 11.25.19 - OFFICIAL 62. 8FLiX.com FYC SCREENPLAY DATABASE 20221226HONDO Alright. That s enough man.Rooster, that s enough.Hondo pats Rooster on the shoulder. HONDO (CONT D) Tomorrow s another day.Rooster sits up, exhausted.Feet appear next to him.He looks up to see Phoenix above him.PHOENIX What is going on with you? You trying to get kicked out?Breaking the hard deck.Insubordination.That wasn t you up there. Talk to me.What s up?ROOSTER Don t worry about it. PHOENIX I m going on this mission.But if you get kicked out, you could leave us flying with Hangman.So what the hell was that- ROOSTER HE PULLED MY PAPERS. PHOENIX What?Who?ROOSTER Maverick. He pulled my application to the Naval academy. He set me back four years.Phoenix processes.PHOENIX Why would he do that? Rooster does not answer.INT.READY ROOM 119 119 Hangman is staring at something on the wall. HANGMAN Yo, Coyote.CONTINUED: 118 118 (CONTINUED)CHERRY 11.25.19 - OFFICIAL 63. 8FLiX.com FYC SCREENPLAY DATABASE 20221226Coyote walks over and follows Hangman s eyes to a photo from the CLASS OF 86. |

Table 28: Example 4: Comparison of retrieval results between the Mamba retriever 1.3B and NV-Embed-v2-7B in the "5 chunks" setting. A portion of the document is displayed, with ==retrieved sentences== highlighted for both models in ==yellow==. Information important for answering the question but missed by NV-Embed-v2-7B is highlighted in red in the text.

| Question: How much younger is Aelis' first husband than Birdy's father? | |
|---|---|
| Reference Answer: 32 years. | |
| **Mamba retriever 1.3B** | **NV-Embed** |
| Attempt: Aelis' first husband is 32 years younger than Birdy's father. | Attempt: Aelis' first husband is approximately 31-51 years younger than Birdy's father. |
| It is I, Birdy.OVER THE IMAGE, A HAND WRITES, AS IF ON AN ILLUMINATED MANUSCRIPT: CATHERINE CALLED BIRDY. INT.STONEBRIDGE MANOR-SOLAR- SAME TIME- MORNING ==This is Birdy s father, LORD ROLLO S man cave, hung with variously sized antlers and evidence of violent past times. BIRDY (V.O.)I am the Daughter of Lord Rollo.TEXT ON SCREEN: Lord Rollo - 41 years of age- often vain- usually drunk- always greedy (says me)== He takes a drink. Then another...(3000 words omitted)...LORD SIDEBOTTOM, Aelis s father, is nearing seventy but still clanking his old bones together in a push chair that rolls between the two seats. TEXT ON SCREEN: LORD GIDEON SIDE-BOTTOM ==- 81 years of age- oldest man in his province- oldest father in England- wears his armour to sleep BERENICE, Aelis s gorgeous young stepmum, looks a thousand times more bored than AISLINN.== She is rife with the en-nui of entrapment.Aelis leans over the cart s edge and shyly returns Birdy s joyful wave. EXT.STONEBRIDGE MANOR- COURTYARD-MOMENTS LATER- DAY Birdy and Aelis have sequestered themselves gleefully from the grownups on a bench. Aelis bends down behind Birdy, playing with her hair.AELIS Your hair is so long Birdy. You need to brush it.BIRDY I m going to grow it all the way down to my feet...(7000 words omitted)...==AELIS Birdy, I am to be married.BIRDY (stricken) To George? AELIS No, to a boy of only nine.==George has to marry some horrid old widow named Ethelfritha. And now you will not even be my friend!Aelis rushes out.Birdy looks at the nun wearily. BIRDY (V.O.)For the first time in my life, I am choking on my words. My heart has been shaved and boiled like a parsnip.George is to be married. George is to be married.George.Is.To.Be.Married.Birdy looks at the nun wearily. BIRDY I suppose you re not taking joiners at the convent. | It is I, Birdy.OVER THE IMAGE, A HAND WRITES, AS IF ON AN ILLUMINATED MANUSCRIPT: CATHERINE CALLED BIRDY. INT.STONEBRIDGE MANOR-SOLAR- SAME TIME- MORNING This is Birdy s father, LORD ROLLO S man cave, hung with variously sized antlers and evidence of violent past times. BIRDY (V.O.)I am the Daughter of Lord Rollo.TEXT ON SCREEN: Lord Rollo - 41 years of age- often vain- usually drunk- always greedy (says me) He takes a drink. Then another...(3000 words omitted)...==LORD SIDEBOTTOM, Aelis s father, is nearing seventy but still clanking his old bones together in a push chair that rolls between the two seats. TEXT ON SCREEN: LORD GIDEON SIDE-BOTTOM - 81 years of age- oldest man in his province- oldest father in England- wears his armour to sleep BERENICE, Aelis s gorgeous young stepmum, looks a thousand times more bored than AISLINN. She is rife with the en-nui of entrapment.Aelis leans over the cart s edge and shyly returns Birdy s joyful wave. EXT.STONEBRIDGE MANOR- COURTYARD-MOMENTS LATER- DAY Birdy and Aelis have sequestered themselves gleefully from the grownups on a bench. Aelis bends down behind Birdy, playing with her hair.AELIS Your hair is so long Birdy. You need to brush it.BIRDY I m going to grow it all the way down to my feet...(7000 words omitted)...AELIS Birdy, I am to be married.BIRDY (stricken) To George? AELIS No, to a boy of only nine.George has to marry some horrid old widow named Ethelfritha. And now you will not even be my friend!Aelis rushes out.Birdy looks at the nun wearily. BIRDY (V.O.)For the first time in my life, I am choking on my words. My heart has been shaved and boiled like a parsnip.George is to be married. George is to be married.George.Is.To.Be.Married.Birdy looks at the nun wearily. BIRDY I suppose you re not taking joiners at the convent.== |

# F  FORMULATION OF MAMBA RETRIEVER

During training, formally, given a query $Q$ and a document $D$, let $n$ be the number of sentences in $D$. We are also given the list of binary relevance labels for these $n$ sentences as $R = [r_1, \ldots, r_n]$, $r_i \in \{0, 1\}$. The relevance labels and query are generated by our link-based synthetic data method.

Mamba retriever takes in $(Q, D, R)$ and output a list of logit values $z = [z_1, \ldots, z_n]$ corresponding to each sentence in $D$. Each logit value represents the degree of relevance that the preceding sentence holds towards $Q$. Using $R$ and $z$, the binary cross entropy loss is used to train Mamba retriever

$$\text{Mamba Retriever}(Q, D) = z; \quad \text{Binary Cross Entropy Loss}(R, z)$$

Specifically, given a query $Q$ and a document $D$, we concatenate them and tokenize $Q + D$ into a list of tokens $u_0, u_1, \ldots, u_T$, where $T$ represents the time axis. Denote the index of the last token of each sentence as $s_1, \ldots, s_n$ where $0 < s_i \leq T$. Note $s_n = T$.

For each $s_i$ position, during training, there is a binary label $r_i$ pre-assigned to it.

Following Mamba-2 (Dao & Gu, 2024), we denote the head dimension as $P$, and we denote the state expansion factor as $N$. WLOG, we assume the number of head is 1, so the model dimension is also $P$.

The input list of tokens $u_0, \ldots, u_T$ are projected to latent space as $x_0, \ldots, x_T$ where $x_t \in \mathbb{R}$

We give the recursion formula that maps a 1-dimensional sequence $x_t \in \mathbb{R} \mapsto y_t \in \mathbb{R}$ through an implicit latent state $h_t \in \mathbb{R}^N$

$$h_0 = Bx_0 \quad \ldots \quad \begin{cases} h_t = Ah_{t-1} + Bx_t \\ y_t = C^T h_t \end{cases} \quad \ldots \quad \begin{cases} h_T = Ah_{T-1} + Bx_T \\ y_T = C^T h_T \end{cases}$$

where $A \in \mathbb{R}^{N \times N}, B \in \mathbb{R}^{N \times 1}, C \in \mathbb{R}^{N \times 1}$.

The above equation defines a sequence transformation for $P = 1$, and it can be generalized to $P > 1$ for $x_t, y_t \in \mathbb{R}^P$ by broadcasting across this dimension.

We denote the binary classification head as $H \in \mathbb{R}^{P \times 1}$, and logit $z_t$ can be computed as

$$z_t = y_t H$$

We then give the formula for our loss function.

Note that we are only interested in the end of sentence logits $z_{s_1}, z_{s_2}, \ldots, z_{s_n}$ with corresponding labels $r_1, r_2, \ldots, r_n$.

We use Binary Cross Entropy loss

$$\sum_{i=1}^{n} -w_i \Big[ \big( r_i \log z_{s_i} + (1 - r_i) \log(1 - z_{s_i}) \big) \Big]$$

where $w_i$ is a data-dependent weight to upsample the number of positive labels, due to a class imbalance issue

$$w_i = \begin{cases} \frac{1}{2n - 2\sum_{j=1}^{n} r_j}, & \text{if } r_i = 0, \\ \\ \frac{1}{2\sum_{j=1}^{n} r_j}, & \text{otherwise } r_i = 1 \end{cases}$$