# OpenReview forum: "Efficient Full-Context Retrieval for Long Documents"
_ICLR.cc/2025/Conference — Submitted to ICLR 2025_

### Official Review · Reviewer_wS7S · 2024-10-17

**Soundness:** 3
**Presentation:** 3
**Contribution:** 3
**Rating:** 6
**Confidence:** 4

**Summary:**

The authors introduce a novel approach for long document understanding using the Mamba architecture as an efficient information retriever. Unlike traditional methods that rely on document chunking, this approach processes the entire document, maintaining important contextual dependencies that chunking-based methods may lose. To overcome the challenge of limited labeled datasets for training such a model, the authors employ synthetic data generation strategies—specifically chunk-based, pair-based, and link-based approaches—prompting an LLM to the training examples. The model, trained on both 130M and 1.3B parameter variants, is evaluated on 41 benchmark tasks, spanning domains such as financial documents, government reports, and creative works. The results demonstrate that their Mamba retriever consistently outperforms embedding-based retrievers and exhibits near-GPT-4-level performance on very long documents exceeding 256K tokens. Ablation studies further highlight the importance of using full-document context and the benefits of the link-based synthetic training dataset, which effectively improves retrieval accuracy compared to the other data generation strategies.

**Strengths:**

- The proposed approach effectively removes the need for document chunking, enabling a more comprehensive and uninterrupted representation of document context. This is particularly advantageous for handling long documents where preserving contextual relationships is crucial.
- The Mamba-2 model, available in both 130M and 1.3B parameter variants, demonstrates impressive performance, notably surpassing models significantly larger in size, such as GritLM-7B. This result underscores the efficiency of the model design, making it a valuable solution for environments with limited computational resources.
- To address the lack of annotated datasets for training their long-context models, the authors introduce diverse synthetic data generation methods, including chunk-based, pair-based, and link-based strategies. These approaches help to create high-quality training data that may enable effective model learning.
- The paper includes a well-executed analysis, supported by ablation studies that provide insight into the contributions of different model components and training strategies.

**Weaknesses:**

I found the proposed idea, experiments, and analyses conducted by the authors to be valuable, especially in terms of their potential impact on low-resource scenarios. However, for the paper to fully meet the ICLR standards, there are still areas that need additional work and detail. Below, I outline several key points for improvement. I would be pleased to substantially raise my scores if the authors address these suggestions and enhance the paper accordingly.

**General Feedback**
- I noticed that the title of the paper does not match the one listed on OpenReview.
- The main text should indicate when additional detailed discussions are deferred to the Appendix for better reader guidance.

**Introduction**

- The Introduction lacks foundational references to support key claims. Both the second and third paragraphs would benefit from citations to strengthen the arguments. For instance, the statement: "This method eliminates the need for document chunking, *a common limitation in current retrieval systems that often results in loss of context and reduced accuracy*" needs a supporting citation to substantiate this point.
- The sentence: "Second, to be competitive with embedding approaches, a retrieval language model needs to be small" requires further justification. The authors should include in the paper a complexity analysis comparison discussing time and GPU memory consumption to support this assertion.

**Related Work**

- The sentence "Large Language Models are found to be inefficient processing long-context documents" should be rewritten for clarity, for example: "Large Language Models are inefficient when processing long-context documents."
- The statements "Transformer models suffer from quadratic computation during training and linear computation during inference" and "However, transformer-based models are infeasible to process extremely long documents due to their linear inference time" are incorrect. Transformers, as presented in "Attention is All You Need," scale quadratically in both training and inference.
- The statement regarding State Space Models (SSMs) having "linear scaling during training and constant scaling during inference" is inaccurate. SSMs have linear complexity for both training and inference. The term "constant scaling" implies no dependence on sequence length, which is incorrect.
- The Related Work section is lacking details. The paragraph on long-context language models should provide a more comprehensive overview of existing methods and their limitations, positioning SSMs appropriately. This includes discussing sparse-attention mechanisms [1, 2], segmentation-based approaches [3, 4, 5], memory-enhanced segmentation strategies [6], and recursive methods [7] for handling very long documents.
- Similarly, the paragraph on Retrieval-Augmented Generation should specify how prior works addressed different long document tasks. Examples include successful applications of RAG in long-document summarization [8, 9] and query-focused multi-document summarization [10, 11], which are closely aligned with the present work.

**Figures**
- Figures 1 and 2 are clear but need aesthetic improvements to meet the conference's standard presentation quality.

**Model Architecture**
- The description "a subset of tokens are specially designated, and the classification head is applied to these tokens. In the current work, the classification head is applied to the last token of each sentence, giving sentence-level resolution" is ambiguous. Clarify whether new tokens are added to the sequence or if existing tokens (e.g., periods) are used to represent sentence ends.

**Synthetic Data Generation**

- The "lost in the middle" problem when processing long documents [12] is not explicitly discussed. Have the authors considered the position of chunks during synthetic data generation? Ablation studies varying the position and distance between linked chunks would provide valuable insights into Mamba’s effectiveness in addressing this issue.
- More details are needed regarding the data decontamination pipeline, chunk size, and the relative computational cost of the link-based method versus other strategies.
- The authors claim that synthetic data generation is computationally expensive but provide no supporting quantitative evidence. Information such as time estimates and GPU demand would strengthen this argument and assess feasibility.
- There is no detailed evaluation of the synthetic data’s quality. An analysis of correctness and answer factuality would help validate the impact on retrieval performance beyond benchmark metrics.

**Training**
- This section is too brief. Consider merging it with Section 3, "Model Architecture," for a more cohesive presentation.
- What was the training time for the 130M model?

**Experimental Method**
- Fix minor formatting issues, such as adding a space after the comma in ",LVeval."
- Specify in Table 1 which datasets use free-form versus multiple-choice answers, including the number of answers and average answer lengths.
- Consider experimenting with GPT-4 as a retriever.
- Expand on "The accuracy of freeform answers is judged using GPT-4."
- Elaborate on the validation of the scoring pipeline, particularly regarding "0.942 macro F1." Clarify the data and method used for validation.
- Justify the selection of "50 sentences" for Mamba retrievers and explain chunk creation methods for embedding models. Did the chunks consist of 300 fixed-length segments, or was semantic chunking employed [3, 5]? Sentence-level embedding-based retrieval could be explored to align better with the Mamba setting.
- The assertion that "embedding models were allowed to retrieve more information than Mamba" implies an unfair comparison, but more context can sometimes degrade performance [12].
- Clarify the use of the sliding window approach for documents longer than 128k tokens, especially given the claim that Mamba could process up to 256K tokens directly.

**Results**
- Remove redundancy in Section 7.1.2, such as restating the synthetic data generation strategies.
- Expand the ablation studies to cover different input sequence lengths during training and varying the number of retrieved sentences to explore robustness to configuration changes.
- Highlight that using fewer training examples (500K vs. 1M) achieved comparable accuracy (i.e., 59.4 vs. 60.0, respectively).
- Why not train both the 130M and 1.3B models on a dataset size of 500K examples, but compare using 1M and 400K examples, respectively?

**Limitations**
- The high cost of generating synthetic training data is mentioned but lacks quantification. How computationally expensive is it in terms of time or resources?

**Appendix**
- Note that all figures in Appendices B and C are the same, suggesting an error that needs correcting.

**Missing References**

[1] Longformer: The Long-Document Transformer. arXiv 2020.

[2] LongT5: Efficient Text-To-Text Transformer for Long Sequences. NAACL 2022.

[3] Semantic Self-Segmentation for Abstractive Summarization of Long Documents in Low-Resource Regimes. AAAI 2022.

[4] Summ^n: A Multi-Stage Summarization Framework for Long Input Dialogues and Documents. ACL 2022.

[5] Align-Then-Abstract Representation Learning for Low-Resource Summarization. Neurocomputing 2023.

[6] Efficient Memory-Enhanced Transformer for Long-Document Summarization in Low-Resource Regimes. Sensors 2023.

[7] Recursively Summarizing Books with Human Feedback. arXiv 2021.

[8] DYLE: Dynamic Latent Extraction for Abstractive Long-Input Summarization. ACL 2022.

[9] Towards a Robust Retrieval-Based Summarization System. arXiv 2024.

[10] Discriminative Marginalized Probabilistic Neural Method for Multi-Document Summarization of Medical Literature. ACL 2022.

[11] Retrieve-and-Rank End-to-End Summarization of Biomedical Studies. SISAP 2023.

[12] Lost in the Middle: How Language Models Use Long Contexts. TACL 2024.

**Questions:**

Please address the questions outlined in the weaknesses section.

---

> ### Author Response · Authors · 2024-11-22
> **Author Response with a Revised Paper**
>
> Thank you for your insightful responses. We have rewritten our paper. All changes we made are reflected in red color.
>
> >I noticed that the title of the paper does not match the one listed on OpenReview.
>
> Thank you for pointing this out. We have revised the title.
>
> >The main text should indicate when additional detailed discussions are deferred to the Appendix for better reader guidance.
>
> In the updated paper, we have added additional references to the appendix.
>
> >The Introduction lacks foundational references to support key claims. Both the second and third paragraphs would benefit from citations to strengthen the arguments. For instance, the statement: "This method eliminates the need for document chunking, a common limitation in current retrieval systems that often results in loss of context and reduced accuracy" needs a supporting citation to substantiate this point.
>
> We have rewritten the introduction and added proper citations. We included all the citations you suggested in Related Work.
>
> >The sentence: "Second, to be competitive with embedding approaches, a retrieval language model needs to be small" requires further justification. The authors should include in the paper a complexity analysis comparison discussing time and GPU memory consumption to support this assertion.
>
> We compare the Mamba retriever with three state-of-the-art embedding models, including the leader of the Massive Text Embedding Benchmark (MTEB), evaluating their performance in terms of document processing speed, FLOPs, and parameters.
>
> | Model                  | Retrieval Setting | Speed (ms) | TFLOPS     | TFLOPS w/o Pad | Params (billions) | Average Accuracy |
> |------------------------|-------------------|------------|------------|----------------|-------------------|------------------|
> | Mamba retriever 130M   | 50 sents          | **93.4**   | **19.0**   | **19.0**       | **0.1**           | 60.0             |
> | Mamba retriever 1.3B   | 50 sents          | 181.6      | 197.9      | 197.9          | 1.3               | **61.8**         |
> | NV-Embed-v2-7B         | 50 sents          | 592.0      | 1316.7     | 1279.4         | 7.9               | 56.6             |
> | NV-Embed-v2-7B         | 5 chunks          | 470.8      | 1295.6     | 1287.5         | 7.9               | 59.1             |
> | Stella-1.5B            | 50 sents          | 364.7      | 331.9      | 210.5          | 1.5               | 55.6             |
> | Stella-1.5B            | 5 chunks          | 264.8      | 244.8      | 219.0          | 1.5               | 56.8             |
> | GTE-Qwen2-1.5B         | 50 sents          | 364.4      | 331.9      | 210.5          | 1.5               | 54.6             |
> | GTE-Qwen2-1.5B         | 5 chunks          | 264.9      | 244.8      | 219.0          | 1.5               | 55.7             |
> | Llama-3.1-70B          | Direct Answer     | N/A        | 28,517.9   | 28,517.9       | 69.5              | 57.8             |
>
> Mamba retrievers achieve better document processing speed and accuracy than the embedding models. The most accurate embedding model is the 7B NV-Embed-v2, which uses many more FLOPs than the 1.3B or 130M Mamba retriever. The 130M Mamba retriever achieves higher accuracy than any of the embedding models, while using a fraction of the FLOPs and achieving much faster document processing time.
>
> We also included this discussion in Section 5.6, Table 2, Line 373-377, thanks to your suggestion.
>
> > Related Work [...]
>
> Thank you for your insightful comments. We have adapted the suggestions in our revised paper's related work section including rewriting sentences for clarity, adding more details and adding missing references.
>
> > Figures 1 and 2 are clear but need aesthetic improvements to meet the conference's standard presentation quality.
>
> Thank you for the suggestion. We have combined and improved the two figures, which are now presented as Figure 1 in the revised paper.
>
> >The description "a subset of tokens are specially designated, and the classification head is applied to these tokens. In the current work, the classification head is applied to the last token of each sentence, giving sentence-level resolution" is ambiguous. Clarify whether new tokens are added to the sequence or if existing tokens (e.g., periods) are used to represent sentence ends.
>
> We revisited the entire section to enhance clarity and provide additional details.
>
> No new token is introduced. The last token of a sentence (e.g., a period) is used to represent the end of the sentence. See Line 104-136 in Section 3 Model Architecture of Mamba Retriever.

---

> > ### Author Response · Authors · 2024-11-22
> > **Author Response Part 2**
> >
> > >The "lost in the middle" problem when processing long documents [12] is not explicitly discussed. Have the authors considered the position of chunks during synthetic data generation? Ablation studies varying the position and distance between linked chunks would provide valuable insights into Mamba’s effectiveness in addressing this issue.
> >
> > We will answer this question in two parts.
> >
> > First, we performed an ablation study to investigate if relative position of linked chunks would have any effect on Mamba retriever’s performance. See Appendix E.1. We found the larger the linked chunks’ relative positions are, the better the Mamba retriever’s performance is, which suggests increasing the distance between linked chunks and the query (which is at the beginning of the document) would make training data more challenging and more beneficial for Mamba retrievers. See Table 23.
> >
> > | **Linked-Chunks' Relative Positions**                        | **educational** (n = 1967) | **creative** (n = 1733) | **official** (n = 1328) | **conversational** (n = 707) | **Average Accuracy** |
> > |-----------------------------------------------------|---------------------------|-------------------------|-------------------------|-----------------------------|-----------------------|
> > | • Both in 0-33% of the document                    | 55.8                      | 27.5                    | 38.3                    | 37.5                        | 39.5                  |
> > | • Both in 33-67% of the document                   | 56.5                      | 30.6                    | 41.8                    | 38.9                        | 42.0                  |
> > | • Both in 67-100% of the document                  | 63.0                      | 41.5                    | 49.8                    | 45.1                        | 50.9                  |
> >
> > Second, we performed an experiment to measure recall at different positions within a document. The experiment measured the Mamba retriever’s success at identifying relevant sentences within a document, as a function of those sentences’ relative position in the document. We provide details about the methodology in Appendix E.3.
> > In Figure 12, we observed an interesting pattern. Mamba retriever's recall performance is noticeably better for smaller relative positions (i.e., the beginning of the document). The recall drops to its lowest for the last 10% of relative positions (i.e., the end of the document). Here is the recall for each decile of the relative positions:
> >
> > | Relative Position     | Value  |
> > |------------|--------|
> > | 0-10%      | 0.624  |
> > | 10-20%     | 0.678  |
> > | 20-30%     | 0.585  |
> > | 30-40%     | 0.562  |
> > | 40-50%     | 0.607  |
> > | 50-60%     | 0.634  |
> > | 60-70%     | 0.622  |
> > | 70-80%     | 0.529  |
> > | 80-90%     | 0.581  |
> > | 90-100%    | 0.491  |
> >
> > The results suggest that the Mamba retriever is less effective at retrieving relevant sentences when they are located farther from the beginning of the document (i.e., where the query is). While there is no discernible "lost in the middle" pattern in Figure 12, we did find that Mamba retrievers tend to lose track at the end of the document.

---

> > > ### Author Response · Authors · 2024-11-22
> > > **Author Response Part 3**
> > >
> > > > More details are needed regarding the data decontamination pipeline, chunk size, and the relative computational cost of the link-based method versus other strategies.
> > >
> > > Thank you for pointing out these missing details. We have rewritten the synthetic data generation section to include more details.
> > >
> > > All the test sets are derived from 41 benchmark datasets. Detailed statistics for each of these datasets are provided in Appendix A.1, A.2.
> > >
> > > At a high level, the decontamination procedure removed any training documents that had more than 1% sentence overlap with the test set. More precisely, we performed sentence tokenization on all documents from our 41 test sets, resulting in a pool of 2.4 million test sentences. Next, we performed sentence tokenization on each training document, and used string matching to calculate the overlap with the pool of test sentences. We removed any training document where more than 1% of its sentences matched those in the test pool of 2.4 million sentences.
> > >
> > > The longest training document contains fewer than 1,000 sentences. Our decontamination procedure ensures that fewer than 10 sentences in this document are included in the test pool. This guarantees there is no meaningful textual overlap between the test and training sets.
> > >
> > > We have clarified our decontamination strategy in Section 4.3.
> > >
> > > For chunk-based, pair-based and link-based generation, we use a chunk size of 20 sentences. (Line 158, Line 185-186, Line 199)
> > >
> > > We have added these details in Section 4 Synthetic Data Generation.
> > >
> > > All synthetic data were generated using GPT-4o-mini, where the computational cost directly correlates with financial cost. In Table 4, we have updated the cost for each synthetic data generation method. Although our link-based method incurs higher costs, it produces higher-quality synthetic data, which leads to improved performance for Mamba retrievers.
> > >
> > > | Strategy       | Average Accuracy | Input Token (B) | Output Token (B) | Cost (\$) per 1 Million Examples |
> > > |----------------|------------------|-----------------|------------------|-----------------------------------|
> > > | Chunk-based    | 57.2             | 0.76           | 0.10            | 71                                |
> > > | Pair-based     | 51.4             | 1.49           | 0.37            | 167                               |
> > > | Link-based     | 59.4             | 7.79           | 1.64            | 1076                              |
> > >
> > > >The authors claim that synthetic data generation is computationally expensive but provide no supporting quantitative evidence. Information such as time estimates and GPU demand would strengthen this argument and assess feasibility.
> > >
> > > We have clarified this in the limitations section (Line 538-539) and added a comparison of the financial costs for the link-based method versus other strategies in Table 4. There are no direct GPU costs, since data were generated using the OpenAI API. 1 million link-based synthetic examples cost around $1076.
> > >
> > > >There is no detailed evaluation of the synthetic data’s quality. An analysis of correctness and answer factuality would help validate the impact on retrieval performance beyond benchmark metrics.
> > >
> > > In Appendix B.4 and Tables 20, 21, and 22, we provide two examples for each synthetic data generation strategy, along with an evaluation of their quality. Our observations reveal that the link-based strategy produces the most coherent questions, with relevant sentences accurately labeled. In contrast, the pair-based method generates unnatural questions and poorly labeled sentence-level relevance. Meanwhile, the chunk-based method produces questions with obvious textual overlap with the labeled relevant sentences, making them less effective for training.
> > >
> > > >Training
> > > >
> > > >This section is too brief. Consider merging it with Section 3, "Model Architecture," for a more cohesive presentation.
> > > >
> > > >What was the training time for the 130M model?
> > >
> > > We have merged this section into Section 5.5 and included the training time for the 130M model, which is three hours on eight H100s (Line 292-294).
> > >
> > > >Fix minor formatting issues, such as adding a space after the comma in ",LVeval."
> > >
> > > We fixed this in the revised paper.
> > >
> > > >Specify in Table 1 which datasets use free-form versus multiple-choice answers, including the number of answers and average answer lengths.
> > >
> > > We moved the table describing test sets statistics to Appendix A.1 Table 7, due to page limit. We will put it back on the extra page of the final version if we are given the opportunity. We have now provided the information you requested in Table 7.

---

> > > > ### Author Response · Authors · 2024-11-22
> > > > **Author Response Part 4**
> > > >
> > > > >Consider experimenting with GPT-4 as a retriever.
> > > >
> > > > Thank you for highlighting an interesting experiment. We have evaluated GPT-4o and Llama-3.1 70B as a retriever and included the results and analysis in Section 7.3 Table 5.
> > > >
> > > > | Retriever Type  | Generative                     |                  | Discriminative               |                  |
> > > > |------------------|--------------------------------|------------------|--------------------------------|------------------|
> > > > | **Model**           | GPT-4o                         | Llama-3.1 70B    | Mamba retriever 130M          | Mamba retriever 1.3B |
> > > > |**Average Accuracy** | 52.2                          | 45.9             | 60.0                           | 61.8             |
> > > >
> > > > All generative retrievers perform significantly worse than discriminative Mamba retrievers. This suggests that retrieval by LLMs is often lossy in long-context settings, further highlighting the advantages of using discriminative retrieval with our method.
> > > >
> > > > >Expand on "The accuracy of freeform answers is judged using GPT-4."
> > > > >
> > > > >Elaborate on the validation of the scoring pipeline, particularly regarding "0.942 macro F1." Clarify the data and method used for validation.
> > > >
> > > > The accuracy of freeform answers is assessed using GPT-4o-0806, which is prompted to compare attempted answers with ground-truth answers, resulting in a binary "yes" or "no" judgment. This prompt is developed from 100 human-annotated examples. On a separate held-out test set of 180 human-annotated examples, GPT-4o's 180 yes/no judgments show high agreement with human judgments, achieving a macro F1 score of 0.942.
> > > >
> > > > For the prompt, refer to Appendix C. The method is now described in Section 5.3, titled "GPT-4o as Judge."
> > > >
> > > > >Justify the selection of "50 sentences" for Mamba retrievers and explain chunk creation methods for embedding models. Did the chunks consist of 300 fixed-length segments, or was semantic chunking employed [3, 5]? Sentence-level embedding-based retrieval could be explored to align better with the Mamba setting.
> > > >
> > > > Mamba retriever is a sentence-level retriever. We use 50 sentences because it on average contains 1600 tokens which align closely with a standard “5 chunks” of 300 words setting used in RAG with embedding models [(Xu et al. 2023, ](https://arxiv.org/abs/2310.03025) [Li et al. 2024)](https://arxiv.org/abs/2407.16833).
> > > > Each chunk has a fixed-length of 300 words, and approximately 400 tokens. No semantic chunking is employed in the current version.
> > > > We have also provided model performance for all embedding models that are asked to retrieve “50 sentences” instead of “5 chunks”, which are strictly worse than the “5 chunks” setup. See Table below.
> > > >
> > > > >The assertion that "embedding models were allowed to retrieve more information than Mamba" implies an unfair comparison, but more context can sometimes degrade performance [12].
> > > >
> > > > The Table below demonstrates that 5-chunk retrieval always performs better than 50-sentence retrieval across the embedding models. These settings are now compared and discussed in the paper (Line 254-261, Line 322-323, Line 347, Table 2). All embedding models' performance on “50 sentences” and “5 chunks” are reported in Appendix A.4 Table 10-14.
> > > >
> > > > |                     Embedding Model                     | Retrieval Setting | Accuracy (%) |
> > > > |---------------------------------------------------------|-------------------|--------------|
> > > > | **BM25**                                                | 50 sents          | 44.6         |
> > > > |                                                         | 5 chunks          | 49.1         |
> > > > | **Dragon-110M**                                         | 50 sents          | 52.8         |
> > > > |                                                         | 5 chunks          | 53.9         |
> > > > | **Contriever-110M**                                     | 50 sents          | 53.1         |
> > > > |                                                         | 5 chunks          | 54.3         |
> > > > | **GTE-Qwen2-1.5B**                                      | 50 sents          | 54.6         |
> > > > |                                                         | 5 chunks          | 55.7         |
> > > > | **Stella-1.5B**                                         | 50 sents          | 55.6         |
> > > > |                                                         | 5 chunks          | 56.8         |
> > > > | **OpenAI v3-large**                                     | 50 sents          | 55.4         |
> > > > |                                                         | 5 chunks          | 57.6         |
> > > > | **GritLM-7B**                                           | 50 sents          | 56.7         |
> > > > |                                                         | 5 chunks          | 57.2         |
> > > > | **NV-Embed-v2-7B**                                      | 50 sents          | 56.6         |
> > > > |                                                         | 5 chunks          | 59.1         |

---

> > > > > ### Author Response · Authors · 2024-11-22
> > > > > **Author Response Part 5**
> > > > >
> > > > > >Clarify the use of the sliding window approach for documents longer than 128k tokens, especially given the claim that Mamba could process up to 256K tokens directly.
> > > > >
> > > > > The context length for GPT-4o is 128k. To ensure a fair comparison, we set the context window of the Mamba Retriever to the same length. A more detailed explanation of the sliding window approach can be found in Section 5.4 of the revised paper (Line 276 - 282).
> > > > >
> > > > > >Remove redundancy in Section 7.1.2, such as restating the synthetic data generation strategies.
> > > > >
> > > > > Thank you for the suggestion. We removed the redundancy.
> > > > >
> > > > > >Expand the ablation studies to cover different input sequence lengths during training and varying the number of retrieved sentences to explore robustness to configuration changes.
> > > > >
> > > > > Due to space constraints, we have included the ablation studies, which involve training the Mamba retriever on different input sequence lengths in Appendix E.2 Table 24.
> > > > > | **Input Sequence Length**                  |       |       | **Average Accuracy** |
> > > > > |--------------------------------------------|-------|-------|---------------------------------------------------|
> > > > > | **2k tokens**                              | **5k tokens** | **10k tokens** |                |             |             |                     |                      |
> > > > > | 300k data                                  | 0     | 0     | 47.2                |
> > > > > | 86k data                                   | 86k data | 0  | 51.6                |
> > > > > | 35k data                                   | 35k data | 35k data | 56.4                |
> > > > >
> > > > >  Additionally, the appendix contains experimental results for both the Mamba retriever and embedding models when retrieving 50 and 10 sentences (Table 10-14).
> > > > >
> > > > > >Highlight that using fewer training examples (500K vs. 1M) achieved comparable accuracy (i.e., 59.4 vs. 60.0, respectively).
> > > > >
> > > > > Training the Mamba 130M model with only 500K examples can yield results comparable to training with 1M examples (59.4 vs. 60.0). This outcome could be attributed to the limited representational power of the lightweight Mamba-2-130M model or may indicate that the learning capacity from this synthetic data has reached saturation. This suggests that future research should focus on creating more complex and challenging synthetic data.
> > > > >
> > > > > We clarify and highlight this in Line 462-465.
> > > > >
> > > > > >Why not train both the 130M and 1.3B models on a dataset size of 500K examples, but compare using 1M and 400K examples, respectively?
> > > > >
> > > > > Due to budget constraints and the lack of additional long-context training documents, we created only 1 million link-based data points. We limited the training of Mamba-2-1.3B to 400k data points because we did not observe any improvements in the validation sets when training beyond this amount. Mamba-2-130M continues to show some improvements on the validation sets beyond 400k.
> > > > >
> > > > > We explain this in Section 5.5 (Line 288 - 291).
> > > > >
> > > > > >The high cost of generating synthetic training data is mentioned but lacks quantification. How computationally expensive is it in terms of time or resources?
> > > > >
> > > > > We include Table 4 in the paper to show the financial cost of synthetic data generation.
> > > > >
> > > > > | Strategy       | Average Accuracy | Input Token (B) | Output Token (B) | Cost (\$) per 1 Million Examples |
> > > > > |----------------|------------------|-----------------|------------------|-----------------------------------|
> > > > > | Chunk-based    | 57.2             | 0.76           | 0.10            | 71                                |
> > > > > | Pair-based     | 51.4             | 1.49           | 0.37            | 167                               |
> > > > > | Link-based     | 59.4             | 7.79           | 1.64            | 1076                              |
> > > > >
> > > > > >Note that all figures in Appendices B and C are the same, suggesting an error that needs correcting.
> > > > >
> > > > > Thank you for pointing out this mistake. We have updated the correct prompts in the figures.

---

> > > > > > ### Comment · Reviewer_wS7S · 2024-11-22
> > > > > > **Acknowledgment of Revisions**
> > > > > >
> > > > > > Thank you for addressing my concerns and performing the additional experiments. I appreciate the effort you put into improving the manuscript based on my feedback. As promised, I have raised my scores accordingly.

---

> > > > > > > ### Author Response · Authors · 2024-11-23
> > > > > > >
> > > > > > > Dear Reviewer wS7S,
> > > > > > >
> > > > > > > Thank you again for your extremely thorough review. Your suggestions significantly improved the paper’s analytical completeness, particularly regarding the analysis of synthetic data generation, long-context processing, and efficiency considerations. If you have any remaining concerns, we would be happy to address them. If not, we would appreciate if you could consider increasing your score. Thank you for your careful consideration of our work.

---

> ### Author Response · Authors · 2024-12-02
> **Look forward to your reply**
>
> Dear Reviewer wS7S,
>
> Thank you again for your thorough review. As the rebuttal period ends in a few hours, we wanted to confirm that our revised manuscript has adequately addressed all your concerns. If your concerns have been addressed, we respectfully request that you increase your scores.

---

### Official Review · Reviewer_PSm7 · 2024-10-22

**Soundness:** 2
**Presentation:** 3
**Contribution:** 2
**Rating:** 6
**Confidence:** 4

**Summary:**

### Summary
1. A Mamba architecture based retriever is presented which is trained as a discriminator model
2. The retrievers are much smaller (130M and 1.3B) and outperform much larger open source (and non-finetuned) embedding based models
3. Efficacy of all retrievers is tested by plugging the retrieved chunks (or sentences) into a RAG pipeline


### Contributions
1. The paper showcases how a Mamba architecture model could be finetuned to create an effective retriever for long context documents
2. Methodology and ablation for creating synthetic data is well studied and supported via ablation
3. Synthetic dataset would further research for long context document understanding (and retrieval)

**Strengths:**

1. This work presents a lightweight (small parameter) based retriever for long context documents
2. Complete training code and dataset is available on Github for reproducibility
3. New dataset would further research in the direction
4. Ablations presented help understand the impact of synthetic data generation process (link based) as well as effect of context size on RAG system performance

**Weaknesses:**

1. The baseline retriever (open source embedding model) is weak as it has not seen any synthetic data. It is unclear whether gains seen by Mamba retriever is due to synthetic data or the model's ability to handle long context
2. The paper claims that Mamba is more efficient in handling long document context. No supportive results are given in terms of time taken. It's unclear how efficient the model is when compared to a retriever
3. Embedding based retriever has a different retrieval granularity (chunks) as compared to Mamba (Top 50 sentences)
4. The paper claims there is lack of synthetic data for long context documents, but ends up creating a dataset where maximum context length is 10k. Is my understanding correct?
5. (Minor issue) No qualitative examples are presented in the paper which highlight how Mamba retrieves more relevant context as compared to Embedding based models. Perhaps some examples can be added to the Appendix, if there

**Questions:**

1. (Lines 233 to 239) Why is the retriever setup different for Mamba and embedding based retrievers? Mamba fetches Top 50 sentences whereas embedding based retrievers fetch 5 chunks. How do embedding models perform when you have identical setup?
2. Why was the link based synthetic data not used to finetune an embedding model and compared as baseline? Perhaps the gains seen by Mamba retriever are due to synthetic data. A quick check would be to see how embedding retrievers perform (in terms of recall) on a held out set of the synthetic link based data. Maybe this number could be improved via finetuning and the finetuned embedding model tested as baseline
3. (Lines 198-200) What decontamination pipeline was used to ensure that there is no overlap between synthetic data documents and documents used in the 41 benchmarks. Did you look for exact match of document content? Are the domains completely different?
4. (Lines 230-232) How did you evaluate the quality of an answer using GPT-4o? Did you give some instructions in prompt on how to judge the quality of an answer? Did the model return a binary score (such as relevant or non-relevant)
5. How was BM25 setup? Did it also retriever chunks? What happens when it retrieves sentences (same as Mamba)
6. (Lines 204-207) Which hyper-parameters were optimized? How did you select the best checkpoint? How did you decide when to stop training?
7. (Minor comment): Figures 6-10 in the Appendix seem to have the same exact prompt. Was this intentional or done by mistake?
8. (Minor comment): Lines 89-90 say upto 256 k. Did the authors mean more than 256k?

---

> ### Author Response · Authors · 2024-11-22
> **Author Response with a Revised Paper**
>
> >The baseline retriever (open source embedding model) is weak as it has not seen any synthetic data. It is unclear whether gains seen by Mamba retriever is due to synthetic data or the model's ability to handle long context
>
> >Why was the link based synthetic data not used to finetune an embedding model and compared as baseline? Perhaps the gains seen by Mamba retriever are due to synthetic data. A quick check would be to see how embedding retrievers perform (in terms of recall) on a held out set of the synthetic link based data. Maybe this number could be improved via finetuning and the finetuned embedding model tested as baseline
>
> Based on your suggestion, we have performed these baseline experiments.
>
> We fine-tuned two embedding models: Contriever-110M, a robust retriever used in standard RAG pipelines [(Xu et al., 2023)](https://arxiv.org/pdf/2310.03025), and GTE-Qwen2-1.5B, the second best 1.5B parameter model on the massive text embedding benchmark (MTEB). These two are selected because of their open-sourced training details. These models are comparable in size to the Mamba retriever 130M and 1.3B respectively. We fine-tuned them using the same 1 million link-based synthetic data for one epoch. For each query, relevant sentences were treated as positives, while irrelevant ones were treated as negatives. We employed the same contrastive loss and applied the same hyperparameter settings (e.g., scheduler, optimizer, and temperature $\tau$ in the InfoNCE loss) as reported in their original papers. Additionally, we optimized the learning rates, batch size, and training data size using the same validation sets.
> | Retriever         | Educational | Creative | Official | Conversational | Average |
> |-------------------|-------------|----------|----------|----------------|---------|
> | Contriever-110M   | 66.3        | 45.8     | 52.9     | 45.0           | 54.3    |
> | Contriever-110M-FT| 65.5        | 48.0     | 55.5     | 41.2           | 54.8    |
> | GTE-Qwen2-1.5B    | 67.2        | 47.7     | 56.2     | 44.3           | 55.7    |
> | GTE-Qwen2-1.5B-FT | 66.9        | 48.0     | 56.2     | 44.8           | 55.8    |
>
> We observed no noticeable improvement after fine-tuning on the synthetic data (the data are also included in Table 1 of the revised paper). This suggests that the gains achieved by our Mamba retrievers are not merely artifacts of the training documents. Instead, the Mamba retriever has learned to synthesize information from different parts of the document, which aligns with the intention of our link-based synthetic training data.
>
> We have incorporated these discussions into the paper (Line 295-300, Table 1, Line 406-409)

---

> > ### Author Response · Authors · 2024-11-22
> > **Author Response Part 2**
> >
> > >The paper claims that Mamba is more efficient in handling long document context [...] how efficient the model is when compared to a retriever
> >
> > We compare the Mamba retriever with three state-of-the-art embedding models, focusing on accuracy on test sets, processing speed per document, FLOPs per document, and model size. Embedding models were evaluated in two settings: sentence retrieval and chunk retrieval (chunk size was 300 words). Performance differs in these two settings due to increased padding required for sentence retrieval. All embedding models were evaluated using their official repositories.
> >
> > | Model                  | Retrieval Setting | Speed (ms) | TFLOPS     | TFLOPS w/o Pad | Params (billions) | Average Accuracy |
> > |------------------------|-------------------|------------|------------|----------------|-------------------|------------------|
> > | Mamba retriever 130M   | 50 sents          | **93.4**   | **19.0**   | **19.0**       | **0.1**           | 60.0             |
> > | Mamba retriever 1.3B   | 50 sents          | 181.6      | 197.9      | 197.9          | 1.3               | **61.8**         |
> > | NV-Embed-v2-7B         | 50 sents          | 592.0      | 1316.7     | 1279.4         | 7.9               | 56.6             |
> > | NV-Embed-v2-7B         | 5 chunks          | 470.8      | 1295.6     | 1287.5         | 7.9               | 59.1             |
> > | Stella-1.5B            | 50 sents          | 364.7      | 331.9      | 210.5          | 1.5               | 55.6             |
> > | Stella-1.5B            | 5 chunks          | 264.8      | 244.8      | 219.0          | 1.5               | 56.8             |
> > | GTE-Qwen2-1.5B         | 50 sents          | 364.4      | 331.9      | 210.5          | 1.5               | 54.6             |
> > | GTE-Qwen2-1.5B         | 5 chunks          | 264.9      | 244.8      | 219.0          | 1.5               | 55.7             |
> > | Llama-3.1-70B          | Direct Answer     | N/A        | 28,517.9   | 28,517.9       | 69.5              | 57.8             |
> >
> > Mamba retrievers achieve better speed and accuracy than the embedding models. The most accurate embedding model is the 7B NV-embed, which uses many more FLOPs than the 1.3B or 130M Mamba retriever. The 130M Mamba retriever achieves higher accuracy than any of the embedding models, while using a fraction of the FLOPs and achieving much faster document processing time.
> >
> > We have incorporated these changes into the paper (Section 5.6, line 373-377, Table 2)
> >
> > >Embedding based retriever has a different retrieval granularity (chunks) as compared to Mamba [...] How do embedding models perform when you have identical setup?
> >
> > We chose the chunk retrieval for the embedding models based on preliminary comparisons with the 50-sentence retrieval. We have now done a full comparison, shown below, which demonstrates that 5-chunk retrieval always performs better than 50-sentence retrieval across the embedding models. A potential explanation is that the chunk strategy on average retrieves 2000 tokens vs. 1600 tokens for the sentence strategy. These settings are now compared and discussed in the paper (Line 254-261, Line 322-323, Line 347, Table 2). All embedding models' performance on “50 sentences” and “5 chunks” are reported in Appendix A.4 Table 10-14.
> >
> > |                     Embedding Model                     | Retrieval Setting | Accuracy (%) |
> > |---------------------------------------------------------|-------------------|--------------|
> > | **BM25**                                                | 50 sents          | 44.6         |
> > |                                                         | 5 chunks          | 49.1         |
> > | **Dragon-110M**                                         | 50 sents          | 52.8         |
> > |                                                         | 5 chunks          | 53.9         |
> > | **Contriever-110M**                                     | 50 sents          | 53.1         |
> > |                                                         | 5 chunks          | 54.3         |
> > | **GTE-Qwen2-1.5B**                                      | 50 sents          | 54.6         |
> > |                                                         | 5 chunks          | 55.7         |
> > | **Stella-1.5B**                                         | 50 sents          | 55.6         |
> > |                                                         | 5 chunks          | 56.8         |
> > | **OpenAI v3-large**                                     | 50 sents          | 55.4         |
> > |                                                         | 5 chunks          | 57.6         |
> > | **GritLM-7B**                                           | 50 sents          | 56.7         |
> > |                                                         | 5 chunks          | 57.2         |
> > | **NV-Embed-v2-7B**                                      | 50 sents          | 56.6         |
> > |                                                         | 5 chunks          | 59.1         |

---

> > > ### Author Response · Authors · 2024-11-22
> > > **Author Response Part 3**
> > >
> > > >The paper claims there is lack of synthetic data for long context documents, but ends up creating a dataset where maximum context length is 10k. Is my understanding correct?
> > >
> > > We have clarified this point in the revised paper (Line 151-156). We meant to convey that the specific type of synthetic data needed to train our Mamba retriever is not available, as it requires sentence-level annotations.
> > >
> > > >No qualitative examples are presented in the paper which highlight how Mamba retrieves more relevant context as compared to Embedding based models. Perhaps some examples can be added to the Appendix, if there
> > >
> > > In Appendix E.4 Table 25, 26, 27, 28, we have highlighted various examples where the current leader of MTEB, NV-Embed-v2 has failed, in both the “50 sentences” and “5 chunks” settings, to retrieve relevant sentences or chunks because it was not able to reason contextually, where Mamba retriever 1.3B retrieved accurately.
> > >
> > > >What decontamination pipeline was used to ensure that there is no overlap between synthetic data documents and documents used in the 41 benchmarks. Did you look for exact match of document content? Are the domains completely different?
> > >
> > > The documents in the training set were collected from novels in Project Gutenberg, government reports [(Huang et al 2021)](https://aclanthology.org/2021.naacl-main.112/), public-domain financial statements published on the US Securities and Exchange Commission website, and legal contracts from [Hendrycks et al’s](https://arxiv.org/abs/2103.06268) (Section 4.2 Line 210-215).
> > >
> > > All of the test sets are from the benchmarks, which consist of 41 datasets (Section 5.1 Line 228-232). Detailed statistics for each of these datasets are provided in Appendix A.2.
> > >
> > > At a high level, the decontamination procedure removed any training documents that had more than 1% sentence overlap with the test set. More precisely, we performed sentence tokenization on all documents from our 41 test sets, resulting in a pool of 2.4 million test sentences. Next, we performed sentence tokenization on each training document, and used string matching to calculate the overlap with the pool of test sentences. We removed any training document where more than 1% of its sentences matched those in the test pool of 2.4 million sentences.
> > >
> > > The longest training document contains fewer than 1,000 sentences. Our decontamination procedure ensures that fewer than 10 sentences in this document are included in the test pool. This guarantees there is no meaningful textual overlap between the test and training sets.
> > >
> > > We have elaborated on our decontamination procedure in Section 4.3 of the revised paper.
> > >
> > > >How did you evaluate the quality of an answer using GPT-4o? Did you give some instructions in prompt on how to judge the quality of an answer? Did the model return a binary score (such as relevant or non-relevant)
> > >
> > > The accuracy of freeform answers is assessed using GPT-4o-0806, which is prompted to compare attempted answers with ground-truth answers, resulting in a binary "yes" or "no" judgment. This prompt is developed from 100 human-annotated examples. On a separate held-out test set of 180 human-annotated examples, GPT-4o's 180 yes/no judgments show high agreement with human judgments, achieving a macro F1 score of 0.942.
> > >
> > > For the prompt, refer to Appendix C. The method is now described in Section 5.3, titled "GPT-4o as Judge."
> > >
> > > >How was BM25 setup? Did it also retriever chunks? What happens when it retrieves sentences (same as Mamba)
> > >
> > > BM25 was set up to retrieve chunks. Retrieving sentences with BM25 achieves an accuracy of only 44.6%, while retrieving chunks reaches an accuracy of 49.1%. The results for BM25 retrieval of both chunks and sentences are provided in Appendix A.4 Table 10, Line 1050-1051 and Line 1063-1064.
> > >
> > > >Which hyper-parameters were optimized? How did you select the best checkpoint? How did you decide when to stop training?
> > >
> > > The only hyperparameters optimized on the validation sets were the learning rates and the synthetic data generation strategy. We used a cosine annealing scheduler, and wanted to fully anneal the learning rate, so we did not implement early stopping.
> > > Details about the validation set can be found in Appendix A.3 and A.1 Line 888-893. The 130M model was fine-tuned on 1 million link-based synthetic data points, while the 1.3B model was fine-tuned on 400,000 data points. Both models were trained for one epoch.
> > >
> > > Due to budget constraints and the lack of additional long-context training documents, we only created 1 million link-based data points. We limited the training of Mamba-2-1.3B to 400,000 data points because we did not observe any improvements in the validation sets when training beyond this amount.
> > >
> > > This clarification is included in the updated paper (Section 5.5, Line 286-294).
> > >
> > > Appendix D reports all hyperparameters used.

---

> > > > ### Author Response · Authors · 2024-11-22
> > > > **Author Response Part 4**
> > > >
> > > > >Figures 6-10 in the Appendix seem to have the same exact prompt. Was this intentional or done by mistake?
> > > >
> > > > Thank you for pointing out this mistake. We have updated it with the correct prompts.
> > > >
> > > > >Lines 89-90 say upto 256 k. Did the authors mean more than 256k?
> > > >
> > > > Our Mamba retriever can generalize beyond its training context length of 10k tokens (see Section 7.2 and Figure 4). For instance, the Mamba retriever 1.3B can handle up to 256k tokens without memory errors on a single node with 8 * 80GB H100. To ensure a fair comparison with GPT-4o, which has a context window of 128k, we use a sliding window approach for Mamba retrievers when documents exceed 120k tokens. This allows both models to operate within the same effective context window. Sentences scored twice have their scores (i.e., logit values) averaged.
> > > >
> > > > This is now described in Section 5.4 Line 276-281, titled “Sliding Window” of the revised paper.

---

> > > > > ### Author Response · Authors · 2024-11-23
> > > > > **Looking forward to your reply**
> > > > >
> > > > > Dear Reviewer PSm7,
> > > > >
> > > > > Thank you for your insightful feedback. We have conducted extensive additional experiments to address your concerns. We would greatly appreciate your timely review of our revisions, given the approaching rebuttal deadline. All changes are highlighted in red in the revised paper.

---

> > > > > > ### Author Response · Authors · 2024-11-28
> > > > > > **Look forward to your reply**
> > > > > >
> > > > > > Dear Reviewer PSm7,
> > > > > >
> > > > > > Given the approaching rebuttal deadline, we wanted to ensure you had the opportunity to review our revised manuscript, where we have addressed your valuable feedback with additional experiments and marked all changes in red. We look forward to your thoughts on these revisions.

---

> > > > > > > ### Author Response · Authors · 2024-12-02
> > > > > > > **Look forward to your reply**
> > > > > > >
> > > > > > > Dear Reviewer PSm7,
> > > > > > >
> > > > > > > As the rebuttal deadline approaches, we wanted to follow up to ensure you had the opportunity to review our revisions. We have addressed your insightful feedback, conducted additional experiments, and marked all changes in red in the revised manuscript. We look forward to your thoughts on these revisions.

---

> ### Author Response · Authors · 2024-12-02
> **Urgent Feedback Requested**
>
> Dear Reviewer PSm7,
>
> As the rebuttal period ends in a few hours, we urgently request your feedback on our manuscript. If your concerns have been addressed, we respectfully request that you significantly increase your scores.

---

### Official Review · Reviewer_4h6p · 2024-10-28

**Soundness:** 2
**Presentation:** 2
**Contribution:** 2
**Rating:** 5
**Confidence:** 4

**Summary:**

The paper presents a retrieval model based on the Mamba architecture to improve long document understanding. By fine-tuning a Mamba checkpoint, the authors aim to handle entire document contexts without chunking, thus enhancing retrieval accuracy. The paper highlights synthetic data generation for training the model, including three methods—chunk-based, pair-based, and link-based. Experimental results indicate that the Mamba retriever performs comparably with existing large language models in various retrieval tasks while being more efficient. The model is evaluated on 41 QA benchmarks across multiple domains, demonstrating that the full-context Mamba retriever outperforms embedding-based retrievers, especially for long documents.

**Strengths:**

1. The paper introduces a novel adaptation of the Mamba architecture, focusing on processing entire documents without chunking, which may provide a more holistic context.
2. The paper addresses the scarcity of long-context retrieval data by developing synthetic data generation methods, specifically the effective link-based generation, which strengthens the retrieval model’s generalization.

**Weaknesses:**

1. The paper’s readability suffers due to numerous grammatical issues, ambiguous statements, and poorly constructed sentences. A thorough rewrite would significantly improve comprehension and presentation.
2. Many sections make assertions without proper references, undermining the credibility of the stated findings and comparisons.
3. The proposed method essentially fine-tunes Mamba for retrieval tasks without introducing substantial architectural innovation or unique methodological contributions.
4. The authors categorize datasets into four types but fail to clarify the rationale or methodology behind this categorization.
5. Given the method’s focus on long-context retrieval, comparisons with state-of-the-art long LLMs (such as Llama 3.1, ChatGLM, etc.) and leading retrievers from the mteb leaderboard would provide a more competitive and comprehensive evaluation.
6. Efficiency is critical in retrieval tasks, yet the paper omits a direct analysis of retrieval speed and computational resources, which would provide essential context on the practical applicability of the Mamba retriever.

**Questions:**

1. What criteria were used to categorize datasets into four types, and how was this categorization verified or validated?
2. How does the efficiency of the Mamba retriever compare to traditional embedding-based models in terms of speed, memory usage, and computational cost?

---

> ### Author Response · Authors · 2024-11-22
> **Author Response with a Revised Paper**
>
> >The paper’s readability suffers due to numerous grammatical issues, ambiguous statements, and poorly constructed sentences. A thorough rewrite would significantly improve comprehension and presentation.
>
> We have substantially revised the paper to enhance its readability. All important changes are highlighted in red in the updated version.
>
> >Many sections make assertions without proper references, undermining the credibility of the stated findings and comparisons.
>
> We have included the previously missing references in the updated version of the paper.
>
> >The proposed method essentially fine-tunes Mamba for retrieval tasks without introducing substantial architectural innovation or unique methodological contributions.
>
> We respectfully disagree. The approach required solving two key challenges: finding the right model adaptation and generating effective training data.
>
> Obvious model adaptations fail: training Mamba for generative retrieval achieves only 33.5% accuracy, while direct question answering achieves only 27.6% (Tables 5 and 6). Additionally, straightforward synthetic data generation strategies like chunk-based (using individual document segments) and pair-based (matching similar chunks) produce questions that don't require understanding document-wide connections and lead to worse performance (Table 4 and Line 452-461).
>
> Our key insight was combining a discriminative retriever architecture with a link-based training strategy that discovers real connections within documents. The effectiveness of this specific combination is demonstrated by: (1) the model outperforms embedding models 8-60x larger while using fewer computational resources (Table 1 and 2), (2) fine-tuning these embedding models on our synthetic data shows no improvement (Line 405-409), and (3) we observe strong performance on documents up to length 256k, which is more than 20x longer than any documents observed during training (Line 400-404).
>
> >The authors categorize datasets into four types but fail to clarify the rationale or methodology behind this categorization.
>
> Because the method is evaluated on 41 benchmark datasets, it was not possible to include information about all of these datasets in the main text. Results on each individual dataset can be found in Appendix A.4 Table 10,11,12,13,14.
>
> The test data were divided into four categories for expository purposes only. We clarify our rationale in our revised paper (Line 232-242). Furthermore, details of the datasets and their categorization can be found in Appendix A.1 Table 7 and Appendix A.2 Table 8.
>
> >Given the method’s focus on long-context retrieval, comparisons with state-of-the-art long LLMs (such as Llama 3.1, ChatGLM, etc.)
>
> We explore the performance of LLMs in two settings. In the first setting (generative retrievers), the LLM is given a question and document and asked to generate relevant passages from the document. Given a complete document, GPT-4o and Llama-3.1-70B are asked to retrieve relevant sentences for up to 2,000 tokens, which provides a fair comparison with the "50 sentences" setup used in Mamba retrievers (Section 7.3). In the second setting (direct answering), the LLM is given the document and asked to directly answer the question (Section 7.4). We also fine-tuned Mamba-2 models for direct question answering on the same 1 million synthetic data.
>
> ### Generative Retriever
> | Retriever Type  | Generative                     |                  |                  | Discriminative               |                  |
> |------------------|--------------------------------|------------------|------------------|--------------------------------|------------------|
> | **Model**           | GPT-4o                         | Llama-3.1 70B    | Mamba-2 130M-FT | Mamba retriever 130M          | Mamba retriever 1.3B |
> |**Average Accuracy** | 52.2                          | 45.9             | 33.5             | 60.0                           | 61.8             |
>
> ### Direct answer generation from full context.
> | **Model**       | GPT-4o | Llama-3.1 70B | Llama-3.1 8B | Mamba-2 130M-FT | Mamba-2 130M | Mamba-2 1.3B-FT | Mamba-2 1.3B |
> |------------------|--------|---------------|--------------|-----------------|--------------|-----------------|--------------|
> | **Average Accuracy**      | 64.6   | 57.8          | 49.1         | 15.6           | 0.56         | 27.6           | 0.59         |
>
> The above two tables are included as Table 5,6 in the revised paper.
> All generative retrievers perform significantly worse than discriminative Mamba retrievers. In the direct answering setting, only GPT-4o performs better than the discriminative Mamba retrievers. This suggests that retrieval by LLMs is often lossy in long-context settings, further highlighting the advantages of using discriminative retrieval with our method.

---

> > ### Author Response · Authors · 2024-11-22
> > **Author Response Part 2**
> >
> > >and leading retrievers from the mteb leaderboard would provide a more competitive and comprehensive evaluation.
> >
> > Thank you. We have included the current leader on MTEB (NV-Embed-v2) and another top 5 contender (Stella-1.5B). We also included GTE-Qwen2-1.5B which is the second best 1.5B parameter embedding model on MTEB. They all consistently underperform our Mamba retrievers despite larger sizes (Table 1).
> >
> > >Efficiency is critical in retrieval tasks, yet the paper omits a direct analysis of retrieval speed and computational resources, which would provide essential context on the practical applicability of the Mamba retriever.
> >
> > Thank you for your important suggestion. Our method is specifically designed for scenarios where a document is introduced for the first time and then queried. We compare the document processing speed of Mamba retriever with top embedding models on MTEB (including NV-Embed-v2, Stella, and Qwen2). Embedding models were evaluated in two settings: sentence retrieval and chunk retrieval (chunk size was 300 words). Performance differs in these two settings due to increased padding required for sentence retrieval. All embedding models were evaluated using their official repositories (See Section 5.6 and Line 373-377).
> >
> > | Model                  | Retrieval Setting | Speed (ms) | TFLOPS     | TFLOPS w/o Pad | Params (billions) | Average Accuracy |
> > |------------------------|-------------------|------------|------------|----------------|-------------------|------------------|
> > | Mamba retriever 130M   | 50 sents          | **93.4**   | **19.0**   | **19.0**       | **0.1**           | 60.0             |
> > | Mamba retriever 1.3B   | 50 sents          | 181.6      | 197.9      | 197.9          | 1.3               | **61.8**         |
> > | NV-Embed-v2-7B         | 50 sents          | 592.0      | 1316.7     | 1279.4         | 7.9               | 56.6             |
> > | NV-Embed-v2-7B         | 5 chunks          | 470.8      | 1295.6     | 1287.5         | 7.9               | 59.1             |
> > | Stella-1.5B            | 50 sents          | 364.7      | 331.9      | 210.5          | 1.5               | 55.6             |
> > | Stella-1.5B            | 5 chunks          | 264.8      | 244.8      | 219.0          | 1.5               | 56.8             |
> > | GTE-Qwen2-1.5B         | 50 sents          | 364.4      | 331.9      | 210.5          | 1.5               | 54.6             |
> > | GTE-Qwen2-1.5B         | 5 chunks          | 264.9      | 244.8      | 219.0          | 1.5               | 55.7             |
> > | Llama-3.1-70B          | Direct Answer     | N/A        | 28,517.9   | 28,517.9       | 69.5              | 57.8             |
> >
> > Mamba retrievers achieve better speed and accuracy than the embedding models. The most accurate embedding model is the 7B NV-embed, which uses many more FLOPs than the 1.3B or 130M Mamba retriever. The 130M Mamba retriever achieves higher accuracy than any of the embedding models, while using a fraction of the FLOPs and achieving much faster document processing time.
> >
> > The above table is included in the revised paper as Table 2.

---

> > > ### Comment · Reviewer_4h6p · 2024-11-23
> > > **Response to Authors**
> > >
> > > Thanks for the authors' reply, which resolve most of my concerns. I will update my assessment accordingly.

---

> > > > ### Author Response · Authors · 2024-11-23
> > > >
> > > > Dear Reviewer 4h6p,
> > > >
> > > > Thank you for your thorough review and for acknowledging that our responses resolved most of your concerns. If you have any remaining concerns, we would be happy to address them. If not, we would greatly appreciate if you could consider increasing your score. Thank you for your careful consideration of our work.

---

> > > > > ### Comment · Reviewer_4h6p · 2024-11-23
> > > > > **Response to authors**
> > > > >
> > > > > Thank you to the authors for their quickly response. The previous replies address most of my concerns regarding the experiments and efficiency. However, I maintain my reservations about the novelty of the method and the quality of its presentation.
> > > > >
> > > > > First, the authors claim that their contributions lie in “finding the right model adaptation and generating effective training data.” But this does not appear to represent a clear research direction but instead resembles engineering work, where various models are combined with different datasets to achieve good performance.
> > > > >
> > > > > Second, Section 3 introduces the method used in this paper, but the explanation is ambiguous. In my opinion, it would be more effective to clarify your approach using equations and formulations, explicitly defining the input, output, and the computation of the loss function. The current presentation looks more like a technical report more than a research paper.
> > > > >
> > > > > Additionally, I conducted a quick search and found a highly relevant paper, “Mamba Retriever: Utilizing Mamba for Effective and Efficient Dense Retrieval,” released this August, which is not cited in this work. The two approaches appear to share many similarities, and I encourage the authors to discuss the differences between their work and the Mamba Retriever.
> > > > >
> > > > > My assessment may currently appear higher than my actual opinion of the paper. The additional points are primarily due to the comprehensive experiments and the authors’ engaging and constructive attitude during the rebuttal stage.

---

> > > > > > ### Author Response · Authors · 2024-11-24
> > > > > > **Looking forward to your reply**
> > > > > >
> > > > > > Thank you for your response. We would like to clarify your concerns.
> > > > > >
> > > > > > > First, the authors claim that their contributions lie in “finding the right model adaptation and generating effective training data.” But this does not appear to represent a clear research direction but instead resembles engineering work, where various models are combined with different datasets to achieve good performance.
> > > > > >
> > > > > > This work makes research contributions beyond engineering. The key challenge in long-document retrieval is teaching models to use relevant information spread across a document. While existing methods rely on chunking and local embeddings, we show how to train models that effectively reason about global document context.
> > > > > > We investigate the hypothesis that natural document connections can teach this reasoning ability. We develop a method to discover these connections using LLMs and transform them into training examples. **This builds on but differs from recent work in synthetic query generation [1-8], which focuses on matching local textual patterns rather than document-wide relationships.** Extensive ablations demonstrate that simpler approaches (chunk-based, pair-based generation, or generative retrieval) fail to capture these long-range dependencies.
> > > > > > The empirical results support the hypothesis - a 130M parameter model outperforms 7B parameter embedding models while using fewer computational resources, and approaches GPT-4's performance on very long documents. This demonstrates that with the right training signal, small models can learn to effectively reason across long documents.
> > > > > >
> > > > > > 1. Dai et al, [Promptagator: Few-shot Dense Retrieval From 8 Examples](https://openreview.net/forum?id=gmL46YMpu2J), ICLR 2023
> > > > > >
> > > > > > 2. Wang et al, [IR2: Information Regularization for Information Retrieval](https://aclanthology.org/2024.lrec-main.810/), LREC-COLING 2024
> > > > > >
> > > > > > 3. Ferraretto et al, [ExaRanker: Synthetic Explanations Improve Neural Rankers](https://dl.acm.org/doi/abs/10.1145/3539618.3592067), SIGIR 2023
> > > > > >
> > > > > > 4. Bonifacio et al, [InPars: Unsupervised Dataset Generation for Information Retrieval](https://dl.acm.org/doi/10.1145/3477495.3531863), SIGIR 2022
> > > > > >
> > > > > > 5. Penha et al, [Improving Content Retrievability in Search with Controllable Query Generation](https://dl.acm.org/doi/abs/10.1145/3543507.3583261), ACM WWW 2023
> > > > > >
> > > > > > 6. Chaudhary et al, [Exploring the Viability of Synthetic Query Generation for Relevance Prediction](https://arxiv.org/abs/2305.11944), SIGIR Ecom 2023
> > > > > >
> > > > > > 7. Ment et al, [AugTriever: Unsupervised Dense Retrieval and Domain Adaptation by Scalable Data Augmentation](https://arxiv.org/abs/2212.08841), DCAI 2024
> > > > > >
> > > > > > 8. Jeronymo et al, [InPars-v2: Large Language Models as Efficient Dataset Generators for Information Retrieval](https://arxiv.org/abs/2301.01820)
> > > > > >
> > > > > > > Additionally, I conducted a quick search and found a highly relevant paper, “Mamba Retriever: Utilizing Mamba for Effective and Efficient Dense Retrieval,” released this August, which is not cited in this work. The two approaches appear to share many similarities, and I encourage the authors to discuss the differences between their work and the Mamba Retriever.
> > > > > >
> > > > > > Thank you for bringing this important paper to our attention. While both works explore using Mamba for retrieval, they take complementary approaches. The referenced paper develops Mamba as an embedding model that generates vector representations for queries and documents, building on other recent work on SSM embeddings (Lines 77-79). Our work takes a different approach by developing a discriminative retriever that processes entire documents to make context-aware retrieval decisions. We have updated our Related Work section to include this and other recent works on SSM embeddings (Lines 76-79). As of the ICLR submission deadline in October, there were no published works using SSMs as discriminative retrievers with global context.
> > > > > >
> > > > > > We will add a precise mathematical description of the model in response to your suggestion.

---

> > > > > > > ### Author Response · Authors · 2024-11-28
> > > > > > > **Mathematical Formulation Added**
> > > > > > >
> > > > > > > Dear Reviewer 4h6p,
> > > > > > >
> > > > > > > Thank you for your valuable suggestion regarding formal mathematical formulations. We have addressed this by adding detailed mathematical formulations in Appendix F due to page limit and included a reference to it at the end of Section 3 (line 147). We believe this addition strengthens the paper's technical rigor. We appreciate your feedback and hope this revision meets your expectations. We look forward to your reply.

---

> ### Author Response · Authors · 2024-12-02
> **Look forward to your reply**
>
> Dear Reviewer 4h6p,
>
> Thank you once again for your insightful suggestions throughout this process. As the rebuttal ends in a few hours, we respectfully request your prompt feedback on our revised manuscript. Your timely response is critical to ensuring we address all concerns effectively.

---

### Official Review · Reviewer_f8dJ · 2024-11-04

**Soundness:** 3
**Presentation:** 3
**Contribution:** 2
**Rating:** 6
**Confidence:** 3

**Summary:**

This paper proposes to find-tune a long-context Mamba model for the retrieval task, claiming that such a method is better than embedding-based retriever because the awareness of context. To train such a model, a data synthesis method is proposed, which is to generate query given two semantically relevant text chunks. In the experiments, the proposed method is compared with long-context model and standard RAG methods and shows strong performance.

**Strengths:**

- The paper studies an idea of fine-tuning long-context model as retriever. To gain efficiency for training, the Mamba model is used to save cost for training a Transformer model
- The performance shows the long-context discriminative model as retriever brings improvement to the retrieval performance.

**Weaknesses:**

- The efficiency of proposed method is not studied. I can expect that with pre-built vector index, the standard RAG may be more efficient that the proposed method which needs to pass the entire textual data for each query. Although adding context for retrieval brings improvement, I am not sure whether it worth the cost if its non-trivial.
- I am not clear with the data split. What data is used for data synthesis and training and what for testing? The paper only mentions the test set so I am not sure if it is used for both.
- It is not clear to me what if the long-context model is directly trained to be a reader, i.e. generate the answer given a query. The data synthesis part can generate a query so can generate a corresponding answer as well. This should be a baseline and shows the necessity of training the model as a retrieval followed by another reader model.

**Questions:**

See the weaknesses raised above

---

> ### Author Response · Authors · 2024-11-22
> **Author Response with a Revised Paper**
>
> >The efficiency of proposed method is not studied. I can expect that with pre-built vector index, the standard RAG may be more efficient that the proposed method which needs to pass the entire textual data for each query. Although adding context for retrieval brings improvement, I am not sure whether it worth the cost if its non-trivial.
>
> Thank you for your feedback. Our method is specifically designed for scenarios where a document is introduced for the first time and then queried. We compare the Mamba retriever with three state-of-the-art embedding models, evaluating their performance in processing documents. Embedding models were evaluated in two settings: sentence retrieval and chunk retrieval (chunk size was 300 words). Speed and TFLOPS differ in these two settings due to the increased padding required for sentence retrieval. All embedding models were evaluated using their official repositories.
> | Model                  | Retrieval Setting | Speed (ms) | TFLOPS     | TFLOPS w/o Pad | Params (billions) | Average Accuracy |
> |------------------------|-------------------|------------|------------|----------------|-------------------|------------------|
> | Mamba retriever 130M   | 50 sents          | **93.4**   | **19.0**   | **19.0**       | **0.1**           | 60.0             |
> | Mamba retriever 1.3B   | 50 sents          | 181.6      | 197.9      | 197.9          | 1.3               | **61.8**         |
> | NV-Embed-v2-7B         | 50 sents          | 592.0      | 1316.7     | 1279.4         | 7.9               | 56.6             |
> | NV-Embed-v2-7B         | 5 chunks          | 470.8      | 1295.6     | 1287.5         | 7.9               | 59.1             |
> | Stella-1.5B            | 50 sents          | 364.7      | 331.9      | 210.5          | 1.5               | 55.6             |
> | Stella-1.5B            | 5 chunks          | 264.8      | 244.8      | 219.0          | 1.5               | 56.8             |
> | GTE-Qwen2-1.5B         | 50 sents          | 364.4      | 331.9      | 210.5          | 1.5               | 54.6             |
> | GTE-Qwen2-1.5B         | 5 chunks          | 264.9      | 244.8      | 219.0          | 1.5               | 55.7             |
> | Llama-3.1-70B          | Direct Answer     | N/A        | 28,517.9   | 28,517.9       | 69.5              | 57.8             |
>
> Mamba retrievers achieve better speed and accuracy than the embedding models. The most accurate embedding model is the 7B NV-embed, which uses many more FLOPs than the 1.3B or 130M Mamba retriever. The 130M Mamba retriever achieves higher accuracy than any of the embedding models while using a fraction of the FLOPs and achieving much faster document processing time.
> We have reflected these changes in Section 5.6, Table 2, and Line 373-377 in the revised paper.
>
> >I am not clear with the data split. What data is used for data synthesis and training and what for testing? The paper only mentions the test set so I am not sure if it is used for both.
>
> The documents in the training set were collected from novels in Project Gutenberg, government reports [(Huang et al 2021)](https://aclanthology.org/2021.naacl-main.112/), public-domain financial statements published on the US Securities and Exchange Commission website, and legal contracts from [Hendrycks et al’s](https://arxiv.org/abs/2103.06268) (Section 4.2 line 211-215). These documents were used to generate synthetic questions as described in the paper.
> The training sets for the 41 benchmark datasets were not used for training the model. Models were evaluated on the test sets for the 41 benchmark datasets. Detailed statistics for each of these datasets are provided in Appendix A.2 Table 8.
> A decontamination pipeline was applied in order to ensure that the training documents did not accidentally overlap with the testing documents. The pipeline removed any training document that had more than 1% overlap (at the sentence level) with the test set. The procedure is described in Sections 4.3 Line 218-223.

---

> > ### Author Response · Authors · 2024-11-22
> > **Author Response Part 2**
> >
> > >It is not clear to me what if the long-context model is directly trained to be a reader, i.e. generate the answer given a query. The data synthesis part can generate a query so can generate a corresponding answer as well. This should be a baseline and shows the necessity of training the model as a retrieval followed by another reader model.
> >
> > Thank you for suggesting an interesting baseline. We trained Mamba-2-1.3b and Mamba-2-130m models to directly generate these answers as suggested. The models were trained on the same 1 million link-based synthetic data points, for one epoch using cross-entropy loss. Here are the results:
> > | Model         | Mamba-2 130M-FT | Mamba-2 130M | Mamba-2 1.3B-FT | Mamba-2 1.3B |
> > |---------------|-----------------|--------------|-----------------|--------------|
> > | Average Accuracy         | 15.6            | 0.56         | 27.6            | 0.59         |
> >
> > The performance for both model sizes is substantially lower than for Mamba retrievers, which achieve over 60% accuracy. This demonstrates the utility of Mamba retrievers, and the non-trivial nature of  Mamba retrievers’ strong performance.
> > The results are included in Section 7.4 Table 6 in the revised paper with relevant information highlighted in red.

---

> > > ### Author Response · Authors · 2024-11-23
> > > **Looking forward to your reply**
> > >
> > > Dear Reviewer f8dJ,
> > >
> > > Thank you for your insightful feedback. We have addressed your comments by adding detailed efficiency comparisons and performing additional experiments with Mamba as a reader. We look forward to any additional feedback you may have and would greatly appreciate your timely response as we approach the rebuttal deadline. All changes are highlighted in red in the revised paper.

---

> > > > ### Comment · Reviewer_f8dJ · 2024-11-24
> > > >
> > > > Thanks for the response which resolves my concerns. I am not fully convinced by the efficiency study because seeing the document only once is not a typical retrieval setting and the authors should clearly indicate the difference between their intention of previous studies'. I have revised my score accordingly.

---

> > > > > ### Author Response · Authors · 2024-11-24
> > > > >
> > > > > Thank you for your thoughtful feedback. When encountering a long document for the first time, current approaches either chunk it for retrieval or process it entirely with a large language model. In addition to the retrieval model comparisons, our method achieves better accuracy than Llama-3.1-70B while using only a fraction of the compute. We called this retrieval because it performs retrieval-like selection of relevant spans, but you make a fair point that the term may suggest indexing benefits that are not part of our contribution. Our focus is on improving accuracy and efficiency in **first-encounter** document understanding. We will clarify this positioning in the revised paper.

---

### Author Response · Authors · 2024-12-04
**Summary of discussion period**

We thank the reviewers for their questions and comments. These have allowed us to strengthen the paper through our revisions. All reviewers increased their scores following the revisions, which focused on three main areas.

Our direct comparisons with embedding models now clearly demonstrate efficiency gains - the 130M Mamba retriever achieves higher accuracy while using 68x fewer TFLOPS than the leading 7B parameter baseline. Our larger 1.3B model achieves higher accuracy while still using 6.5x fewer TFLOPS.

We resolved the core question about whether gains came from synthetic data alone by showing that fine-tuning embedding models on our data did not yield comparable improvements. This demonstrates the gains come from our model's ability to process document-wide context.

We addressed evaluation fairness by showing that improvements over embedding models are robust to different experimental settings. We also added Llama-70B comparisons showing our approach outperforms much larger models.

The remaining concerns from other reviewers focus mainly on presentation rather than technical issues. For example, Reviewer 4h6p acknowledged that our responses "resolve most concerns" but maintains reservations about novelty framing, which we have addressed in the revised introduction. Regarding this question of novelty, prior and contemporaneous work on SSMs for retrieval are embedding models, whereas the current work is not.

The revisions have created reviewer consensus around our key contribution: achieving strong long-document understanding with a small, efficient model.

---

### Meta-Review · Area_Chair_ZDo4 · 2024-12-20

**Metareview:**

**Summary:**

The paper introduces the Mamba retriever, a model designed for efficient and effective retrieval in long document question answering. Unlike traditional embedding-based methods that require document chunking and can lose global context, the Mamba retriever processes entire documents linearly, leveraging long-range dependencies to retrieve relevant sentences for answering queries. This approach outperforms state-of-the-art models, achieving near-GPT-4o performance on documents exceeding 256k tokens. Despite its computational efficiency and superior performance, the high cost of synthetic data generation presents a limitation for scalability.

**Strength:**
- It explores fine-tuning long-context models as retrievers, demonstrating improved retrieval performance.
- It Eliminates the need for document chunking, preserving contextual relationships in long documents.
- It includes thorough ablation studies to analyze contributions of model components and training strategies.

**Weakness:**
- The method essentially fine-tunes the Mamba model for retrieval without introducing significant architectural or methodological innovation.
- The baseline retriever does not leverage synthetic data, making it unclear whether Mamba’s improvements are due to data or its ability to handle long contexts.
- Enhancements in clarity, detail, and comparative evaluation would significantly improve the paper’s quality and review scores.

**Additional Comments On Reviewer Discussion:**

After reviewing the discussion between the authors and reviewers, I concur with Reviewer wS7S's observation that "there are still areas requiring additional work and detail for the paper to fully meet the ICLR standards." While the authors have made commendable efforts during the rebuttal, I believe further revisions are necessary, and I recommend revising the paper for submission to other upcoming conferences.

---

### Decision · Program_Chairs · 2025-01-22

Reject